# A Hierarchy of Graphical Models
# for Counterfactual Inferences

**Hongshuo Yang**     **Elias Bareinboim**

Causal Artificial Intelligence Lab

Columbia University

hy2712@columbia.edu    eb@cs.columbia.edu

## Abstract

Graphical models have been widely used as parsimonious encoders of assumptions of the underlying causal system and provide a basis for causal inferences. Models encoding stronger constraints tend to require higher expressive power, which are also harder, and sometimes impossible to empirically falsify. In this paper, we introduce two new collections of distributions that include counterfactual quantities which are experimentally accessible under counterfactual randomizations. Correspondingly, we define two new classes of graphical models for encoding empirically testable constraints in these distributions. We further present a sound and complete calculus, based on counterfactual calculus, which licenses inferences in these two new models with rules that are within the empirically falsifiable boundary. Finally, we formulate a hierarchy over several graphical models based on the constraints they encode and study the fundamental trade-off between the expressive power and empirical falsifiability of different models across the hierarchy.

## 1   Introduction

Causal information is fundamental across a wide range of scientific disciplines and human decision-making, and it is increasingly recognized as a necessary ingredient for advancing AI and machine learning in enhancing robustness, interpretability, and generalizability [16, 1]. The *Pearl Causal Hierarchy* (PCH) organizes such information into three layers: the *observational*, the *interventional*, and the *counterfactual*, corresponding roughly to the ordinary human capabilities of *seeing, doing, and imagining* [16, 3]. Each layer is formalized through a distinct symbolic language and encodes causal quantities with increasingly expressive semantics. For example, consider a system with two observed variables, $X$ (*treatment*, e.g., diet) and $Y$ (*outcome*, e.g., BMI). Layer 1 ($\mathcal{L}_1$) includes *observational* distributions, like $P(y|x)$, which represents the probability of observing BMI $y$ among those who naturally follow diet $x$. Layer 2 ($\mathcal{L}_2$) contains *interventional* distributions, like $P(y|do(x))$, which represents the probability of having BMI $y$ among those who were externally assigned to diet $x$. Layer 3 ($\mathcal{L}_3$) comprises *counterfactual* distributions, like $P(y_x|x')$, which represents the probability of having BMI $y$ if the diet had been set to $x$ among those who would naturally follow diet $x'$.

When the true causal mechanism underpinning a phenomenon of interest — formally represented by a *Structural Causal Model (SCM)* — is known, all layers of the PCH are immediately computable. Unfortunately, it is rare for SCMs to be known at this level of precision in most real-world scenarios. This limitation gives rise to the field of *causal inference*, which seeks to understand the conditions under which valid inferences can be made given access to limited features and data from the causal system. The inferential process can be illustrated through the *causal inference engine* [1, Sec. 1.3.4], as illustrated in Fig. 1. The engine takes three inputs: {(1) *Query*, (2) *Data*, (3) *Model*}, each reflecting a different aspect of the underlying SCM. The *Query* specifies the causal quantity of interest, the *Data* consists of data gathered through interactions with the environment like random

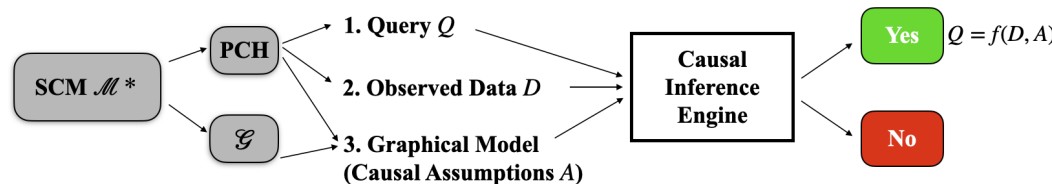

Figure 1: Unobserved SCM and the causal inference engine. The engine takes as input a query, a model, and datasets, and returns whether the query is computable from the assumptions and data.

samplings or randomized experiments, and the *Model* encodes assumptions about the SCM. A common language for articulating such assumptions is provided by *graphical models*, particularly *causal diagrams* [14, 15, 3, 1] and their variants, which encode constraints describing how different quantities within the PCH relate to one another. For example, Pearl's celebrated treatise *Probabilistic Reasoning in Intelligent Systems* developed a comprehensive account of Bayesian Networks (BN) as encoders of conditional independencies — that is, $\mathcal{L}_1$ equality constraints within an observational distribution, such as $P(y \mid x) = P(y)$ [14]. In contrast, Causal Bayesian Networks (CBN) encode equality constraints across distributions in both $\mathcal{L}_1$ and $\mathcal{L}_2$, like $P(y|do(x)) = P(y|x)$ [15, 2, 3]. Counterfactual Bayesian Networks (CTFBN) further extend this framework to encode constraints across $\mathcal{L}_3$ distributions, like $P(y_x, x) = P(y, x)$ [1, Sec. 13.2].

For a graphical model to be sufficient for supporting inference on a query, there must be a match in *expressiveness* between the model's constraints and the query, as illustrated in Fig. 2. This requirement aligns with Nancy Cartwright's famous motto "no causes in, no causes out" [5], as mathematically formalized by the *Causal Hierarchy Theorem* (CHT): to perform inferences on a quantity in layer $i$, one needs knowledge from layer $i$ or above [3, Corollary 1]. For instance, given an $\mathcal{L}_2$ query, a BN encoding only $\mathcal{L}_1$ constraints is insufficient, while a CBN encoding both $\mathcal{L}_1$ and $\mathcal{L}_2$ constraints is both sufficient and necessary for inference. A CTFBN encoding $\mathcal{L}_3$ constraints, while sufficient for the target query in $\mathcal{L}_2$, imposes assumptions that are stronger than necessary [6, 1].

While models that encode constraints higher in the PCH support inferences about more expressive queries, it is also generally preferable to avoid unnecessary assumptions for a given query. This notion of parsimony is grounded by the concept of *empirical falsification* in the sciences. As Popper emphasized, a system is scientific only if it is refutable by empirical tests [18]. At the same time, the degree of falsifiability varies across domains. In some fields, opportunities for direct refutation are abundant; in others, such as cognitive science and AI, they may be scarcer, motivating the introduction of additional assumptions that render counterfactuals articulable. This does not weaken the principle of falsifiability, but reflects a continuum of empirical accessibility across scientific disciplines. For graphical models, this continuum becomes a concrete question of empirical accessibility: can the assumptions encoded in a model be subjected to

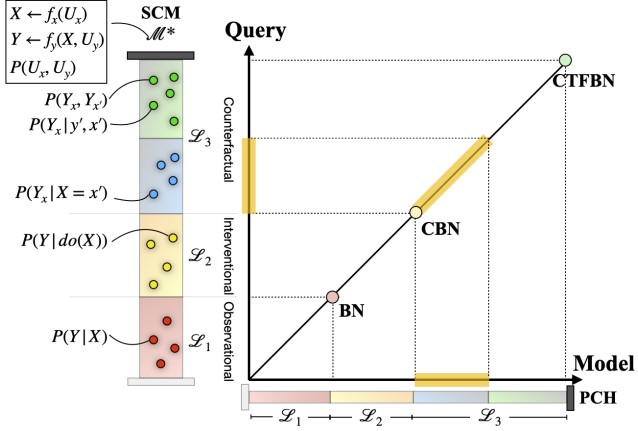

Figure 2: Expressive power of queries and graphical models along the PCH. The model's constraints should be at least as expressive as the query for the causal inference engine to work. Layer 3 is partitioned into two sub-regions: the green region represents $\mathcal{L}_3$ distributions that cannot be accessed via any experiments, while the blue region represents those that can, at least in principle, be sampled via experiments.

empirical testing with available data? Concretely, the falsifiability of an assumption in a graphical model depends on the feasibility of drawing samples from its underlying distributions (also known as *realizability*,[19]). Among the three layers of the PCH, it is generally understood that data from

$\mathcal{L}_1$ and $\mathcal{L}_2$ distributions are, at least in principle, attainable via *random sampling* and *randomized controlled trials* [8]. In contrast, $\mathcal{L}_3$ encodes counterfactual knowledge traditionally considered beyond the reach of physical experimentation. For example, the probability of necessity and sufficiency (PNS), $P(y_x, y'_{x'})$, is an $\mathcal{L}_3$ quantity that cannot be obtained via any randomized experiments. Yet, recent work by Bareinboim, Forney and Pearl have revealed that an $\mathcal{L}_3$ quantity known as the effect of the treatment on the treated (ETT), $P(y_x|x')$, can be sampled through a new experimental procedure called *counterfactual randomization* [4]. Subsequent work further refined and characterized the set of $\mathcal{L}_3$ distributions that are realizable in principle [19]. These advances reveal that $\mathcal{L}_3$ is not monolithic but instead contains distributions with varying degrees of empirical accessibility. This heterogeneity naturally raises the question of what assumptions are sufficient and necessary to support counterfactual inference — a question we now address directly.

The empirical heterogeneity of $\mathcal{L}_3$ distributions naturally raises a central question: what assumptions are sufficient and necessary to support counterfactual inference? In this paper, we addresses that question by analyzing the region between $\mathcal{L}_2$ and $\mathcal{L}_3$ in the PCH, illustrated in the orange zone of Fig. 2. We introduce formal languages, models, and inferential machinery for two families of realizable distributions that extend beyond the Fisherian interventional world yet remain empirically accessible. In doing so, we give a precise mathematical form to Cartwright's principle: only when the assumptions built into a model ("causes in") are adequate can the corresponding counterfactual queries ("causes out") be answered. Our main contributions are as follows:

(1) **Graphical Models & Inferential Machinery:** We introduce symbolic languages and valuation semantics for two new collections of distributions, each entail quantities that become experimentally accessible by a distinct implementation of *counterfactual randomization*. We then define two new classes of graphical models, CBN2.25 and CBN2.5, that encode constraints within these distributions which are amenable to empirical testing. We prove that counterfactual calculus with graphical checks form a sound and complete inferential machinery for CBN2.25 and CBN2.5.

(2) **Hierarchy of Graphical Models:** We formally define a hierarchical structure for graphical models based on constraints they encode and analyze this structure from two angles: (a) expressive power and (b) empirical falsifiability. We show that models higher in the hierarchy encode stronger assumptions that permit more expressive queries, but are increasingly harder to falsify.

**Notations.** We denote variables by capital letters, $X$, and values by small letters, $x$. Bold letters, $\mathbf{X}$ represent a set of variables and $\mathbf{x}$ a set of values. The domain of $X$ is denoted by $Val(X)$. Two values $\mathbf{x}$ and $\mathbf{z}$ are consistent if they share common values for $\mathbf{X} \cap \mathbf{Z}$. We denote by $\mathbf{x}\backslash\mathbf{Z}$ the value of $\mathbf{X}\backslash\mathbf{Z}$ consistent with $\mathbf{x}$ and by $\mathbf{x} \cap \mathbf{Z}$ the subset of $\mathbf{x}$ corresponding to variables in $\mathbf{Z}$. We assume the domain of every variable is finite. $\mathbf{W}_*$ denotes an arbitrary counterfactual event, and $\mathbf{V}(\mathbf{W}_*) = \{W \in \mathbf{V}|W_t \in \mathbf{W}_*\}$. $\mathcal{G}[\mathbf{W}]$ denotes a vertex-induced subgraph over $\mathbf{W}$. We use kinship notation for graphical relationships: parents ($Pa$), children ($Ch$), descendants ($De$), ancestors ($An$).

**Definitions and Background.** We use *Structural Causal Models* (SCM) as the underlying semantical framework [15]. An SCM $\mathcal{M}$ is a 4-tuple $\langle \mathbf{V}, \mathbf{U}, \mathcal{F}, P(\mathbf{u}))\rangle$, where $\mathbf{U}$ is a set of exogenous (latent) variables, distributed according to $P(\mathbf{u})$; $\mathbf{V}$ is a set of endogenous (observable) variables; $\mathcal{F}$ is a set of functions such that for each $V_i \in \mathbf{V}$, $f_i$ maps from a set of exogenous variables $\mathbf{U}_i \subseteq \mathbf{U}$ and a set of endogenous variables $\mathbf{Pa}_i \subseteq \mathbf{V}$ to the $Val(V_i)$ [3]. An SCM $\mathcal{M}$ induces a *causal diagram* $\mathcal{G}$ over $\mathbf{V}$ where directed edges reflect functional arguments and bidirected edges reflect shared or correlated latent confounders. We assume the model has no cyclic dependencies among variables. Two variables belong to the same *c-component* if they are connected by a path made entirely of bidirected edges.

Intervention $do(\mathbf{x})$ in an SCM $\mathcal{M}$ creates a *submodel* $\mathcal{M}_\mathbf{x}$, where functions generating $\mathbf{X}$ are replaced with constant values $\mathbf{x}$. The functions in $\mathcal{M}_\mathbf{x}$ are denoted as $\mathcal{F}_\mathbf{x}$. Given a set of variables $\mathbf{Y} \subseteq \mathbf{V}$, the solution for $\mathbf{Y}$ in $\mathcal{M}_\mathbf{x}$ defines a *potential outcome* denoted as $\mathbf{Y}_\mathbf{x}(u)$. $\|Y_\mathbf{x}\|$ denotes the *exclusion operator* such that $\|Y_\mathbf{x}\| = Y_\mathbf{z}$ with $\mathbf{Z} = \mathbf{X} \cap An(Y)_{\mathcal{G}_{\overline{\mathbf{X}}}}$, $\mathbf{z} = \mathbf{x} \cap \mathbf{Z}$ and $\mathcal{G}_{\overline{\mathbf{X}}}$ is $\mathcal{G}$ with all incoming edges into $X$ removed. An SCM $\mathcal{M}$ also induces all quantities within the *Pearl Causal Hierarchy* (PCH): for any $\mathbf{Y}, \mathbf{Z}, ..., \mathbf{X}, \mathbf{W} \subseteq \mathbf{V}$, $\mathcal{L}_1$ (Observational): $\mathbf{P}^\mathcal{M}(\mathbf{y}) = \sum_\mathbf{u} \mathbf{1}[\mathbf{Y}(\mathbf{u}) = \mathbf{y}]P(\mathbf{u})$; $\mathcal{L}_2$ (Interventional): $\mathbf{P}^\mathcal{M}(\mathbf{y}_\mathbf{x}) = \sum_\mathbf{u} \mathbf{1}[\mathbf{Y}_\mathbf{x}(\mathbf{u}) = \mathbf{y}]P(\mathbf{u})$; $\mathcal{L}_3$ (Counterfactual): $\mathbf{P}^\mathcal{M}(\mathbf{y}_\mathbf{x}, ..., \mathbf{z}_\mathbf{w}) = \sum_\mathbf{u} \mathbf{1}[\mathbf{Y}_\mathbf{x}(\mathbf{u}) = \mathbf{y}, ..., \mathbf{Z}_\mathbf{w}(\mathbf{u}) = \mathbf{z}]P(\mathbf{u})$. We denote the collection of all $\mathcal{L}_1$ distributions as $\mathbf{P}^{\mathcal{L}_1}$, the collection of all $\mathcal{L}_2$ distributions as $\mathbf{P}^{\mathcal{L}_2}$, and the collection of all $\mathcal{L}_3$ distributions as $\mathbf{P}^{\mathcal{L}_3}$.

Equalities or inequalities between polynomials over $\mathcal{L}_i$ terms represent special marks an SCM imprints on its distributions, called *invariance constraints*. A *graphical model* (also known as a

*compatibility relation*) is a pair $\langle \mathcal{G}, \mathbf{P} \rangle$, where $\mathcal{G}$ is a graph and $\mathbf{P}$ is a collection of distributions over $\mathbf{V}$. The missing edges in $\mathcal{G}$ represent certain invariance constraints within $\mathbf{P}$. Some examples of graphical models corresponding to three layers of the PCH are *Bayesian Network* (BN) [14], *Causal Bayesian Network* (CBN) [3], and *Counterfactual Bayesian Network* (CTFBN, [1]).

The *counterfactual randomization* action (CTF-RAND$(X \rightarrow \mathbf{C})^{(i)}$) [4, 19] is an experimental procedure to fix the value of $X$ as an input to functions generating $\mathbf{C} \subseteq Ch(X)$ using a randomising device having support over $Val(X)$, for unit $i$, where $Ch(X)$ stands for variables taking $X$ as an argument in their functions. A *feasible action set* describes all experimental actions allowed in a system. The *maximal feasible action set* contains all sampling, intervention and CTF-RAND actions over all variables and gives the agent the most granular experimental capabilities. More detailed background definitions and examples are provided in Appendix A for reference.

## 2 CBN2.25 and CBN2.5: Graphical Models for Realizable Constraints

In this section, we provide a fine-grained analysis of the counterfactual layer ($\mathcal{L}_3$) by circumscribing subsets of distributions that are realizable given a feasible action set. We assume all actions required to sample from any $\mathcal{L}_2$ distribution are available, together with certain counterfactual randomization capabilities. Specifically, we define two collections of realizable distributions, each determined by a different degree of flexibility in how counterfactual randomization propagates to downstream variables (Sec. 2.1). We then introduce the corresponding graphical models encoding constraints in these distribution sets (Sec. 2.2), followed by the inferential machinery for each model (Sec. 2.3).

### 2.1 Formal Languages for Realizable Counterfactual Distributions

Before introducing the two layers of language, we first provide two definitions of interventional sets that help distinguish between them.

**Definition 1** (Interventional Variable Set). *Given a set of random variables $\mathbf{V}$, an* interventional variable set *is a subset of $\mathbf{V}$ on which an intervention is performed.*

**Definition 2** (Interventional Value Set). *Given a set of random variables $\mathbf{V}$, an* interventional value set, $\mathbf{x} \in Val(\mathbf{X})$, *is a specific assignment of values to an interventional variable set $\mathbf{X} \subseteq \mathbf{V}$.*

For each interventional variable set $\mathbf{X}$, there may exist multiple corresponding interventional value sets $\mathbf{x}$ drawn from $Val(\mathbf{X})$. For example, given a binary variable $X$, its corresponding interventional variable set is $\{X\}$, while the interventional value set can either be $\{X = 0\}$ or $\{X = 1\}$.

The first collection of realizable distributions is defined under the assumption that each CTF-RAND on $X$ fixes a single value of $x$ across all its children. [1] The symbolic representation and valuation of distributions in this collection, given an SCM, are provided below.

**Definition 3** (Layer 2.25 ($\mathcal{L}_{2.25}$)). *An SCM $\mathcal{M} = \langle \mathbf{U}, \mathbf{V}, \mathcal{F}, P(\mathbf{u}) \rangle$ induces a family of joint distributions over $\mathbf{V}$, indexed by each interventional value set $\mathbf{x}$. For each $\mathbf{X}, \mathbf{Y} \subseteq \mathbf{V}$, $\mathbf{x} \in Val(\mathbf{X})$:*

$$
P^{\mathcal{M}} \left( \bigwedge_{V_i \in \mathbf{Y} \setminus \mathbf{X}} V_{i_{[\mathbf{x}_i]}} = v_i, \bigwedge_{V_i \in \mathbf{Y} \cap \mathbf{X}, \, v_i = V_i \cap \mathbf{x}} V_{i_{[\mathbf{x}_i \setminus v_i]}} = v_i \right)
$$
$$
= \sum_{\mathbf{u}} \mathbf{1} \left[ \bigwedge_{V_i \in \mathbf{Y} \setminus \mathbf{X}} V_{i_{[\mathbf{x}_i]}}(\mathbf{u}) = v_i, \bigwedge_{V_i \in \mathbf{Y} \cap \mathbf{X}, \, v_i = V_i \cap \mathbf{x}} V_{i_{[\mathbf{x}_i \setminus v_i]}}(\mathbf{u}) = v_i \right] P(\mathbf{u}).
$$

(1)

*such that (i) $\mathbf{x}_i \subseteq \mathbf{x}$ and $\bigcup_i \mathbf{x}_i = \mathbf{x}$; and (ii) for any $v_i \in \mathbf{x}$ and all $V_j \in \mathbf{Y}$, if $V_i \in An(V_j)$ in $\mathcal{M}_{\mathbf{x} \setminus V_j}$, then $v_i \in \mathbf{x}_j$. The collection of all such distributions is denoted $\mathbf{P}^{\mathcal{L}_{2.25}}$.*

Condition (i) of Def. 3 ensures that only assignments from the intervention value set $\mathbf{x}$ appear in the subscripts, and that each value in $\mathbf{x}$ appears at least once. This prevents redundancy in representing the same distribution under different intervention value sets, e.g., when $\mathbf{x} \subset \mathbf{x}'$. Condition (ii) requires all descendants of an intervened variable $X$ to share the same value $x$, unless the path from $X$ to a descendant is blocked by another variable in the intervention set. These two conditions reflect the limited flexibility allowed under the restricted counterfactual randomization action.

---

[1] For both $\mathcal{L}_{2.25}$ and $\mathcal{L}_{2.5}$, we assume that counterfactual randomization is allowed for all variables in $\mathbf{V}$.

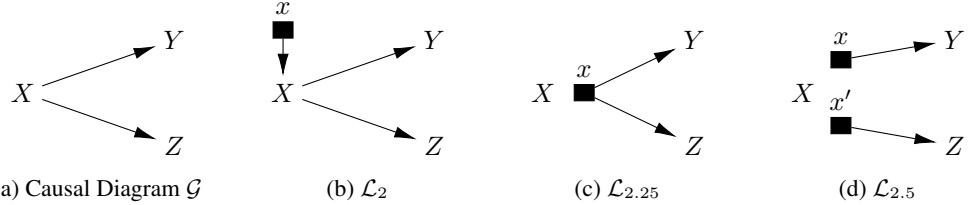

(a) Causal Diagram $\mathcal{G}$      (b) $\mathcal{L}_2$      (c) $\mathcal{L}_{2.25}$      (d) $\mathcal{L}_{2.5}$

Figure 3: Differences in how intervention on $X$ affects downstream variables in $\mathcal{L}_2$, $\mathcal{L}_{2.25}$ and $\mathcal{L}_{2.5}$.

The second collection of distributions relaxes this restriction by allowing each child of $X$ to receive a potentially different randomized value. This more flexible form of counterfactual randomization expands the class of realizable distributions beyond those in the first collection.

**Definition 4** (Layer 2.5 ($\mathcal{L}_{2.5}$)). *An SCM $\mathcal{M} = \langle \mathbf{U}, \mathbf{V}, \mathcal{F}, P(\mathbf{u}) \rangle$ induces a family of probability distributions over $\mathbf{V}$ indexed by each interventional variable set $\mathbf{X}$. For each $\mathbf{Y}, \mathbf{X} \subseteq \mathbf{V}$:*

$$
P^{\mathcal{M}}\Big( \bigwedge_{V_i \in \mathbf{Y} \setminus \mathbf{X}} V_{i_{[\mathbf{x}_i]}} = v_i, \bigwedge_{V_i \in \mathbf{Y} \cap \mathbf{X}, \, v_i = V_i \cap \mathbf{x}} V_{i_{[\mathbf{x}_i \setminus v_i]}} = v_i \Big)
$$
$$
= \sum_{\mathbf{u}} \mathbf{1}\Big[ \bigwedge_{V_i \in \mathbf{Y} \setminus \mathbf{X}} V_{i_{[\mathbf{x}_i]}}(\mathbf{u}) = v_i, \bigwedge_{V_i \in \mathbf{Y} \cap \mathbf{X}, \, v_i = V_i \cap \mathbf{x}} V_{i_{[\mathbf{x}_i \setminus v_i]}}(\mathbf{u}) = v_i \Big] P(\mathbf{u}) \tag{2}
$$

*such that (i) $\mathbf{X}_i \subseteq \mathbf{X}$, $\mathbf{x}_i \in Val(\mathbf{X}_i)$, and $\bigcup_i \mathbf{X}_i = \mathbf{X}$; and (ii) for any $V_i$ and any $B \in \mathbf{X} \cap \mathbf{Pa}(V_i)$, and for all $V_j \in \mathbf{Y}$: if $V_i \notin \mathbf{X}_j$ and $V_i \in An(V_j)$ in $\mathcal{M}_{\mathbf{x}_j}$, then $\mathbf{x}_i \cap B = \mathbf{x}_j \cap B$. The collection of all such distributions is denoted by $\mathbf{P}^{\mathcal{L}_{2.5}}$.*

Def. 3 and Def. 4 serve as templates for enumerating distributions in the two layers. Their key distinction lies in indexing: $\mathcal{L}_{2.25}$ distributions are indexed by specific interventional value sets $\mathbf{x} \in Val(\mathbf{X})$, whereas $\mathcal{L}_{2.5}$ distributions are indexed by interventional variable sets $\mathbf{X}$. The more specific indexing in $\mathcal{L}_{2.25}$ imposes stronger restrictions on the expressiveness of its distributions, as reflected in the corresponding conditions. As in Def. 3, condition (i) of Def. 4 requires each intervention variable to appear at least once in the subscript, but it relaxes the former by allowing multiple value assignments for $\mathbf{X}$. Condition (ii) is likewise weakened, enforcing value consistency only at the level of $X$'s children rather than from $X$ itself.

**Example 1** (SCM inducing $\mathcal{L}_{2.25}/\mathcal{L}_{2.5}$). *Consider the SCM $\mathcal{M} = \langle \mathbf{U} = \{U_x, U_y, U_z\}, \mathbf{V} = \{X, Y, Z\}, \mathcal{F}, P(\mathbf{u})\rangle$, where $\mathcal{F} = \{X \leftarrow f_x(U_x); Z \leftarrow f_z(X, U_z); Y \leftarrow f_y(X, U_y)\}$ and $U_x \perp\!\!\!\perp U_z \perp\!\!\!\perp U_y$. The distribution $P(X, Y_x, Z_x)$, indexed by the interventional value set $\{X = x\}$, belongs to $\mathcal{L}_{2.25}$. It satisfies conditions in Def. 3 by having consistent subscripts $x$ across all children of $X$, i.e., $Y$ and $Z$. In contrast, the distribution $P(X, Y_x, Z_{x'})$ does not belong to $\mathcal{L}_{2.25}$ because it contains conflicting value assignments for $X$, making it unindexable by any specific interventional value set. However, it does belong to $\mathcal{L}_{2.5}$ since the conditions in Def. 4 allow different value assignments for the same variable in the intervention variable set. This difference between the two layers is illustrated in Fig. 3(c) and (d). Finally, the $\mathcal{L}_3$ distribution $P(Y_x, Y)$ lies outside both languages, as it includes the same variable $Y$ under two different submodels, which is not permitted in $\mathcal{L}_{2.25}$ or $\mathcal{L}_{2.5}$.* ∎

The evaluation processes for distributions in these two new layers are illustrated in Fig. 4, with interventional distributions ($\mathcal{L}_2$) shown on the left and full counterfactual distributions ($\mathcal{L}_3$) on the right. In $\mathcal{L}_{2.25}$ and $\mathcal{L}_{2.5}$, a variable in $\mathbf{Y}$ is always evaluated within one submodel: each intervened variable in $V_i \in \mathbf{X}$ is evaluated in its own submodel $\mathcal{M}_{\mathbf{x}_i \setminus v_i}$, while each non-intervened variable is evaluated in $\mathcal{M}_{\mathbf{x}_i}$ according to the value of $\mathbf{X}$ it receives. The submodels in $\mathcal{L}_{2.25}$ and $\mathcal{L}_{2.5}$ are further constrained by the conditions in Def. 3 and Def. 4, respectively. Comparing across layers in the PCH, $\mathcal{L}_{2.25}$ and $\mathcal{L}_{2.5}$ are more expressive than $\mathcal{L}_2$, since $\mathcal{L}_2$ evaluates all variables in $\mathbf{Y}$ within a single submodel, while $\mathcal{L}_{2.25}$ and $\mathcal{L}_{2.5}$ allow joint evaluation across multiple submodels (i.e., counterfactual worlds). At the same time, they are less expressive than the full $\mathcal{L}_3$, which imposes no such restrictions on which submodels may be joined.

## 2.2 Graphical Models

With these new collections of distributions defined, we now introduce two graphical models that encode the corresponding constraints and compatibility relations.

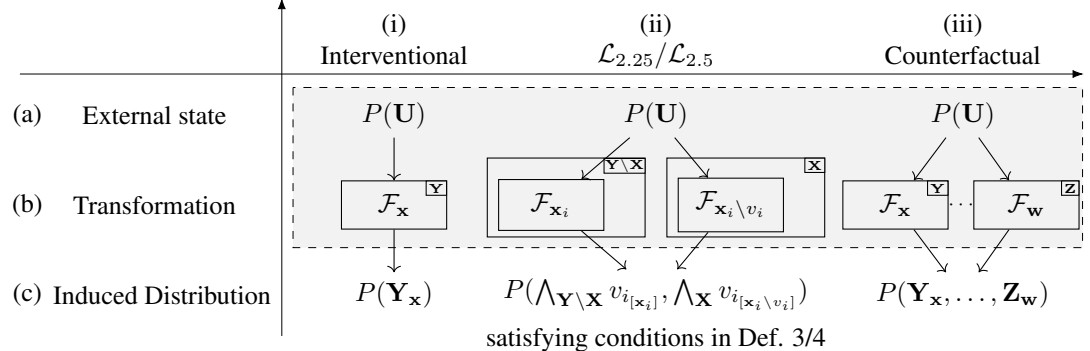

Figure 4: Given an SCM's initial state (i.e., population) (a), we show the different functional transformations (b) and the corresponding induced distribution (c) of each layer of the hierarchy. (i) represents the transformation (i.e., $\mathcal{F}$) from the natural state of the system ($P(\mathbf{U})$) to an interventional world (i.e., with modified mechanisms $\mathcal{F}_{\mathbf{x}}$), (ii) to multiple counterfactual worlds representing $\mathcal{L}_{2.25}/\mathcal{L}_{2.5}$, and (iii) to multiple counterfactual worlds with no constraints on the worlds joint.

**Definition 5** (CBN2.25 Semi-Markovian). *Given a causal diagram $\mathcal{G}$, and let $\mathbf{P}^{\mathcal{L}_{2.25}}$ be the collection of all $\mathcal{L}_{2.25}$ distributions over $\mathbf{V}$. $\mathcal{G}$ is a CBN2.25 for $\mathbf{P}^{\mathcal{L}_{2.25}}$ if the following hold:*

*(i) [Independence Restrictions] For a fixed intervention value set $\mathbf{v}$ in $Val(\mathbf{V})$ and a subset of variables $\mathbf{W} \subseteq \mathbf{V}$. Let $\mathbf{W}_*$ be the set of counterfactuals of the form $W_{\mathbf{pa}_w}$ with $\mathbf{pa}_w$ taking values in $\mathbf{v}$, $\mathbf{C}_1, ..., \mathbf{C}_l$ the c-components of $G[\mathbf{V}(\mathbf{W}_*)]$, and $\mathbf{C}_{1*}, ..., \mathbf{C}_{l*}$ the corresponding partition over $\mathbf{W}_*$. Then $P(\mathbf{W}_*)$ factorizes as:*

$$P(\bigwedge_{W_{pa_w} \in \mathbf{W}_*} W_{pa_w}) = \prod_{j=1}^{l} P(\bigwedge_{W_{pa_w} \in \mathbf{C}_{j_*}} W_{pa_w}) \tag{3}$$

*(ii) [Exclusion Restrictions] For every variable $Y \in \mathbf{V}$ with parents $\mathbf{Pa}_y$, for every set $\mathbf{Z} \subseteq \mathbf{V} \setminus (\mathbf{Pa}_y \cup \{Y\})$, and any counterfactual set $\mathbf{W}_*$ such that $P(Y_{\mathbf{pa}_y, \mathbf{z}}, \mathbf{W}_*) \in \mathbf{P}^{\mathcal{L}_{2.25}}$:*

$$P(Y_{\mathbf{pa}_y, \mathbf{z}}, \mathbf{W}_*) = P(Y_{\mathbf{pa}_y}, \mathbf{W}_*) \tag{4}$$

*(iii) [Consistency Restrictions] For every variable $Y \in \mathbf{V}$ with parents $\mathbf{Pa}_y$, $\mathbf{X} \subseteq \mathbf{Pa}_y$, for every set $\mathbf{Z} \subseteq \mathbf{V} \setminus (\mathbf{X} \cup \{Y\})$, and any counterfactual set $\mathbf{W}_*$ such that $P(Y_{\mathbf{xz}} = y, \mathbf{X}_{\mathbf{z}} = \mathbf{x}, \mathbf{W}_*) \in \mathbf{P}^{\mathcal{L}_{2.25}}$:*

$$P(Y_{\mathbf{z}} = y, \mathbf{X}_{\mathbf{z}} = \mathbf{x}, \mathbf{W}_*) = P(Y_{\mathbf{xz}} = y, \mathbf{X}_{\mathbf{z}} = \mathbf{x}, \mathbf{W}_*) \tag{5}$$

This definition closely resembles CTFBN [1], sharing the same types of constraints but restricted to distributions circumscribed by $\mathcal{L}_{2.25}$. Condition (i) requires variables not sharing latent confounders be jointly independent once their parents are fixed by intervention. Condition (ii) states that once the parents of a variable $Y$ are fixed, no further intervention can affect its value, regardless of any other observation. Finally, condition (iii) connects observations and interventions: if a parent $X$ of $Y$ is observed to be $x$ while both $X$ and $Y$ are under the same intervention $do(Z = z)$, this is equivalent to intervening on $Y$ by $do(Z = z, X = x)$. Importantly, the next proposition establishes that a causal diagram $\mathcal{G}$ induced by an SCM $\mathcal{M}$ is a CBN2.25 for the $\mathcal{L}_{2.25}$ distribution generated by $\mathcal{M}$.

**Theorem 1** ($\mathcal{L}_{2.25}$-Connection — CBN2.25). *The causal diagram $\mathcal{G}$ induced by the SCM $\mathcal{M}$ is a CBN2.25 for $\mathbf{P}^{\mathcal{L}_{2.25}}$, the collection of all $\mathcal{L}_{2.25}$ distributions induced by $\mathcal{M}$.*

**Example 2** (CBN2.25). *Given the SCM in Example 1, the fork, the pair $\langle \mathcal{G}, \mathbf{P}^{\mathcal{L}_{2.25}} \rangle$ is a CBN2.25, where $\mathcal{G}$ denotes the causal diagram in Fig. 3(a) and $\mathbf{P}^{\mathcal{L}_{2.25}}$ satisfies the following constraints:*

*(i) [Independence Restrictions]*

$$P(X, Y_x, Z_x) = P(X)P(Y_x)(Z_x) \tag{6}$$

*(ii) [Exclusion Restrictions]*

$$P(X_{\mathbf{a}} = x, \mathbf{W}_*) = P(X = x, \mathbf{W}_*), \mathbf{a} \subseteq \{z, y\} \tag{7}$$

$$P(Y_{xz} = y, \mathbf{W}_*) = P(Y_x = y, \mathbf{W}_*) \tag{8}$$

$$P(Z_{xy} = z, \mathbf{W}_*) = P(Z_x = z, \mathbf{W}_*) \tag{9}$$

*(iii) [Local Consistency]*

$$P(Y = y, X = x) = P(Y_x = y, X = x) \tag{10}$$

$$P(Y_z = y, X_z = x, \mathbf{W}_*) = P(Y_{zx} = y, X_z = x, \mathbf{W}_*) \tag{11}$$

$$P(Z = z, X = x, \mathbf{W}_*) = P(Z_x = z, X = x, \mathbf{W}_*) \tag{12}$$

$$P(Z_y = z, X_y = x, \mathbf{W}_*) = P(Z_{yx} = z, X_y = x, \mathbf{W}_*) \tag{13}$$

*Here, $\mathbf{W}_*$ can be any set of counterfactual variables such that $P(\cdot) \in \mathbf{P}^{\mathcal{L}_{2.25}}$.* ∎

Similarly, a graphical model for $\mathcal{L}_{2.5}$ can be defined by imposing the same types of constraints on distributions restricted to $\mathcal{L}_{2.5}$. In this case, the causal diagram $\mathcal{G}$ induced by an SCM $\mathcal{M}$ is also a CBN2.5 for the $\mathcal{L}_{2.5}$ distributions generated by $\mathcal{M}$. Details on CBN2.5 are given in Appendix B.

## 2.3 Inferential Machinery

It can be observed that the listed constraints in the definitions of CBN2.25 and CBN2.5 are *local*: that is, they involve counterfactual variables with their parents. These local constraints serve as the building blocks to derive more *global* statements involving variables that may be far apart in the system.

**Example 3** (Local to Global Constraints)**.** *Consider the CBN2.25 from Example 2 and the distributions $P(y_z, x)$ and $P(y, x)$. One may ask how these two distributions are related, like, whether $P(y_z, x) = P(y, x)$. This relation cannot be read off directly from the model, since it does not appear as a local constraint in the basis (Example 2). However, it can be derived by composing several local constraints, as shown on the right.* ∎

$$
\begin{aligned}
P(y_z, x) &= P(y_z, x_z) & (Eq.(7)) & \quad(14)\\
&= P(y_{xz}, x_z) & (Eq.(11)) & \quad(15)\\
&= P(y_x, x_z) & (Eq.(8)) & \quad(16)\\
&= P(y_x, x) & (Eq.(7)) & \quad(17)\\
&= P(y, x) & (Eq.(10)) & \quad(18)
\end{aligned}
$$

The inferential machinery associated with a graphical model facilitates the process of composing the local constraints and determining whether a query can be expressed as a function of the available data. For $\mathcal{L}_1$ assumptions, the standard machinery for the probabilistic constraints encoded in a BNs is *d-separation* [14]. For CBNs ($\mathcal{L}_2$), Pearl's celebrated *do-calculus* serves this role [15], while for CTFBNs ($\mathcal{L}_3$), the corresponding tool is the *ctf-calculus* [6]. As discussed earlier, the key distinction between CBN2.25/CBN2.5 and CTFBN lies in the distributions to which the constraints apply. Building on *ctf-calculus*, we develop an inferential machinery for CBN2.25 and CBN2.5 by restricting the rules to distributions in their respective layers.

**Definition 6** (Counterfactual Calculus (ctf-calculus) for CBN2.25(CBN2.5))**.** *Let $\mathcal{G}$ be a CBN2.25 (CBN2.5) for $\mathbf{P}^{\mathcal{L}_{2.25}}$ ($\mathbf{P}^{\mathcal{L}_{2.5}}$), then $\mathbf{P}^{\mathcal{L}_{2.25}}$ ($\mathbf{P}^{\mathcal{L}_{2.5}}$) satisfies the Counterfactual-Calculus rules according to $\mathcal{G}$. Namely, for any disjoint sets $\mathbf{X}, \mathbf{Y}, \mathbf{Z}, \mathbf{W}, \mathbf{T}, \mathbf{R} \subseteq \mathbf{V}$ the following three rules hold:*

**Rule 1** *(Consistency Rule - Observation/Intervention Exchange)*

$$P(\mathbf{y}_{\mathbf{T}_*\mathbf{x}}, \mathbf{x}_{\mathbf{T}_*}, \mathbf{w}_*) = P(\mathbf{y}_{\mathbf{T}_*}, \mathbf{x}_{\mathbf{T}_*}, \mathbf{w}_*) \tag{19}$$

**Rule 2** *(Independence Rule - Adding/Removing Counterfactual Observations)*

$$P(\mathbf{y}_\mathbf{r}|\mathbf{x}_\mathbf{t}, \mathbf{w}_*) = P(\mathbf{y}_\mathbf{r}|\mathbf{w}_*) \; if \; (\mathbf{Y}_\mathbf{r} \perp\!\!\!\perp \mathbf{X}_\mathbf{t}|\mathbf{W}_*) \; in \; \mathcal{G}_A \tag{20}$$

**Rule 3** *(Exclusion Rule - Adding/Removing Interventions)*

$$P(\mathbf{y}_{\mathbf{x}\mathbf{z}}, \mathbf{w}_*) = P(\mathbf{y}_\mathbf{z}, \mathbf{w}_*) \; if \; (\mathbf{X} \cap An(\mathbf{Y}) = \emptyset) \; in \; \mathcal{G}_{\overline{\mathbf{Z}}} \tag{21}$$

*where $\mathcal{G}_A$ is the AMWN $\mathcal{G}_A(\mathcal{G}, \mathbf{Y}_\mathbf{r} \cup \mathbf{X}_\mathbf{t} \cup \mathbf{W}_*)$ [2], and all $P(\cdot)$ in the rules belong to $\mathbf{P}^{\mathcal{L}_{2.25}}$ ($\mathbf{P}^{\mathcal{L}_{2.5}}$).*

The three rules of the calculus can be viewed as global counterparts to the three conditions in the definitions of CBN2.25 and CBN2.5. It is important to note that the ctf-calculus rules must, for each model, be restricted to distributions in the appropriate layer. For example, the calculus for CBN2.25 is limited to distributions in $\mathbf{P}^{\mathcal{L}_{2.25}}$, while for CBN2.5 it is limited to $\mathbf{P}^{\mathcal{L}_{2.5}}$. To enforce this, we introduce a graphical check to verify that all $P(\cdot)$ appearing in the rules correspond to valid distributions in the given model. In $\mathcal{L}_{2.5}$, it checks the counterfactual ancestor set, whereas in $\mathcal{L}_{2.25}$ it must also examine the descendants of ancestors, since the more restrictive CTF-RAND imposes stronger consistency requirements across downstream variables sharing the same intervened parents.

---

[2]Definition and algorithm for Ancestral Multi-World Network (AMWN) are given in Appendix B.

**Definition 7** (Counterfactual Reachability Set). *Given a graph $\mathcal{G}$ and a potential outcome $Y_{\mathbf{x}}$, the counterfactual reachability set of $Y_{\mathbf{x}}$, denoted $CRS(Y_{\mathbf{x}})$, consists of each $\|W_{\mathbf{x}}\|$ such that $W \in (An(Y) \cup \{De(V) : \forall V \in \mathbf{X}\}) \setminus \mathbf{X}$, and $\|W_{\mathbf{x} \setminus w}\|$ such that $W \in (An(Y) \cup \{De(V) : \forall V \in \mathbf{X}\}) \cap \mathbf{X}$. For a set $\mathbf{W}_*$, $CRS(\mathbf{W}_*)$ is defined as the union of the CRS of each potential outcome in the set, with the following merging rule: if $\{W_{i_{[\mathbf{x}_i]}}\}_i \subseteq \mathbf{W}_*$ have CRS sets containing counterfactual variables $\{R_{[\mathbf{x}_i]}\}_i$ over the same variable $R$, then $\{R_{[\mathbf{x}_i]}\}_i$ are merged into a single variable $\|R_{[\cup_i \mathbf{x}_i]}\|$ whenever $\|W_{i_{[\cup_i \mathbf{x}_i]}}\| = W_{i_{[\mathbf{x}_i]}}$ for all $i$.*

**Lemma 1.** *A distribution $Q = P(\mathbf{W}_*)$ induced by any SCM compatible with a given graph $\mathcal{G}$ belongs to: (a) $\underline{\mathcal{L}_{2.25}}$ if and only if $CRS(\mathbf{W}_*)$ satisfies: (i) it does not contain any pair of potential outcomes $W_{\mathbf{s}}, \overline{W_{\mathbf{t}}}$ of the same variable $W$ under different regimes ($\mathbf{s} \neq \mathbf{t}$); and (ii) it does not contain any pair of potential outcomes $R_{\mathbf{s}}, W_{\mathbf{t}}$ with inconsistent subscripts, i.e., $\mathbf{s} \cap \mathbf{T} \neq \mathbf{t} \cap \mathbf{S}$. (b) $\underline{\mathcal{L}_{2.5}}$ if and only if $An(\mathbf{W}_*)$ does not contain any pair of potential outcomes $W_{\mathbf{s}}, W_{\mathbf{t}}$ of the same variable $W$ under different regimes ($\mathbf{s} \neq \mathbf{t}$).*

**Example 4** (CRS Check - Not in $\mathcal{L}_{2.25}$). *Consider the causal diagram in Fig. 3(a) and whether $P(Z_x, Y_{x'})$ belongs to $\mathcal{L}_{2.25}$ induced by the corresponding SCM. The reachability set is $CRS(Z_x, Y_{x'}) = \{X, Z_x, Y_x, Z_{x'}, Y_{x'}\}$. Since the joint counterfactual $\{Z_x, Z_{x'}\}$ appears in this set with $Z$ under different regimes, Lemma 1 implies that $P(Z_x, Y_{x'})$ is not an $\mathcal{L}_{2.25}$ distribution.* ∎

With Lemma 1 ensuring that the relevant distributions lie within the corresponding layers, we can apply the ctf-calculus in CBN2.25 and CBN2.5.

**Theorem 2** (Soundness and Completeness for CBN2.25/CBN2.5 Identifiability). *An $\mathcal{L}_{2.25}$ or $\mathcal{L}_{2.5}$ quantity $Q$ is identifiable from a given set of observational and interventional distributions and a causal diagram $\mathcal{G}$ if and only if there exists a sequence of applications of the rules of the ctf-calculus for CBN2.25/CBN2.5, together with the probability axioms restricted to $\mathcal{L}_{2.25}/\mathcal{L}_{2.5}$, that reduces $Q$ to a function of the available distributions.*

$$X \qquad Y_x$$

Figure 5: AMWN $\mathcal{G}_A(\mathcal{G}, \{Y_x, X\})$ of $\mathcal{G}$ in Fig.3 (a).

**Example 5** (Effect of the Treatment on the Treated). *Consider the causal diagram $\mathcal{G}$ in Fig. 3(a) and the effect of treatment on the treated (ETT), defined as $Q = P(y_x \mid x')$, with observational distribution $P(\mathbf{v})$ as input. $Q$ can be derived using the ctf-calculus rules as follows:*

$$P(y_x | x') = P(y_x) \qquad \text{(Rule 2: } Y_x \perp\!\!\!\perp X \text{ in } \mathcal{G}_A(\mathcal{G}, \{Y_x, X\}) \text{ Fig. 5)} \qquad (22)$$

$$= P(y_x | x) \qquad \text{(Rule 2: } Y_x \perp\!\!\!\perp X \text{ in } \mathcal{G}_A(\mathcal{G}, \{Y_x, X\}) \text{ Fig. 5)} \qquad (23)$$

$$= P(y | x) \qquad \text{(Rule 1: Consistency).} \qquad (24)$$

*Steps Eq. (22) and (23) follow from Lemma 1, since $CRS(X, Y_x) = \{X, Z_x, Y_x\}$ is in $\mathcal{L}_{2.25}$.* ∎

## 3 Hierarchy of Graphical Models

In this section, we develop a refined view of the PCH by incorporating the two new graphical models that allow counterfactual inference between $\mathcal{L}_2$ and the full $\mathcal{L}_3$. We then illustrate how models in the hierarchy differ in the queries they support and in the falsifiability of the assumptions they encode. First, note that the two new collections of distributions can be positioned naturally within the PCH.

**Theorem 3** (PCH*, or Augmented PCH). *Given an SCM $\mathcal{M}$ and its induced collections of observational ($\mathbf{P}^{\mathcal{L}_1}$), interventional ($\mathbf{P}^{\mathcal{L}_2}$), $\mathcal{L}_{2.25}$ ($\mathbf{P}^{\mathcal{L}_{2.25}}$), $\mathcal{L}_{2.5}$ ($\mathbf{P}^{\mathcal{L}_{2.5}}$), and counterfactual ($\mathbf{P}^{\mathcal{L}_3}$) distributions:*

$$\mathbf{P}^{\mathcal{L}_1} \subseteq \mathbf{P}^{\mathcal{L}_2} \subseteq \mathbf{P}^{\mathcal{L}_{2.25}} \subseteq \mathbf{P}^{\mathcal{L}_{2.5}} \subseteq \mathbf{P}^{\mathcal{L}_3}. \qquad (25)$$

The illustration in Fig. 6 provides a global view of the components involved in the analysis. The SCM $\mathcal{M}^*$ sits at the top of the generative process and induces both the PCH distributions on the left and the causal diagram on the right. This augments the original PCH by explicitly incorporating the intermediate layers $\mathbf{P}^{\mathcal{L}_{2.25}}$ and $\mathbf{P}^{\mathcal{L}_{2.5}}$. The connection between each collection of distributions (left) and its associated graphical model (right) defines the corresponding compatibility relations. Building on this hierarchy of distributions, we turn to the constraints encoded by each graphical model. Given a causal diagram $\mathcal{G}$, the constraints it encodes arise from the interpretation of its missing edges. As we move higher in the hierarchy of graphical models, the missing edges correspond to increasingly stronger constraints on the distributions, and, by implication, to a finer partitioning of the space of SCMs, $\Omega$. This progression is illustrated in the example below.

**Example 6** (Constraints from Missing Edges). *Consider the causal diagram in Fig. 9(a). The constraints encoded by the* missing directed edge *from $Z$ to $Y$ across different layers are ($\mathcal{P}(\cdot)$ denotes the power set):*

BN: $\qquad P(Y \mid X, Z) = P(Y \mid X)$ $\hfill(26)$

CBN: $\qquad P(Y_{xz}) = P(Y_x)$ $\hfill(27)$

CBN2.25: $\qquad P(Y_{xz}, \mathbf{W}_*) = P(Y_x, \mathbf{W}_*), \ \forall \mathbf{W}_* \in \mathcal{P}(\{X, Z_x\})$ $\hfill(28)$

CBN2.5: $\qquad P(Y_{xz}, \mathbf{W}_*) = P(Y_x, \mathbf{W}_*), \ \forall \mathbf{W}_* \in \mathcal{P}(\{X, Z_x\}) \cup \mathcal{P}(\{X, Z_{x'}\})$ $\hfill(29)$

CTFBN: $\qquad P(Y_{xz}, \mathbf{W}_*) = P(Y_x, \mathbf{W}_*), \ \forall \mathbf{W}_*$ $\hfill(30)$

*Moving from BN to CBN augments the model with $\mathcal{L}_2$ constraints, and moving further to CBN2.25, CBN2.5, and CTFBN introduces $\mathcal{L}_3$ constraints. Among the counterfactual models, higher layers allow increasingly flexible forms of $\mathbf{W}_*$, corresponding to stronger assumptions.*

*The* missing bidirected edge *encodes independence constraints at different layers:*

CBN: $\quad P(Z_x) = P(Z \mid X = x), \quad P(Y_x) = P(Y \mid X = x)$ $\hfill(31)$

CBN2.25: $\quad P(Z_x, Y_x, X) = P(Z_x)\, P(Y_x)\, P(X)$ $\hfill(32)$

CBN2.5: $\quad P(Z_x, Y_{x'}, X) = P(Z_x)\, P(Y_{x'})\, P(X)$ $\hfill(33)$

CTFBN: $\quad P\left( \bigwedge_{x \in Val(X)} Z_x, \ \bigwedge_{x' \in Val(X)} Y_{x'}, \ X \right) = P\left( \bigwedge_{x \in Val(X)} Z_x \right) P\left( \bigwedge_{x' \in Val(X)} Y_{x'} \right) P(X)$

$\hfill(34)$

*As we move up the hierarchy, independence constraints involve richer sets of counterfactual variables, reflecting the stronger assumptions imposed. CBN encodes the parent do/see constraints restricted to $\mathbf{P}^{\mathcal{L}_2}$. In contrast, CBN2.25 introduces counterfactual constraints beyond $\mathcal{L}_2$, and together with consistency conditions, these imply the parent do/see restrictions of CBN. CBN2.5 uses the same constraint forms as CBN2.25 but permits more flexible subscripts in joint counterfactuals. At the top, CTFBN allows the most expressive independence constraints, spanning broad joint counterfactual distributions and implying those in CBN2.5 via marginalization.* ∎

In fact, the constraints encoded by graphical models higher in the hierarchy always subsume those of the models lower in the hierarchy. This monotonicity property defines the hierarchy of graphical models, as illustrated in Fig. 6.

**Theorem 4** (Hierarchy of Graphical Models, PCH*). *Given a causal diagram $\mathcal{G}$, the set of constraints it encodes when interpreted as a graphical model at layer $i$ is always a subset of the constraints it encodes when interpreted at a higher layer $j$, for all $i \leq j$.*

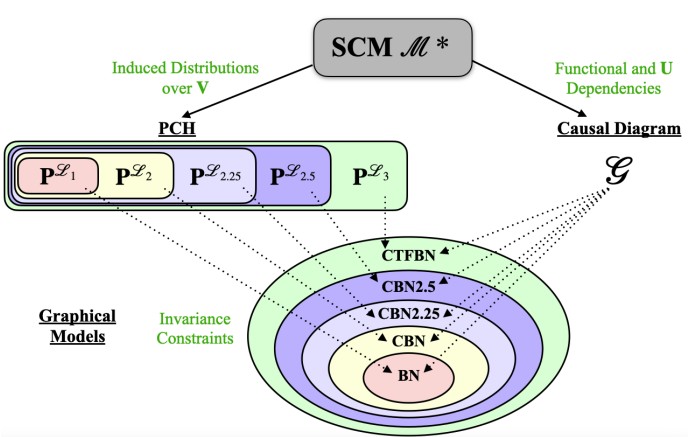

Figure 6: Pearl Causal Hierarchy (PCH*) and hierarchy of graphical models induced by an SCM

As discussed earlier in the context of Fig. 1, the causal inference engine operates by matching a query with a model that encodes a sufficient — and ideally only necessary — set of assumptions. This matching logic can be understood from two complementary perspectives.

The first concerns the *expressive power* of a model: the extent to which its assumptions are sufficient to support valid inference for a given query. When the expressiveness of the query exceeds that of the model's assumptions, the causal inference engine lacks the necessary ingredients to proceed. For

example, a BN encodes only $\mathcal{L}_1$ constraints and is therefore blind to $\mathcal{L}_2$ structure, making it unable to evaluate queries such as $P(y \mid do(x))$. Similarly, a CBN, which encodes only $\mathcal{L}_2$ constraints, cannot support inference for $\mathcal{L}_3$ queries like $P(Y_x, X)$. In contrast, when a model's assumptions are expressive enough to support the query, we say that the query and the model are *matched*. A CTFBN, which sits at the top of the hierarchy in terms of expressive power, can in principle match the most demanding queries in the PCH.[3]

The second perspective concerns the *empirical falsifiability* of a model: whether the assumptions it encodes are not only sufficient but also necessary for the query at hand. As one ascends the hierarchy, models impose increasingly stronger counterfactual constraints, many of which cannot be directly falsified with the given data collections, whether observational ($\mathbf{P}^{\mathcal{L}_1}$), interventional ($\mathbf{P}^{\mathcal{L}_2}$), or those realizable via counterfactual randomization ($\mathbf{P}^{\mathcal{L}_{2.25}}, \mathbf{P}^{\mathcal{L}_{2.5}}$). If a model encodes assumptions beyond what is strictly required, these may become empirically untestable and ontologically burdensome. Accordingly, the preferred model for a given query is the most parsimonious one that still supports valid inference, striking a balance between expressive power and empirical falsifiability.

**Example 7** (Natural Direct Effect (NDE)). *Consider $\mathcal{G}$ in Fig. 7. The natural direct effect from $X$ to $Y$, $NDE_{x,x'}(y) = P(y_{x',Z_x}) - P(y_x)$. Applying unnesting, the first term becomes $\sum_z P(y_{x'z}, z_x)$, which is ID if and only if NDE is ID. Let $Q$ be $P(y_{x'z}, z_x)$, which is an $\mathcal{L}_{2.25}$ query in this case. $Q$ can be identified in the CBN2.5 associated with $\mathcal{G}$ via ctf-calculus as $P(y_{x'z}, z_x) = P(y|x', z)P(z|x)$. A CTFBN, which encodes stronger constraints than CBN2.5, can also identify $Q$, but it brings in unnecessary assumptions such as $P(Z_x, Z_{x'}, X) = P(Z_x, Z_{x'})P(X)$, which cannot be empirically falsified with current data collection methods. In contrast, a CBN is not expressive enough to represent $Q$ at all. This example highlights the parsimony dimension: although both CBN2.5 and CTFBN can in principle support the query, the more parsimonious CBN2.5 is preferable since it avoids unnecessary, unfalsifiable commitments.* ∎

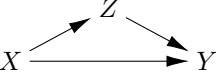

Figure 7: Causal diagram for NDE

This example highlights the trade-off between the expressiveness of queries and the parsimony of models. The optimal match occurs when the assumptions in the model are both sufficient and necessary for the intended inference (illustrated in Table 1). The introduction of CBN2.25 and CBN2.5 refines the necessity boundaries in $\mathcal{L}_3$, ensuring that queries in $\mathcal{L}_{2.25}$ and $\mathcal{L}_{2.5}$ can be matched with more parsimonious and empirically testable models. In short, higher models in the hierarchy gain inferential power by encoding constraints over increasingly expressive distributions, but at the expense of falsifiability. It is therefore crucial for researchers to choose models that strike an appropriate balance between expressive power and empirical testability for the task at hand.

| Q Layer | GM | Suff. | Nec. |
|---------|------|-------|------|
| $\mathcal{L}_{2.5}$ | CBN | x | ✓ |
| $\mathcal{L}_{2.5}$ | CBN2.5 | ✓ | ✓ |
| $\mathcal{L}_{2.5}$ | CTFBN | ✓ | x |

Table 1: Examples of Matching between Graphical Models and Queries. 'Suff.':= Sufficient and 'Nec.':= Necessary

## 4 Conclusions

In this paper, we introduced two new classes of graphical models, CBN2.25 and CBN2.5, which encode constraints over distinct collections of distributions realizable under counterfactual randomization. We showed that these models are naturally induced by SCMs (Thm. 1) and established a sound and complete inferential machinery for them (Thm. 2). We then placed the new distribution classes within the PCH (Thm. 3) and proved that graphical models over the PCH form a hierarchy (Thm. 4). Finally, we highlighted the trade-off between expressive power and empirical falsifiability across the hierarchy. Taken together, these results refine the landscape of graphical models and provide a more nuanced map of the space between $\mathcal{L}_2$ and $\mathcal{L}_3$. They also offer practical guidance for selecting models that balance inferential power with empirical testability, while accounting for the societal risks and ethical considerations inherent to each application domain. We further hope that these results serve as a theoretical foundation for future work on broader empirical evaluation and computational cost analysis of counterfactual graphical models across diverse domains.

---

[3]This does not immediately imply that the query is identifiable, only that it can be represented in principle.

## Acknowledgments and Disclosure of Funding

This research is supported in part by the NSF, ONR, AFOSR, DoE, Amazon, JP Morgan, and The Alfred P. Sloan Foundation. We thank Arvind Raghavan for their thoughtful comments.

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

# Appendices

## Contents

# A  Background and Definitions

## A.1  SCMs and Graphical Models

In this section, we review key concepts and definitions that are fundamental to this work.

We adopt *Structural Causal Models* (SCMs) as the baseline generative framework, following the presentation in [3]. The discussion there is more detailed and may be consulted if additional context on these foundational notions is needed.

**Definition 8** (Structural Casual Model (SCM) [3]). *A structural causal model $\mathcal{M}$ is a 4-tuple $\langle \mathbf{U}, \mathbf{V}, \mathcal{F}, P(\mathbf{U}) \rangle$, where*

- $\mathbf{U}$ *is a set of background variables, also called exogenous variables, that are determined by factors outside the model;*

- $\mathbf{V}$ *is a set $\{V_1, V_2, \ldots, V_n\}$ of variables, called endogenous, that are determined by other variables in the model — that is, variables in $\mathbf{U} \cup \mathbf{V}$;*

- $\mathcal{F}$ *is a set of functions $\{f_1, f_2, \ldots, f_n\}$ such that each $f_i$ is a mapping from (the respective domains of) $U_i \cup \mathbf{Pa}_i$ to $V_i$, where $U_i \subseteq \mathbf{U}$, $\mathbf{Pa}_i \subseteq \mathbf{V} \setminus V_i$, and the entire set $\mathcal{F}$ forms a mapping from $\mathbf{U}$ to $\mathbf{V}$. That is, for $i = 1, \ldots, n$, each $f_i \in \mathcal{F}$ is such that*

$$v_i \leftarrow f_i(\mathbf{pa}_i, \mathbf{u}_i), \tag{35}$$

*i.e., it assigns a value to $V_i$ that depends on (the values of) a select set of variables in $\mathbf{U} \cup \mathbf{V}$; and*

- $P(\mathbf{U})$ *is a probability function defined over the domain of $\mathbf{U}$.*

Intervention in an SCM can be viewed as a modification of the model by changing the mechanism of the intervened variables, while keeping all other components of the SCM intact.

**Definition 9** (Submodel — "Interventional SCM" [15]). *Let $\mathcal{M}$ be a structural causal model, $\mathbf{X}$ a set of variables in $\mathbf{V}$, and $\mathbf{x}$ a particular realization of $\mathbf{X}$. A submodel $\mathcal{M}_{\mathbf{x}}$ of $\mathcal{M}$ is the causal model*

$$\mathcal{M}_{\mathbf{x}} = \langle \mathbf{U}, \mathbf{V}, \mathcal{F}_{\mathbf{x}}, P(\mathbf{U}) \rangle, \tag{36}$$

*where*

$$\mathcal{F}_{\mathbf{x}} = \{f_i : V_i \notin \mathbf{X}\} \cup \{\mathbf{X} \leftarrow \mathbf{x}\}. \tag{37}$$

The impact of the intervention on an outcome variable $Y$ is commonly called the potential outcome:

**Definition 10** (Potential Outcomes [15]). *Let $\mathbf{X}$ and $\mathbf{Y}$ be two sets of variables in $\mathbf{V}$, and $\mathbf{u}$ be a unit. The potential outcome $\mathbf{Y}_{\mathbf{x}}(\mathbf{u})$ is defined as the solution for $\mathbf{Y}$ of the set of equations $\mathcal{F}_{\mathbf{x}}$ with respect to SCM $\mathcal{M}$ (or, $\mathbf{Y}_{\mathcal{M}_{\mathbf{x}}}(\mathbf{u})$). That is, $\mathbf{Y}_{\mathbf{x}}(\mathbf{u}) \stackrel{\triangle}{=} \mathbf{Y}_{\mathcal{M}_{\mathbf{x}}}(\mathbf{u})$.*

An SCM induces observational, interventional, and counterfactual distributions over the endogenous variables, which form three layers known as the Pearl Causal Hierarchy (PCH).

**Definition 11** (Pearl Causal Hierarchy (PCH) ([3]) ). *An SCM $\mathcal{M} = \langle \mathbf{U}, \mathbf{V}, \mathcal{F}, P(\mathbf{u}) \rangle$ induces three layers of probability distributions that form the Pearl Causal Hierarchy. For any $\mathbf{Y}, \mathbf{Z}, ..., \mathbf{X}, \mathbf{W} \subseteq \mathbf{V}$, the three layers of distributions are given by:*

- $\mathcal{L}_1$ *(Observational):*

$$\mathbf{P}^{\mathcal{M}}(\mathbf{y}) = \sum_{\mathbf{u}} \mathbf{1}[\mathbf{Y}(\mathbf{u}) = \mathbf{y}] P(\mathbf{u}) \tag{38}$$

- $\mathcal{L}_2$ *(Interventional):*

$$\mathbf{P}^{\mathcal{M}}(\mathbf{y_x}) = \sum_{\mathbf{u}} \mathbf{1}[\mathbf{Y_x}(\mathbf{u}) = \mathbf{y}] P(\mathbf{u}) \tag{39}$$

- $\mathcal{L}_3$ *(Counterfactual):*

$$\mathbf{P}^{\mathcal{M}}(\mathbf{y_x}, ..., \mathbf{z_w}) = \sum_{\mathbf{u}} \mathbf{1}[\mathbf{Y_x}(\mathbf{u}) = \mathbf{y}, ..., \mathbf{Z_w}(\mathbf{u}) = \mathbf{z}] P(\mathbf{u}) \tag{40}$$

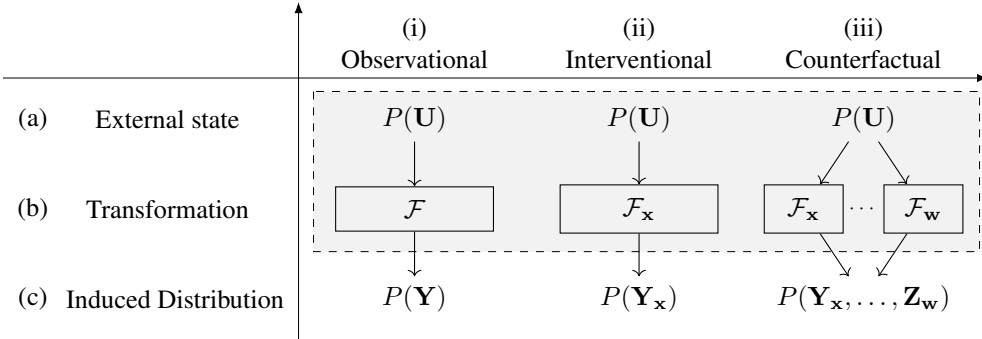

Figure 8: Given an SCM's initial state (i.e., population) (a), we show the different functional transformations (b) and the corresponding induced distribution (c) of each layer of the hierarchy. (i) represents the transformation (i.e., $\mathcal{F}$) from the natural state of the system ($P(\mathbf{U})$) to an observational world, (ii) to an interventional world (i.e., with modified mechanisms $\mathcal{F}_{\mathbf{x}}$), and (iii) to multiple counterfactual worlds (i.e., with multiple modified mechanisms).

*The collection of all $\mathcal{L}_1$ (Observational) is denoted as $\mathbf{P}^{\mathcal{L}_1}$, the collection of all $\mathcal{L}_2$ (Interventional) is denoted as $\mathbf{P}^{\mathcal{L}_2}$, and the collection of all $\mathcal{L}_3$ (Counterfactual) is denoted as $\mathbf{P}^{\mathcal{L}_3}$.*

PCH specifies both the symbolic representation and the valuation of each probabilistic quantity given an underlying SCM. If the SCM is fully specified, every quantity in any layer of the PCH can be computed directly via Def. 11 (Fig.8). In the causal inference engine (Fig. 1), this correspondence is depicted by the arrow from $\mathcal{M}^*$ to the PCH.

In practice, however, only partial knowledge of the SCM is available, so only a subset of the PCH can be observed. For example, the observational distribution may be available (Fig. 1, item (2)), while the interventional distribution remains unobserved and must be queried (item (1)). Each causal inference task therefore rests on assumptions about the structural "marks" left by the SCM on its distributions. These assumptions take the form of invariance constraints, defined as follows:

**Definition 12** (Invariance Constraint). *Given an SCM $\mathcal{M}^*$, an invariance constraint is an equality or inequality between polynomials over $\mathcal{L}_i$ terms of the PCH.*

A common example is conditional independence in the observational distribution. For instance, $P(y \mid x) = P(y)$ encodes that $X$ is probabilistically independent of $Y$.

Invariance constraints can be seen as coarsening the PCH: they abstract away from specific numerical values and instead capture relationships among distributions. As more invariance constraints are included, the granularity of knowledge about the underlying SCM increases. To avoid enumerating constraints individually, we exploit graphical models, which encode them systematically and parsimoniously by linking invariance constraints to graph topology (e.g., relations among parents, neighbors, and ancestors). The natural first step in this process is to construct a causal diagram from a given SCM, since the diagram directly captures the topological relations that determine which invariances hold.

**Definition 13** (Causal Diagram [3]). *Consider an SCM $\mathcal{M} = \langle \mathbf{U}, \mathbf{V}, \mathcal{F}, P(\mathbf{u}) \rangle$. Then $\mathcal{G}$ is a causal diagram of $\mathcal{M}$ if constructed as follows:*

*(1) add a vertex for every endogenous variable in the set $\mathbf{V}$*

*(2) add an edge $V_i \longrightarrow V_j$ for every $V_i, V_j \in \mathbf{V}$ if $V_i$ appears as an argument of $f_j$*

*(3) Add a bidirected edge $V_i \leftarrow\!\!-\!\!-\!\!-\!\!\rightarrow V_j$ for every $V_i, V_j \in \mathbf{V}$ if*

   *(a) the corresponding functions $f_i, f_j$ share some common $U \in \mathbf{U}$ as an argument, or*
   *(b) the corresponding $U_i, U_j \in \mathbf{U}$ are correlated.*

The causal diagram $\mathcal{G}$ can be viewed as a non-parametric coarsening of the SCM $\mathcal{M}$: it preserves the structural signatures (i.e., the arguments of the functions and dependency relationships among exogenous variables) while abstracting away from their specific parametrization. In Fig. 1, this is depicted by the arrow from $\mathcal{M}^*$ to $\mathcal{G}$.

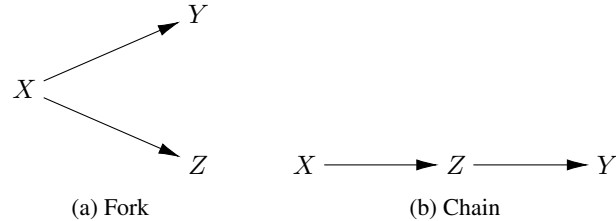

(a) Fork                                    (b) Chain

Figure 9: Two causal diagrams encoding knowledge about the causal mechanisms governing three observable variables $X$, $Z$ and $Y$.

| Layer | Graphical Model | Distribution | Physical Sampling Procedure | Inferential Machinery | Related Lit. |
|-------|-----------------|--------------|-----------------------------|----------------------|--------------|
| $\mathcal{L}_1$ | BN | $P(\mathbf{V})$ | random sampling | d-separation | [14, 15] |
| $\mathcal{L}_2$ | CBN | $P(\mathbf{V}\|do(\mathbf{x})), \forall \mathbf{x}$ | randomized controlled trials (RCT) | do-calculus | [15, 2, 3] |
| $\mathcal{L}_3$ | CTFBN | $P(\mathbf{W}_*), \forall \mathbf{W}_*$ | counterfactual randomization | ctf-calculus | [6, 1] |

Table 2: Sample graphical models and their corresponding distributions, physical sampling procedures, and inferential machinery for each layer of the PCH.

Pairing a causal diagram with the set of invariance constraints it encodes over a collection of distributions defines a graphical model, also referred to as a compatibility relation. For any SCM, its induced causal diagram $\mathcal{G}$ together with the corresponding PCH distributions naturally form such a relation. In this way, the invariance constraints encoded in $\mathcal{G}$ serve as surrogates for the empirical content of the underlying SCM, summarizing knowledge about the different layers of distributions it induces.

**Definition 14** (Graphical Model). *Let* $\mathbf{V}$ *denote a set of endogenous variables. A graphical model is a pair* $\langle \mathcal{G}, \mathbf{P} \rangle$, *where*

1. $\mathcal{G} = (\mathbf{V}, E)$ *is a graph whose nodes correspond to the variables in* $\mathbf{V}$, *and*

2. $\mathbf{P}$ *is a collection of probability distributions defined over* $\mathbf{V}$,

*such that the absence of an edge* $(V_i, V_j) \notin E$ *encodes one or more invariance constraints that are satisfied by distributions in* $\mathbf{P}$.

Depending on the assumptions made on different layers of the PCH, a different graphical model can be defined. As alluded to earlier, some examples of models corresponding to the three layers of the PCH are Bayesian Network (BN) [14], Causal Bayesian Network (CBN) [3], and Counterfactual Bayesian Network (CTFBN) [1, Sec. 13.2]. These graphical models are powerful tools for encoding assumptions to perform causal inference tasks such as identification (Fig. 1), with each model accompanied by its own inferential machinery like *d-separation* for BNs, *do-calculus* for CBNs, and *ctf-calculus* for CTFBNs [14, 15, 6].

As we ascend the PCH, the corresponding graphical models encode invariance constraints over increasingly richer sets of distributions. These enlarging sets of constraints naturally induce a hierarchy: models higher in the hierarchy support more powerful inferences but depend on stronger assumptions, which are correspondingly harder to verify empirically.

**Example 8** (SCM and Graphical Models). *Consider the SCM* $\mathcal{M} = \langle \mathbf{U} = \{U_x, U_z, U_y\}, \mathbf{V} = \{X, Z, Y\}, \mathcal{F}, P(\mathbf{u}) \rangle$, *where*

$$\mathcal{F} = \begin{cases} X \leftarrow U_x \\ Z \leftarrow X \oplus U_z \\ Y \leftarrow X \oplus U_y \end{cases} \tag{41}$$

$$P(\mathbf{u}) : U_x \sim Bernoulli(0.2), U_z \sim Bernoulli(0.4), U_y \sim Bernoulli(0.3) \tag{42}$$

*The endogenous variables* $\mathbf{V}$ *represent, respectively, a treatment* $X$ *(e.g., a diet) and outcomes* $Z$ *and* $Y$ *(e.g. patient's BMI and cholesterol level). The exogenous variables* $U_x$, $U_z$, *and* $U_y$ *represent other variables outside the model that affect* $X$, $Z$, *and* $Y$, *respectively.*

| | 1 | 2 | 3 | 4 | 5 | 6 | 7 | 8 |
|---|---|---|---|---|---|---|---|---|
| $U_x$ | 0 | 0 | 0 | 0 | 1 | 1 | 1 | 1 |
| $U_z$ | 0 | 0 | 1 | 1 | 0 | 0 | 1 | 1 |
| $U_y$ | 0 | 1 | 0 | 1 | 0 | 1 | 0 | 1 |
| $X$ | 0 | 0 | 0 | 0 | 1 | 1 | 1 | 1 |
| $Z$ | 0 | 0 | 1 | 1 | 1 | 1 | 0 | 0 |
| $Y$ | 0 | 1 | 0 | 1 | 1 | 0 | 1 | 0 |
| $Z_{X=0}$ | 0 | 0 | 1 | 1 | 0 | 0 | 1 | 1 |
| $Y_{X=0}$ | 0 | 1 | 0 | 1 | 0 | 1 | 0 | 1 |
| $Z_{X=1}$ | 1 | 1 | 0 | 0 | 1 | 1 | 0 | 0 |
| $Y_{X=1}$ | 1 | 0 | 1 | 0 | 1 | 0 | 1 | 0 |
| $X_{Z=0}$ | 0 | 0 | 0 | 0 | 1 | 1 | 1 | 1 |
| $Y_{Z=0}$ | 0 | 1 | 0 | 1 | 1 | 0 | 1 | 0 |
| $X_{Z=1}$ | 0 | 0 | 0 | 0 | 1 | 1 | 1 | 1 |
| $Y_{Z=1}$ | 0 | 1 | 0 | 1 | 1 | 0 | 1 | 0 |
| $X_{Y=0}$ | 0 | 0 | 0 | 0 | 1 | 1 | 1 | 1 |
| $Z_{Y=0}$ | 0 | 0 | 1 | 1 | 1 | 1 | 0 | 0 |
| $X_{Y=1}$ | 0 | 0 | 0 | 0 | 1 | 1 | 1 | 1 |
| $Z_{Y=1}$ | 0 | 0 | 1 | 1 | 1 | 1 | 0 | 0 |
| $X_{Z=0,Y=0}$ | 0 | 0 | 0 | 0 | 1 | 1 | 1 | 1 |
| $X_{Z=0,Y=1}$ | 0 | 0 | 0 | 0 | 1 | 1 | 1 | 1 |
| $X_{Z=1,Y=0}$ | 0 | 0 | 0 | 0 | 1 | 1 | 1 | 1 |
| $X_{Z=1,Y=1}$ | 0 | 0 | 0 | 0 | 1 | 1 | 1 | 1 |
| $Z_{X=0,Y=0}$ | 0 | 0 | 1 | 1 | 0 | 0 | 1 | 1 |
| $Z_{X=0,Y=1}$ | 0 | 0 | 1 | 1 | 0 | 0 | 1 | 1 |
| $Z_{X=1,Y=0}$ | 1 | 1 | 0 | 0 | 1 | 1 | 0 | 0 |
| $Z_{X=1,Y=1}$ | 1 | 1 | 0 | 0 | 1 | 1 | 0 | 0 |
| $Y_{X=0,Z=0}$ | 0 | 1 | 0 | 1 | 0 | 1 | 0 | 1 |
| $Y_{X=0,Z=1}$ | 0 | 1 | 0 | 1 | 0 | 1 | 0 | 1 |
| $Y_{X=1,Z=0}$ | 1 | 0 | 1 | 0 | 1 | 0 | 1 | 0 |
| $Y_{X=1,Z=1}$ | 1 | 0 | 1 | 0 | 1 | 0 | 1 | 0 |
| $P(\mathbf{u})$ | 0.336 | 0.144 | 0.224 | 0.096 | 0.084 | 0.036 | 0.056 | 0.024 |

Table 3: Mapping of events in the space of $\mathbf{U}$ to potential outcomes in the context of Example 8.

*Given the SCM, all quantities from the PCH can be computed following Def. 11, by mapping from each $\mathbf{U} = \mathbf{u}$ to all potential outcomes derived from $\mathcal{M}$, as shown in Table 3. $\mathcal{L}_1$ distributions will only involve the observed variables $X$, $Z$, and $Y$, and $\mathcal{L}_2$ distributions will have additional access to the all potential outcome variables like $Z_x$ and $Y_{x'}$, individually. $\mathcal{L}_3$ includes joint distributions over all potential outcomes in the table, capturing the full range of counterfactual dependencies.*

*However, in practice, the SCM is often not observed with such details, and we analyze the invariance constraints that hold in the distributions it induces. For example, it can be calculated from Table 3 that*

$$P(x) = P(x_y), \forall (x, y) \tag{43}$$
$$P(y_x, x) = P(y, x), \forall (x, y) \tag{44}$$

*These invariance constraints can be represented using the causal diagram shown in Fig. 9(a). When this causal diagram is interpreted as the graphical model for different layers of the PCH, it encodes different constraints according to the definitions of models:*

- *$\mathcal{L}_1$ BN:*

$$P(x, y, z) = P(x)P(z|x)P(y|x) \tag{45}$$

- *$\mathcal{L}_2$ CBN:*

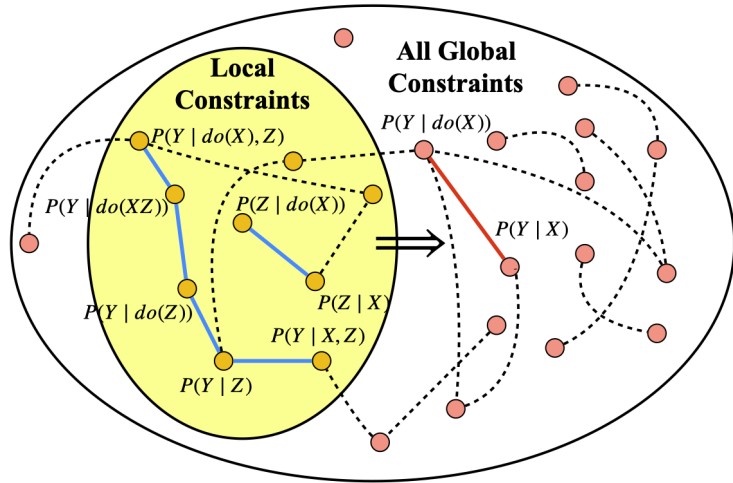

Figure 10: Constraints listed in the definition of a graphical model serves as a local basis that implies all constraints encoded in the model. Blue lines represent a set of local invariance constraints that can be composed to imply the global constraint represented by the red line.

*(i)*

$$P(x, y, z) = P(x)P(z|x)P(y|x) \tag{46}$$
$$P(y, z|do(x)) = P(y|do(x))P(z|do(x)) \tag{47}$$

*(ii)*

$$P(x|do(\mathbf{a})) = P(x), \forall \mathbf{a} \subseteq \{z, y\} \tag{48}$$
$$P(z|do(x, y)) = P(z|do(x)) \tag{49}$$
$$P(y|do(x, z)) = P(y|do(x)) \tag{50}$$

*(iii)*

$$P(z|do(x)) = P(z|x) \tag{51}$$
$$P(z|do(x, y)) = P(z|do(y), x) \tag{52}$$
$$P(y|do(x)) = P(y|x) \tag{53}$$
$$P(y|do(x, z)) = P(y|do(z), x) \tag{54}$$

- *$\mathcal{L}_3$ CTFBN:*

    *(i)*

$$P(X, Z_x, Z_{x'}, Y_x, Y_{x'}) = P(X)P(Z_x, Z_{x'})P(Y_x, Y_{x'}) \tag{55}$$

    *(ii)*

$$P(x_{\mathbf{a}}, \mathbf{w}_*) = P(x, \mathbf{w}_*), \forall \mathbf{a} \subseteq \{z, y\} \tag{56}$$
$$P(z_{xy}, \mathbf{w}_*) = P(z_x, \mathbf{w}_*) \tag{57}$$
$$P(y_{xz}, \mathbf{w}_*) = P(y_x, \mathbf{w}_*) \tag{58}$$

    *(iii)*

$$P(z, x, \mathbf{w}_*) = P(z_x, x, \mathbf{w}_*) \tag{59}$$
$$P(z_y, x_y, \mathbf{w}_*) = P(z_{xy}, x_y, \mathbf{w}_*) \tag{60}$$
$$P(y, x, \mathbf{w}_*) = P(y_x, x, \mathbf{w}_*) \tag{61}$$
$$P(y_z, x_z, \mathbf{w}_*) = P(y_{xz}, x_z, \mathbf{w}_*) \tag{62}$$

*Interestingly, the constraints explicitly listed in the definitions of graphical models represent only a subset of all the constraints implied by the model. These explicitly stated constraints are typically*

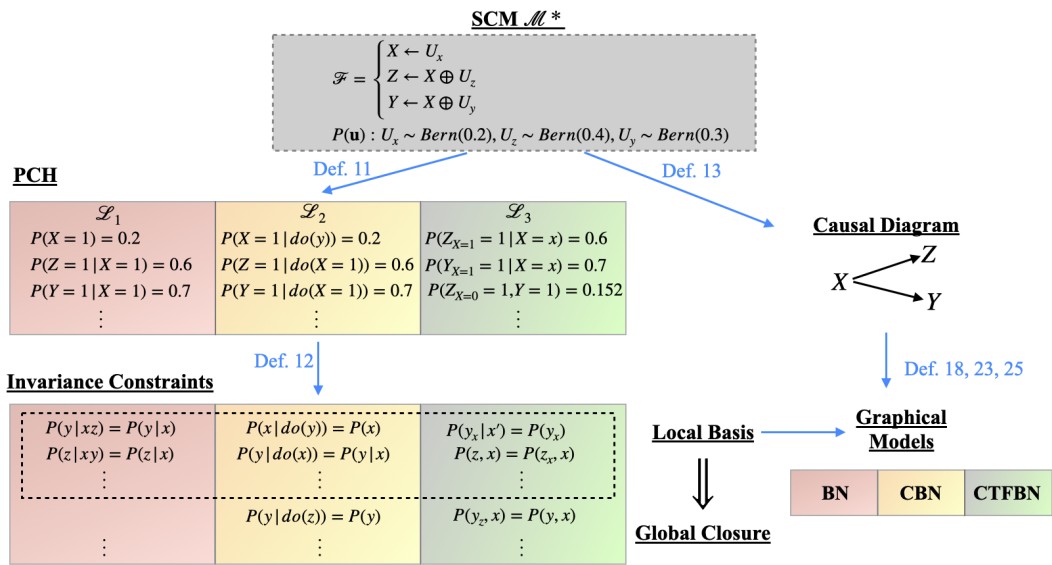

Figure 11: Illustration of how an SCM induces the PCH, invariance constraints, causal diagram, and graphical models, following Example 8.

*local, involving variables and their immediate parents. Importantly, these local constraints form a "basis" from which all other (global) constraints in the model can be spanned, as illustrated in Fig. 10. The process of composing local constraints to derive global ones underpins the operation of the causal inference engine.*

*For example, consider the constraint:*

$$P(y|do(z)) = P(y) \tag{63}$$

*This is a constraint that does not involve the parents and does not appear in the local basis, since $X$ which is in the parents of $Y$ does not appear in the expression. Still, it can be derived by combining various local constraints as shown next:*

$$P(y|do(z)) = \sum_x P(y|do(z), x)P(x|do(z)) \tag{64}$$

$$= \sum_x P(y|do(zx))P(x|do(z)) \qquad (Eq.54) \tag{65}$$

$$= \sum_x P(y|do(x))P(x|do(z)) \qquad (Eq.50) \tag{66}$$

$$= \sum_x P(y|x)P(x|do(z)) \qquad (Eq.53) \tag{67}$$

$$= \sum_x P(y|x)P(x) \qquad (Eq.48) \tag{68}$$

$$= P(y) \tag{69}$$

*The connections among these different moving parts – the SCM, the PCH, the causal diagram, the invariance constraints and the graphical model – are illustrated in Fig. 11. The constraints in each model determines its inferential power. Given the $\mathcal{L}_1$ constraints, the only inference can be drawn is that $Y$ and $Z$ are independent conditional on $X$ in the observational distributions. However, with the $\mathcal{L}_2$ constraints, the causal effect from the treatment to the outcome can be inferred, and in this case it coincides with their observational correlation (i.e. $P(y|do(x)) = P(y|x)$). If we are able to interpret the causal diagram as an $\mathcal{L}_3$ object, say a CTFBN, the local constraints can be leveraged to infer that the effect of the treatment on the treated (ETT). To witness, the ETT is also equal to the conditional distribution, $P(y_x|x') = P(y|x)$.* ∎

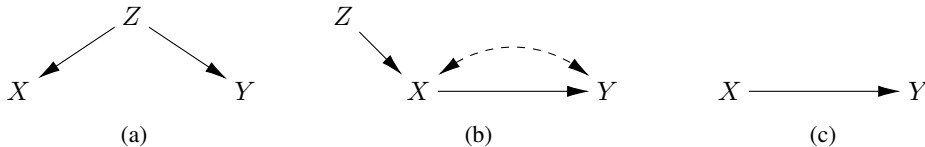

Figure 12: SCM Induced DAG or Causal Diagrams

In the following sections, we give the definitions and examples for graphical models introduced in previous works ([14, 15, 3, 1]).

## A.2 $\mathcal{L}_1$: Bayesian Networks

The first graphical model encodes invariance constraints in the observational distributions. Firstly, we formally define how to construct a graph from an SCM.

**Definition 15** (Confounded Component of an SCM [3]). *Given an SCM* $\mathcal{M} = \langle \mathbf{U}, \mathbf{V}, \mathcal{F}, P(\mathbf{u}) \rangle$, *let* $\mathbf{U}_1^c, \mathbf{U}_1^c, ..., \mathbf{U}_l^c \subseteq \mathbf{U}$ *be disjoint maximal subsets of the exogenous variables in* $\mathcal{M}$ *such that* $P(\mathbf{u}) = \prod_{k=1}^{l} P(\mathbf{U}_k^c)$. *Then, we say that* $V_i, V_j \in \mathbf{V}$ *are in the same confounded component (for short, C-component) of* $\mathcal{M}$ *if* $|\{\mathbf{U}_k^c | \mathbf{U}_k^c \cap \mathbf{U}_i \neq \emptyset, \mathbf{U}_k^c \cap \mathbf{U}_j \neq \emptyset\}| > 0$, *that is, if* $f_i$ *and* $f_j$ *have both latent arguments in some common* $\mathbf{U}_k^c$.

**Definition 16** (SCM-induced DAG [3]). *Consider an SCM* $\mathcal{M} = \langle \mathbf{U}, \mathbf{V}, \mathcal{F}, P(\mathbf{u}) \rangle$. *Then* $\mathcal{G}$ *is a DAG induced by* $\mathcal{M}$ *if it:*

- *has a vertex for every endogenous variable in the set* $\mathbf{V}$

- *has an edge* $V_i \longrightarrow V_j$ *for every* $V_i, V_j \in \mathbf{V}$ *if* $V_i$ *appears as an argument of* $f_j$

- *there exists an order over the functions in* $\mathcal{F}$ *such that for every pair* $V_i, V_j$ *in the same C-component of* $\mathcal{M}$ *such that* $f_i < f_j$, *the edge* $V_i \longrightarrow V_j$ *and the edges* $V_k \longrightarrow V_j, V_k \in \mathbf{Pa}_i$ *are in* $\mathcal{G}$.

**Definition 17** (Markov Relative to [14]). *A probability distribution* $P(\mathbf{V})$ *over a set of observed variables* $\mathbf{V}$ *is said to be Markov relative to a graph* $\mathcal{G}$ *if:*

$$P(\mathbf{V}) = \prod_i P(v_i | \mathbf{pa}_i) \tag{70}$$

*where* $\mathbf{Pa}_i = \{V_j \in \mathbf{V} | (V_j \rightarrow V_i) \in \mathcal{G}\}$.

**Definition 18** (Bayesian Network [14]). *A directed acyclic graph (DAG)* $\mathcal{G}$ *is a Bayesian Network for a probability distribution* $P$ *over the variables in* $\mathbf{V}$ *if* $P$ *is Markov relative to* $\mathcal{G}$.

**Example 9** (SCM-induced BN). *Consider the SCM* $\mathcal{M} = \langle \mathbf{U} = \{U_z, U_x, U_y\}, \mathbf{V} = \{Z, X, Y\}, \mathcal{F}, P(\mathbf{u}) \rangle$ *where*

$$\mathcal{F} = \begin{cases} Z \leftarrow U_z \\ X \leftarrow Z \vee U_x \\ Y \leftarrow Z \oplus U_y \end{cases} \tag{71}$$

$$P(\mathbf{u}) : U_z \sim Bernoulli(0.5), U_x \sim Bernoulli(0.5), U_y \sim Bernoulli(0.5) \tag{72}$$

*Its SCM-induced DAG is shown in Fig. 12(a) and its induced observational distribution* $P(\mathbf{v})$ *satisfies:*

$$P(\mathbf{v}) = P(z)P(x|z)P(y|z) \tag{73}$$

*for all* $x, y, z$ *in* $Val(X) \times Val(Y) \times Val(Z)$. *The DAG in Fig. 12(a) is a BN for* $P(\mathbf{v})$.

## A.3 $\mathcal{L}_2$: Causal Bayesian Networks

The second graphical model encodes invariance constraints in the interventional distributions.

**Definition 19** (CBN Markovian [3]). *Let* $\mathbf{P}_*$ *be the collection of all interventional distributions* $P(\mathbf{V}|do(\mathbf{x})), \mathbf{X} \subseteq \mathbf{V}, \mathbf{x} \in Val(\mathbf{X})$, *including the null intervention,* $P(\mathbf{V})$, *where* $\mathbf{V}$ *is the set of observed variables. A directed acyclic graph* $\mathcal{G}$ *is called a Causal Bayesian Network for* $\mathbf{P}_*$ *if:*

1. *[Markov]* $P(\mathbf{V})|do(\mathbf{x})$ *is Markov relative to* $\mathcal{G}$;

2. *[Missing-link] For every* $V_i \in \mathbf{V}, V_i \notin \mathbf{X}$ *such that there is no arrow from* $\mathbf{X}$ *to* $V_i$ *in* $\mathcal{G}$:
$$P(v_i|do(pa_i), do(\mathbf{x})) = P(v_i|do(pa_i)) \tag{74}$$

3. *[Parents do/see] For every* $V_i \in \mathbf{V}, V_i \notin \mathbf{X}$:
$$P(v_i|do(pa_i), do(\mathbf{x})) = P(v_i|pa_i, do(\mathbf{x})) \tag{75}$$

**Example 10** (SCM-induced CBN Markovian). *Consider the SCM from Example 9. Its induced causal diagram is shown in Fig. 12(a) and its induced set of interventional distributions* $\mathbf{P}_*$ *satisfy:*

1. *[Markov]*
$$P(\mathbf{v}) = P(z)P(x|z)P(y|z) \tag{76}$$
$$P(\mathbf{v}|do(x)) = P(z|do(x))P(y|z, do(x)) \tag{77}$$
$$P(\mathbf{v}|do(y)) = P(z|do(y))P(x|z, do(y)) \tag{78}$$
$$P(\mathbf{v}|do(z)) = P(x|do(z))P(y|do(z)) \tag{79}$$

2. *[Missing-link]*
$$P(x|do(y, z)) = P(x|do(z)) \tag{80}$$
$$P(y|do(x, z)) = P(y|do(z)) \tag{81}$$
$$P(z|do(\mathbf{a})) = P(z), \forall \mathbf{a} \subseteq \{x, y\} \tag{82}$$

3. *[Parents do/see]*
$$P(x|do(z)) = P(x|z) \tag{83}$$
$$P(x|do(y, z)) = P(x|z, do(y)) \tag{84}$$
$$P(y|do(z)) = P(y|z) \tag{85}$$
$$P(y|do(x, z)) = P(y|z, do(x)) \tag{86}$$

*The causal diagram in Fig. 12(a) is a CBN Markovian for* $\mathbf{P}_*$.

Similar to the confounded components in an SCM (Def. 15), there is also a corresponding set of confounded components in the causal diagram induced.

**Definition 20** (Confounded Component [21]). *Let* $\mathbf{C}_1, \mathbf{C}_2, ... \mathbf{C}_k$ *be a partition over the set of variables* $\mathbf{V}$, *where* $\mathbf{C}_i$ *is said to be a confounded component (for short, C-component) of* $\mathcal{G}$ *if for every* $V_i, V_j \in \mathbf{C}_i$ *there exists a path made entirely of bidirected edges between* $V_i$ *and* $V_j$ *in* $\mathcal{G}$ *and* $\mathbf{C}_i$ *is maximal.*

**Definition 21** (Augmented Parents). *Let* $<$ *be a topological order over the variables* $V_1, ..., V_n$ *in* $\mathcal{G}$, *let* $\mathcal{G}(V_i)$ *be the subgraph of* $\mathcal{G}$ *consists only of variables in* $V_1, ..., V_i$, *and let* $\mathbf{C}(V_i)$ *be the C-component of* $V_i$ *in* $\mathcal{G}(V_i)$. *The augmented parents of* $V_i$, *denoted as* $Pa_i^+$, *is the union of parents of all variables in* $\mathbf{C}(V_i)$ *that comes before* $V_i$ *in topological order:*
$$Pa_i^+ = \cup_{j|V_j \in \mathbf{T_i}} Pa_j \backslash \{V_i\} \tag{87}$$
*where* $\mathbf{T}_i = \{X \in \mathbf{C}(V_i) : X \leq V_i\}$.

We use $\mathcal{G}_{\overline{\mathbf{X}}}$ to denote the mutilated graph with all incoming edges to $\mathbf{X}$ removed from $\mathcal{G}$. The augmented parent of $V_i$ in $\mathcal{G}_{\overline{\mathbf{X}}}$ is denoted $Pa_i^{\mathbf{x}+}$.

**Example 11** (Augmented Parents). *Consider the SCM* $\mathcal{M} = \langle \mathbf{U} = \{U_z, U\}, \mathbf{V} = \{Z, X, Y\}, \mathcal{F}, P(\mathbf{u}) \rangle$ *where*

$$\mathcal{F} = \begin{cases} Z \leftarrow U_z \\ X \leftarrow Z \vee U \\ Y \leftarrow X \oplus U \end{cases} \tag{88}$$

$$P(\mathbf{u}) : U_z \sim Bernoulli(0.5), U \sim Bernoulli(0.5) \tag{89}$$

The causal diagram $\mathcal{G}$ it induces is shown in Fig. 12(b). The respective augmented parents of $X, Y, Z$ in $\mathcal{G}$ are:

$$Pa_z^+ = \{\} \tag{90}$$

$$Pa_x^+ = \{Z\} \tag{91}$$

$$Pa_y^+ = \{X, Z\} \tag{92}$$

If we consider the induced subgraph $\mathcal{G}(Y, Z)$ where there are no edges at all, it is the same graph as $\mathcal{G}_{\overline{X}}$. In this graph, nodes $Y$ and $Z$ form their own c-components respectively, so their augmented parents are both empty:

$$Pa_z^{x+} = \{\} \tag{93}$$

$$Pa_y^{x+} = \{\} \tag{94}$$

**Definition 22** (Semi-Markov Relative to [3]). *A probability $P(\mathbf{V})$ is said to be semi-Markov relative to a graph $\mathcal{G}$ if for any topological order $<$ of $\mathcal{G}$:*

$$P(\mathbf{V}) = \prod_i P(v_i | pa_i^+) \tag{95}$$

**Definition 23** (CBN Semi-Markovian [3]). *Let $\mathbf{P}_*$ be the collection of all interventional distributions $P(\mathbf{V}|do(\mathbf{x})), \mathbf{X} \subseteq \mathbf{V}, \mathbf{x} \in Val(\mathbf{X})$, including the null intervention, $P(\mathbf{V})$, where $\mathbf{V}$ is the set of observed variables. A directed acyclic graph $\mathcal{G}$ is called a Causal Bayesian Network for $\mathbf{P}_*$ if, considering $Pa_i^{\mathbf{x}+}$ in all compatible topological orders over $\mathbf{V}$:*

1. *[Semi-Markov] $P(\mathbf{V}|do(\mathbf{x}))$ is semi-Markov relative to $\mathcal{G}$;*

2. *[Missing directed-link] For every $V_i \in \mathbf{V} \backslash \mathbf{X}$, $\mathbf{W} \subseteq \mathbf{V} \backslash (Pa_i^{\mathbf{x}+} \cup \mathbf{X} \cup \{V_i\})$:*

$$P(v_i|do(\mathbf{x}), pa_i^{\mathbf{x}+}, do(\mathbf{w})) = P(v_i|do(\mathbf{x}), pa_i^{\mathbf{x}+}) \tag{96}$$

3. *[Missing bidirected-link] For every $V_i \in \mathbf{V} \backslash \mathbf{X}$, let $Pa_i^{\mathbf{x}+}$ be partitioned into two sets of confounded and unconfounded parents, $Pa_i^c$ and $Pa_i^u$ in $\mathcal{G}_{\overline{\mathbf{x}}}$::*

$$P(v_i|do(\mathbf{x}), pa_i^c, do(pa_i^u)) = P(v_i|do(\mathbf{x}), pa_i^c, pa_i^u) \tag{97}$$

**Example 12** (SCM-induced CBN Semi-Markovian). *Consider the SCM from Example 11. Its induced causal diagram is shown in Fig. 12(b) and its induced set of interventional distributions $\mathbf{P}_*$ satisfy:*

1. *[Semi-Markov]*

$$P(\mathbf{v}) = P(z)P(x|z)P(y|x, z) \tag{98}$$

$$P(\mathbf{v}|do(x)) = P(z|do(x))P(y|do(x)) \tag{99}$$

$$P(\mathbf{v}|do(y)) = P(z|do(y))P(x|z, do(y)) \tag{100}$$

$$P(\mathbf{v}|do(z)) = P(x|do(z))P(y|x, do(z)) \tag{101}$$

2. *[Missing directed-link]*

$$P(x|z, do(y)) = P(x|z) \tag{102}$$

$$P(x|do(z), do(y)) = P(x|do(z)) \tag{103}$$

$$P(y|do(x), do(z)) = P(y|do(x)) \tag{104}$$

$$P(z|do(\mathbf{a})) = P(z), \forall \mathbf{a} \subseteq \{x, y\} \tag{105}$$

3. *[Missing bidirected-link]*

$$P(x|do(z)) = P(x|z) \tag{106}$$

$$P(x|do(y, z)) = P(x|z, do(y)) \tag{107}$$

$$P(y|x, do(z)) = P(y|x, z) \tag{108}$$

*The causal diagram in Fig. 12(b) is a CBN Semi-Markovian for $\mathbf{P}_*$.*

### A.4 $\mathcal{L}_3$: Counterfactual Bayesian Networks

If we climb further up the PCH, we get another graphical model that encodes structural constraints in the counterfactual distributions.

**Definition 24** (CTFBN Markovian [1, Def. 13.2.1]). *A directed acyclic graph $\mathcal{G}$ is a Counterfactual Bayesian Network for $\mathbf{P}_{\#}$ if:*

1. *[Independence Restrictions] Let $\mathbf{W}_*$ be a set of counterfactuals of the form $W_{\mathbf{pa}_w}$, then $P(\mathbf{W}_*)$ factorizes as*

$$P(\bigwedge_{W_{pa_w} \in \mathbf{W}_*} W_{pa_w}) = \prod_{V \in \mathbf{V}(\mathbf{W}_*)} P(\bigwedge_{W_{pa_w}|W \in \mathbf{V}(\mathbf{W}_*)} W_{pa_w}) \tag{109}$$

2. *[Exclusion Restrictions] For every variable $Y \in \mathbf{V}$ with parents $\mathbf{Pa}_y$, for every set $\mathbf{Z} \subseteq \mathbf{V}\backslash(\mathbf{Pa}_y \cup \{Y\})$, and any counterfactual set $\mathbf{W}_*$, we have*

$$P(Y_{\mathbf{pa}_y, \mathbf{z}}, \mathbf{W}_*) = P(Y_{\mathbf{pa}_y}, \mathbf{W}_*) \tag{110}$$

3. *[Local Consistency] For every variable $Y \in \mathbf{V}$ with parents $\mathbf{Pa}_y$, let $\mathbf{X} \subseteq \mathbf{Pa}_y$, then for every set $\mathbf{Z} \subseteq \mathbf{V}\backslash(\mathbf{X} \cup \{Y\})$, and any counterfactual set $\mathbf{W}_*$, we have*

$$P(Y_\mathbf{z} = y, \mathbf{X}_\mathbf{z} = \mathbf{x}, \mathbf{W}_*) = P(Y_{\mathbf{xz}} = y, \mathbf{X}_\mathbf{z} = \mathbf{x}, \mathbf{W}_*) \tag{111}$$

**Example 13** (SCM-induced CTFBN Markovian). *Consider the SCM from Example 9. Its induced causal diagram is shown in Fig. 12(a) and its induced set of counterfactual distributions $\mathbf{P}_{\#}$ satisfy:*

1. *[Independence Restrictions]*

$$P(z, x_z, x'_{z'}, y_{z''}, y'_{z'''}) = P(z)P(x_z, x'_{z'})P(y_{z''}, y'_{z'''}) \tag{112}$$

2. *[Exclusion Restrictions]*

$$P(x_{yz}, \mathbf{w}_*) = P(x_z, \mathbf{w}_*) \tag{113}$$
$$P(y_{xz}, \mathbf{w}_*) = P(y_z, \mathbf{w}_*) \tag{114}$$
$$P(z_\mathbf{a}, \mathbf{w}_*) = P(z, \mathbf{w}_*), \forall \mathbf{a} \subseteq \{x, y\} \tag{115}$$

3. *[Local Consistency]*

$$P(x, z) = P(x_z, z) \tag{116}$$
$$P(x_y, z_y) = P(x_{yz}, z_y) \tag{117}$$
$$P(y, z) = P(y_z, z) \tag{118}$$
$$P(y_x, z_x) = P(y_{xz}, z_x) \tag{119}$$

*The causal diagram in Fig. 12(a) is a CTFBN Markovian for $\mathbf{P}_{\#}$.*

**Definition 25** (CTFBN Semi-Markovian [1, Def. 13.2.2]). *A directed acyclic graph $\mathcal{G}$ is a Counterfactual Bayesian Network for $\mathbf{P}_{\#}$ if:*

1. *[Independence Restrictions] Let $\mathbf{W}_*$ be a set of counterfactuals of the form $W_{\mathbf{pa}_w}$, $\mathbf{C}_1, ..., \mathbf{C}_l$ the c-components of $G[\mathbf{V}(\mathbf{W}_*)]$, and $\mathbf{C}_{1*}, ..., \mathbf{C}_{l*}$ the corresponding partition over $\mathbf{W}_*$. Then $P(\mathbf{W}_*)$ factorizes as*

$$P(\bigwedge_{W_{pa_w} \in \mathbf{W}_*} W_{pa_w}) = \prod_{j=1}^{l} P(\bigwedge_{W_{pa_w} \in \mathbf{C}_{j*}} W_{pa_w}) \tag{120}$$

2. *[Exclusion Restrictions] For every variable $Y \in \mathbf{V}$ with parents $\mathbf{Pa}_y$, for every set $\mathbf{Z} \subseteq \mathbf{V}\backslash(\mathbf{Pa}_y \cup \{Y\})$, and any counterfactual set $\mathbf{W}_*$, we have*

$$P(Y_{\mathbf{pa}_y, \mathbf{z}}, \mathbf{W}_*) = P(Y_{\mathbf{pa}_y}, \mathbf{W}_*) \tag{121}$$

3. [*Local Consistency*] *For every variable* $Y \in \mathbf{V}$ *with parents* $\mathbf{Pa}_y$, *let* $\mathbf{X} \subseteq \mathbf{Pa}_y$, *then for every set* $\mathbf{Z} \subseteq \mathbf{V} \backslash (\mathbf{X} \cup \{Y\})$, *and any counterfactual set* $\mathbf{W}_*$, *we have*

$$P(Y_{\mathbf{z}} = y, \mathbf{X}_{\mathbf{z}} = \mathbf{x}, \mathbf{W}_*) = P(Y_{\mathbf{xz}} = y, \mathbf{X}_{\mathbf{z}} = \mathbf{x}, \mathbf{W}_*) \tag{122}$$

**Example 14** (SCM-induced CTFBN Semi-Markovian). *Consider the SCM from Example 11. Its induced causal diagram is shown in Fig. 12(b) and its induced set of counterfactual distributions* $\mathbf{P}_\#$ *satisfy:*

1. [*Independence Restrictions*]

$$P(z, x_z, x'_{z'}, y_{x''}, y'_{x'''}) = P(z)P(x_z, x'_{z'}, y_{x''}, y'_{x'''}) \tag{123}$$

2. [*Exclusion Restrictions*]

$$P(x_{yz}, \mathbf{w}_*) = P(x_z, \mathbf{w}_*) \tag{124}$$
$$P(y_{xz}, \mathbf{w}_*) = P(y_x, \mathbf{w}_*) \tag{125}$$
$$P(z_{\mathbf{a}}, \mathbf{w}_*) = P(z, \mathbf{w}_*), \forall \mathbf{a} \subseteq \{x, y\} \tag{126}$$

3. [*Local Consistency*]

$$P(x, z) = P(x_z, z) \tag{127}$$
$$P(x_y, z_y) = P(x_{yz}, z_y) \tag{128}$$
$$P(y, x) = P(y_x, x) \tag{129}$$
$$P(y_z, x_z) = P(y_{xz}, x_z) \tag{130}$$

*The causal diagram in Fig. 12(b) is a CTFBN Semi-Markovian for* $\mathbf{P}_\#$.

### A.5 Counterfactual Randomization

An agent may sometimes interact with a system of interest through experiments, thereby collecting data from different layers of the PCH. Counterfactual randomization is an experimental procedure that enables an agent to observe the value of a variable before an intervention takes effect [4]. For instance, a doctor may be able to determine a patient's natural choice of drug prior to randomly assigning treatment in a clinical trial. This extension of experimental capability is formalized in the following definition of a new type of physical action that an agent may be able to perform in an environment.

**Definition 26** (Counterfactual (ctf-) Randomization (Def. 2.3 [19])). *For a variable* $X$ *and some particular unit* $i$[4] *in the target population of the environment, the operation*

$$\text{CTF-RAND}(X \to \mathbf{C})^{(i)} \tag{131}$$

*denotes fixing the value of* $X$ *as an input to the mechanisms generating* $\mathbf{C} \subseteq Ch(X)$ *for this particular unit, where* $Ch(X)$ *is the set of variables whose mechanisms take* $X$ *as an argument.*

*The value of* $X$ *is assigned randomly with support over* $\text{Domain}(X)$.

The essential differences between Fisherian randomization and $\text{CTF-RAND}(X \to \mathbf{C})^{(i)}$ are:

1. CTF-RAND does not erase unit $i$'s natural decision $X^{(i)}$[5].
2. While Fisherian randomization affects all children of $X$, CTF-RAND only affects the chosen subset $\mathbf{C} \subseteq Ch(X)$, leaving $Ch(X) \setminus \mathbf{C}$ untouched.

Importantly, CTF-RAND can only be enacted under certain structural conditions. These include environments where one can measure a unit's natural decision while simultaneously randomizing its actual decision [4], or settings where counterfactual mediators allow altering how a subset of children perceive the value of $X$ [19]. In either case, ctf-randomization enables multiple randomizations on the same variable $X$ for a single unit $i$. Further, CTF-RAND must always be applied with respect to a graphical child variable; it is not possible to bypass a child and directly alter the perception of a descendant.

---

[4]This definition discusses a unit-specific experimental procedure, as it takes a physical perspective on how an agent interacts with the units in a system.

[5]Another way to understand this difference is that the unit's natural inclination is taken into account.

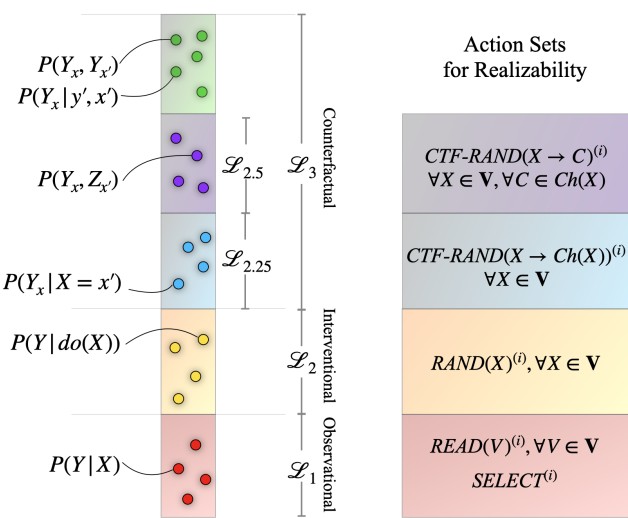

Figure 13: Hierarchy of action sets to realize distributions in different layers

**Example 15** (CTF-RAND). *Consider the SCM from Example 8. Counterfactual randomization on $X$ allows an agent to observe the natural value of $X$, say $x'$, while simultaneously assigning a specific value $x$ as input to its children $Z$ and $Y$. This is illustrated graphically in Fig. 3 (b), and as a result, the $\mathcal{L}_3$ distribution $P(X = x', Z_{X=x}, Y_{X=x})$ becomes experimentally accessible (i.e., realizable).* ∎

By including the counterfactual randomization action into our experimental toolkit, we obtain the action set that gives the agent the most granular experimental capabilities.

**Definition 27** (Maximal Feasible Action Set (SCM) [19]). *Given an SCM $\mathcal{M} = \langle \mathbf{U}, \mathbf{V}, \mathcal{F}, P(\mathbf{u}) \rangle$. The maximal feasible action set $\mathbb{A}^{\dagger}(\mathcal{M})$ is the set of all actions the agent can perform in $\mathcal{M}$ with the most granular interventional capabilities:*

(i) SELECT$^{(i)}$: *randomly choosing, without replacement, a unit $i$ from the target population, to observe in the system;*

(ii) READ$(V)^{(i)}, \forall V \in \mathbf{V}$: *measuring the way in which a causal mechanism $f_V \in \mathcal{F}$ has physically affected unit $i$, by observing its realised feature $V^{(i)}$;*

(iii) RAND$(X)^{(i)}, \forall X \in \mathbf{V}$: *erasing and replacing $i$'s natural mechanism $f_X$ for a decision variable $X$ with an enforced value drawn from a randomising device having support over Domain($X$);*

(iv) CTF-RAND$(X \to C)^{(i)}, \forall X, \forall C \in Ch(X)$: *fixing the value of $X$ as an input to the mechanisms generating $C \in Ch(X)$ using a randomising device having support over Domain($X$), for unit $i$, where $Ch(X)$ stands for the set of variables that take $X$ as an argument in their mechanisms.*

SELECT with READ correspond to random sampling. When SELECT and READ are permitted over all units and variables, all distributions in $\mathcal{L}_1$ are realizable. Adding RAND to the action set gives the agent the ability to perform randomized experiments. When SELECT, READ and RAND are permitted over all units and variables, all distributions in $\mathcal{L}_2$ are realizable. With CTF-RAND, some distributions in $\mathcal{L}_3$ also become realizable. These distributions are the ones that lie within $\mathbf{P}^{\mathcal{L}_{2.25}}$ and $\mathbf{P}^{\mathcal{L}_{2.5}}$. If we can perform all actions from the maximal feasible action set in an environment, we are able to draw samples from any distributions in $\mathbf{P}^{\mathcal{L}_{2.5}}$. For $\mathcal{L}_3$ distributions that lie beyond $\mathbf{P}^{\mathcal{L}_{2.5}}$, there is currently no known experimental procedure to sample from them. From the definitions of action sets, we observe a hierarchical structure in the feasible actions an agent can perform to access distributions at different layers, as illustrated in Fig. 13. This hierarchy of action sets match the hierarchical structure of distributions from the perspective of realizability.

# B  Details on Layers 2.25 and 2.5

## B.1  Nested Counterfactuals

The counterfactual variables in the symbolic representation of $\mathcal{L}_{2.25}$ and $\mathcal{L}_{2.5}$ are all of the form $Y_{\mathbf{x}}$, where the subscript $\mathbf{x}$ indicates that an intervention $do(\mathbf{X} = \mathbf{x})$ has been performed in the system. There is another type of counterfactual variables which represents interventions like $do(\mathbf{X} = \mathbf{X_z})$, where the variable $\mathbf{X}$ is set to behave as another counterfactual variable, say $\mathbf{X_z}$. A random variable $Y$ in such a system is represented with a counterfactual of the form $Y_{\mathbf{X_z}}$, which is called a *nested counterfactual*.

All nested counterfactuals can be unnested via the Counterfactual Unnesting (CUT) process below and be transformed into non-nested ones.

**Corollary 1** (Counterfactual Unnesting (CUT) [6]). *Let $Y, X \in \mathbf{V}, \mathbf{T}, \mathbf{Z} \subseteq \mathbf{V}$, and let $z$ be a set of values for $Z$. Then,the nested counterfactual $P(Y_{\mathbf{T}_* X_{\mathbf{z}}} = y)$ can be written with one less level of nesting as:*

$$P(Y_{\mathbf{T}_* X_{\mathbf{z}}} = y) = \sum_x P(Y_{\mathbf{T}_* x} = y, X_{\mathbf{z}} = x) \tag{132}$$

Nested counterfactuals may also belong to $\mathcal{L}_{2.25}$ and $\mathcal{L}_{2.5}$, provided that their unnested form, derived via the Counterfactual Unnesting Theorem, contains only distributions admissible within the corresponding layer.

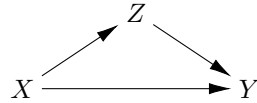

**Lemma 2** (Nested Counterfactuals in $\mathcal{L}_{2.25}/\mathcal{L}_{2.5}$). *A nested counterfactual belongs to $\mathcal{L}_{2.25}/\mathcal{L}_{2.5}$ if and only if there exists a sequence of applications of the CUT procedure that reduces it to a function of unnested counterfactuals in $\mathcal{L}_{2.25}/\mathcal{L}_{2.5}$.*

Figure 14: Causal Diagram: path $X \to Y$ represents the Natural Direct Effect (NDE)

**Example 16** (Natural Direct Effect (NDE)). *Consider the causal diagram in Fig. 14. The natural direct effect from $X$ to $Y$ can be written in counterfactual language as*

$$NDE_{x,x'}(y) = P(y_{x',Z_x}) - P(y_x) \tag{133}$$

*The first term is a nested counterfactual, and we can derive its unnested expression by applying CUT.*

$$P(y_{x',Z_x}) = \sum_z P(y_{x'z}, z_x) \tag{134}$$

*From this unnested expression, we can conclude that it is in $\mathcal{L}_{2.5}$ as $P(Y_{x'z}, Z_x)$ satisfies the conditions in Def. 4. However, it is not in $\mathcal{L}_{2.25}$ due to the conflicting subscript $x$ and $x'$ in the two counterfactual variables joint.*

## B.2  Examples for $\mathcal{L}_{2.25}$ and $\mathcal{L}_{2.5}$

Def. 3 and Def. 4 can be viewed as the template to enumerate distributions in $\mathcal{L}_{2.25}$ and $\mathcal{L}_{2.5}$. The key difference between the two layers is that $\mathcal{L}_{2.25}$ is indexed by specific interventional values, while $\mathcal{L}_{2.5}$ is indexed by interventional variables. This difference is illustrated in Example 1 where different value assignments for the interventional variable set $\mathbf{X}$ is allowed in $\mathcal{L}_{2.5}$ but not in $\mathcal{L}_{2.25}$. We further illustrate this difference in another example below.

**Example 17** (Difference in Indexing between $\mathcal{L}_{2.25}$ and $\mathcal{L}_{2.5}$). *Consider the SCM from Example 11 where the variables $\mathbf{V} = \{Z, X, Y\}$ form a chain $Z \to X \to Y$ topologically. Let the interventional variable set be $\{Y\}$.*

- *For $\mathcal{L}_{2.25}$, it is indexed by a specific interventional value. So we need to fix the value assignment of $Y$ to be $y \in Val(Y)$. Then by Def. 3, $P(Z_y, X_y, Y)$, where $Z$ and $X$ share the same subscript, is a distribution in $\mathcal{L}_{2.25}$.*

- *For $\mathcal{L}_{2.5}$, the interventional variable can take any value in its domain unless it is constrainted by Cond. (ii) of Def. 4 when two variables are descendants of the same child of an intervened value. In this example, $Z$ and $X$ are not in descendants of $Y$. As a result, there is no constraint on value assignment to $Y$ for $Z$ and $X$. Taking any $y, y' \in Val(Y)$, $P(Z_y, X_{y'}, Y)$ is a distribution in $\mathcal{L}_{2.5}$.*

*This example shows that the increased flexibility of indexing in $\mathcal{L}_{2.5}$ compared to $\mathcal{L}_{2.25}$ allows it to include more distributions.*

Cond. (i) of Def. 3 and Def. 4 ensures that all variables in the intervention set must appear at least once as subscript in the counterfactuals joint. This avoids any redundant symbolic representation to appear during the enumeration of distributions in the languages, as illustrated in example below.

**Example 18** (Cond. (i) of Def. 3 and Def. 4). *Consider the same SCM from Example 11 where the variables $\mathbf{V} = \{Z, X, Y\}$ form a chain $Z \to X \to Y$ topologically. Given two interventional variable sets $\{\}$ and $\{Y\}$.*

*The empty interventional set gives the distribution $P(Z, X, Y)$, where all subscripts are empty. This is consistent with our understanding that empty intervention is equivalent to observation. For the interventional variable set $\{Y\}$, if Cond. (i) is not imposed, $P(Z, X, Y)$ would also be compatible with the symbolic representation for distributions in these layers. This means that the same distribution is repetitively enumerated under different interventional variable sets. To avoid this redundancy, we impose Cond. (i) to require the union of all subscripts to cover the interventional variable set. In other words, $y$ must appear as a subscript in at least one of the counterfactuals joint. As a result, the enumeration would not produce $P(Z, X, Y)$, but rather, produces distributions like $P(Z_y, X, Y)$, $P(Z, X_y, Y)$ or $P(Z_y, X_y, Y)$ for $\mathcal{L}_{2.25}$, and also $P(Z_y, X_{y'}, Y)$ for $\mathcal{L}_{2.5}$.*

Cond. (ii) of Def. 3 and Def. 4 reflects how counterfactual randomization enforces consistent values over downstream variables. For $\mathcal{L}_{2.25}$, counterfactual randomization on variable $X$ is restrained such that all children of $X$ shares the same value $x$. As a result, all descendants of $X$ share the same value $x$. In contrast, counterfactual randomization in $\mathcal{L}_{2.5}$ allow each child of $X$ to interpret $X$ differently. Yet, given that counterfactual randomization cannot bypass a child to affect descendants directly, it still imposes a consistent value constraint over the descendants of $X$. This constraint starts at the children of $X$, instead of at $X$ itself.

**Example 19** (Cond. (ii) of Def. 3 and Def. 4). *Consider the causal diagram in Fig. 3(a) and the intervention on $X$. In $\mathcal{L}_2$, the submodel fixes $X = x$ and we obtain the distribution $P(Y, Z|do(x))$. In $\mathcal{L}_{2.25}$, all downstream variables of $X$ must include $x$ in its subscript, i.e., $Y_x, Z_x$. At the same time, counterfactual randomization allows us to join the natural value of $X$ with the other counterfactual variables and obtain the distribution $P(X, Y_x, Z_x)$. In $\mathcal{L}_{2.25}$, the downstream variable consistency is only enforced at the child level. In this example, different subscripts of $X$ for $Y$ and $Z$ are allowed and we obtain the distribution $P(X, Y_x, Z_{x'})$. The difference between the three layers are illustrated graphically in Fig. 3.*

## B.3 Details on CBN2.5

In this section, we give the detailed definition and theorem for CBN2.5.

**Definition 28** (CBN2.5 Semi-Markovian). *Given a mixed graph $\mathcal{G}$ and let $\mathbf{P}^{\mathcal{L}_{2.5}}$ be the collection of all $\mathcal{L}_{2.5}$ distributions. $\mathcal{G}$ is a Causal Bayesian Network 2.5 for $\mathbf{P}^{\mathcal{L}_{2.5}}$ if:*

1. *[Independence Restrictions] Let $\mathbf{W}_*$ be a set of counterfactuals of the form $W_{\mathbf{pa}_w}$ with distinct $W$, $\mathbf{C}_1, ..., \mathbf{C}_l$ the c-components of $G[\mathbf{V}(\mathbf{W}_*)]$, and $\mathbf{C}_{1*}, ..., \mathbf{C}_{l*}$ the corresponding partition over $\mathbf{W}_*$ such that $P(\mathbf{W}_*) \in \mathbf{P}^{\mathcal{L}_{2.5}}$. Then $P(\mathbf{W}_*)$ factorizes as*

$$P(\bigwedge_{W_{pa_w} \in \mathbf{W}_*} W_{pa_w}) = \prod_{j=1}^{l} P(\bigwedge_{W_{pa_w} \in \mathbf{C}_{j*}} W_{pa_w}) \tag{135}$$

2. *[Exclusion Restrictions] For every variable $Y \in \mathbf{V}$ with parents $\mathbf{Pa}_y$, for every set $\mathbf{Z} \subseteq \mathbf{V} \backslash (\mathbf{Pa}_y \cup \{Y\})$, and any counterfactual set $\mathbf{W}_*$ such that $P(Y_{\mathbf{pa}_y, \mathbf{z}}, \mathbf{W}_*) \in \mathbf{P}^{\mathcal{L}_{2.5}}$, we have*

$$P(Y_{\mathbf{pa}_y, \mathbf{z}}, \mathbf{W}_*) = P(Y_{\mathbf{pa}_y}, \mathbf{W}_*) \tag{136}$$

3. *[Local Consistency] For every variable $Y \in \mathbf{V}$ with parents $\mathbf{Pa}_y$, let $\mathbf{X} \subseteq \mathbf{Pa}_y$, then for every set $\mathbf{Z} \subseteq \mathbf{V} \backslash (\mathbf{X} \cup \{Y\})$, and any counterfactual set $\mathbf{W}_*$ such that $P(Y_{\mathbf{xz}} = y, \mathbf{X}_{\mathbf{z}} = \mathbf{x}, \mathbf{W}_*) \in \mathbf{P}^{\mathcal{L}_{2.5}}$, we have*

$$P(Y_{\mathbf{z}} = y, \mathbf{X}_{\mathbf{z}} = \mathbf{x}, \mathbf{W}_*) = P(Y_{\mathbf{xz}} = y, \mathbf{X}_{\mathbf{z}} = \mathbf{x}, \mathbf{W}_*) \tag{137}$$

**Theorem 5** ($\mathcal{L}_{2.5}$-Connection — CBN2.5 (Markovian and Semi-Markovian)). *The Causal diagram $\mathcal{G}$ induced by the SCM $\mathcal{M}$ following the constructive procedure in Def. 13 is a CBN2.5 for $\mathbf{P}^{\mathcal{L}_{2.5}}$, the collection of all $\mathcal{L}_{2.5}$ distributions induced by $\mathcal{M}$.*

**Example 20** (CBN2.5). *Consider the SCM from Example 1.*

*The Causal Diagram is induces is shown in Fig. 3(a) and the collection of realizable distributions $\mathbf{P}^{\mathcal{L}_{2.5}}$ it induces satisfies the following constraints:*

1. [*Independence Restrictions*]
$$P(X, Y_x, Z_{x'}) = P(X)P(Y_x)(Z_{x'}) \tag{138}$$

2. [*Exclusion Restrictions*]
$$P(X_{\mathbf{a}} = x, \mathbf{W}_*) = P(X = x, \mathbf{W}_*), \mathbf{a} \subseteq \{z, y\} \tag{139}$$
$$P(Y_{xz} = y, \mathbf{W}_*) = P(Y_x = y, \mathbf{W}_*) \tag{140}$$
$$P(Z_{xy} = z, \mathbf{W}_*) = P(Z_x = z, \mathbf{W}_*) \tag{141}$$

3. [*Local Consistency*]
$$P(Y = y, X = x) = P(Y_x = y, X = x) \tag{142}$$
$$P(Y_z = y, X_z = x) = P(Y_{zx} = y, X_z = x) \tag{143}$$
$$P(Z = z, X = x) = P(Z_x = z, X = x) \tag{144}$$
$$P(Z_y = z, X_y = x) = P(Z_{yx} = z, X_y = x) \tag{145}$$

### B.4 Independence Constraints and AMWN

The independence rule in ctf-calculus requires the construction of another graphical object, known as the *Ancestral Multi-World Network* (AMWN) [6]. We reproduce the algorithm for AMWN construction and the theorem stating its soundness.

---

**Algorithm 1** AMWN-CONSTRUCT($\mathcal{G}, \mathbf{W}_*$)

---

**Input:** Causal Diagram $\mathcal{G}$ and a set of counterfactual variables $\mathbf{W}_*$
**Output:** $\mathcal{G}_A(\mathbf{W}_*)$, the AMWN constructed from $\mathcal{G}$ and $\mathbf{W}_*$
 1: Initialise $\mathcal{G}'$ by adding variables in $An(\mathbf{W}_*)$ together with the directed arrows witnessing the ancestrality
 2: **for** each node $V \in \mathbf{V}$ appearing more than once in $\mathcal{G}'$ **do**
 3:     Add a node $U_V$ and an edge $U_V \to V_{\mathbf{x}}$ for every instance $V_{\mathbf{x}}$ of $V$.
 4: **end for**
 5: **for** each bidirected $V \leftarrow\!-\!-\!-\!\to W$ where $V$ and $W$ are in $\mathcal{G}'$ **do**
 6:     Add a node $U_{VW}$ and edges from it to $V_{\mathbf{x}}$ and $W_{\mathbf{x}}$ for every instance $V_{\mathbf{x}}$ of $V$ or $W_{\mathbf{x}}$ of $W$ in $\mathcal{G}'$.
 7: **end for**
        return $\mathcal{G}'$.

---

**Theorem 6** ($\mathcal{L}_3$ Independence Constraints – Counterfactual d-separation). *(Theorem 1 in [6]) Consider a causal diagram $\mathcal{G}$ and a collection of counterfactual distributions, $\mathbf{P}^{\mathcal{L}_3}$, induced by the SCM associated with $\mathcal{G}$. For counterfactual variables $X_{\mathbf{t}}, Y_{\mathbf{r}}, \mathbf{Z}_*$,*
$$(\|X_{\mathbf{t}}\| \perp\!\!\!\perp \|Y_{\mathbf{r}}\| \mid \|\mathbf{Z}_*\|)_{\mathcal{G}_A} \implies (\|X_{\mathbf{t}}\| \perp\!\!\!\perp \|Y_{\mathbf{r}}\| \mid \|\mathbf{Z}_*\|)_{\mathbf{P}^{\mathcal{L}_3}} \tag{146}$$
*In words, if $\|X_{\mathbf{t}}\|$ and $\|Y_{\mathbf{r}}\|$ are d-separated given $\|\mathbf{Z}_*\|$ in the diagram $\mathcal{G}_A(X_{\mathbf{t}}, Y_{\mathbf{r}}, \mathbf{Z}_*)$, then $\|X_{\mathbf{t}}\|$ and $\|Y_{\mathbf{r}}\|$ are independent given $\|\mathbf{Z}_*\|$ in every distribution $\mathbf{P}^{\mathcal{L}_3}$ compatible with the causal diagram $\mathcal{G}$.*

When adapting ctf-calculus to CBN2.25 and CBN2.5, there is an extra step to ensure that the distributions belong to the corresponding layers. This can be added as an extra step before Step 1 of Alg. 1 to check that:

- CBN2.25: $CRS(\mathbf{W}_*)$ satisfies Lemma 1
- CBN2.5: $An(\mathbf{W}_*)$ satisfies Lemma 1

The same check applies to the other two rules of ctf-calculus too.

# C  Proofs

## C.1  Supporting Lemmas

**Lemma C.1** (Casual Diagram of Submodel). *Given an SCM $\mathcal{M}$ and its causal diagram $\mathcal{G}$, the causal diagram induced by its submodel $\mathcal{M}_{\mathbf{x}}$ is $\mathcal{G}_{\overline{\mathbf{X}}}$, i.e., $\mathcal{G}$ with all incoming edges to $\mathbf{X}$ removed.*

*Proof.* By Def. 9, $\mathcal{M}_{\mathbf{x}}$ replaces $f_x$ with $X \leftarrow x$ for all $X \in \mathbf{X}$. As a result, $\mathbf{X}$ have no endogenous or exogenous parents. By the causal diagram construction in Def. 13, edges that point to $\mathbf{X}$ are added only when $\mathbf{X}$ have parents. Thus, there is no edges incoming to $\mathbf{X}$ in the causal diagram induced by $\mathcal{M}_{\mathbf{x}}$. In addition, given that $\mathcal{M}_{\mathbf{x}}$ keeps all other components of $\mathcal{M}$ intact, all other edges remain the same. Therefore, the causal diagram induced by $\mathcal{M}_{\mathbf{x}}$ is $\mathcal{G}$ with all incoming edges to $\mathbf{X}$ removed, denoted as $\mathcal{G}_{\overline{\mathbf{X}}}$. $\qquad\square$

**Corollary 2.** *Condition (ii) of Def. 3 and Def. 4 can be translated to an equivalent graphical condition:*

$\mathcal{L}_{2.25}$*: For any $v_i \in \mathbf{x}$, for all $V_j \in \mathbf{Y}$, if $V_i \in An(V_j)$ in $\mathcal{G}_{\overline{\mathbf{X} \setminus V_j}}$, then $v_i \in \mathbf{x}_j$.*

$\mathcal{L}_{2.5}$*: For any $V_i$, $B \in \mathbf{X} \cap \mathbf{Pa}(V_i)$, for all $V_j \in \mathbf{Y}$, if $V_i \notin \mathbf{X}_j$ and $V_i \in An(V_j)$ in $\mathcal{G}_{\overline{\mathbf{x}_j}}$, then $\mathbf{x}_i \cap B = \mathbf{x}_j \cap B$.*

*Proof.* It follows from Lemma C.1. $\qquad\square$

**Lemma C.2.** *Given a causal diagram $\mathcal{G}$ over $\mathbf{V}$ and a set of counterfactual events $\mathbf{W}_*$, if $P(\mathbf{W}_*)$ is in $\mathbf{P}^{\mathcal{L}_{2.25}}$ of all SCMs compatible with $\mathcal{G}$, then $P(\|\mathbf{W}_*\|)$ is also in $\mathbf{P}^{\mathcal{L}_{2.25}}$ of all SCMs compatible with $\mathcal{G}$.*

*Proof.* If $P(\mathbf{W}_*)$ is in $\mathbf{P}^{\mathcal{L}_{2.25}}$ of all SCMs compatible with $\mathcal{G}$, it satisfies both conditions of Def. 3. We prove that after applying the exclusion operator to $\mathbf{W}_*$, the distribution still satisfies both conditions of Def. 3.

Let the set of potential outcome variables in $\mathbf{W}_*$ be denoted as $\{W_{1_{[\mathbf{t}_1]}}, ..., W_{n_{[\mathbf{t}_n]}}\}$. $P(\mathbf{W}_*)$ is indexed by the union of subscripts of all $W_{i_{\mathbf{t}_i}} \in \mathbf{W}_*$, and we denote this index by $\mathbf{t} \triangleq \bigcup_i \mathbf{t}_i$. The exclusion operator does not add subscripts to the variable, so let the new index set be the union of subscripts of all $\|W_{i_{\mathbf{t}_i}}\| \in \|\mathbf{W}_*\|$ and denote it as $\mathbf{t}' \triangleq \bigcup_i \mathbf{t}'_i$. Cond. (i) of Def. 3 still holds.

Given that $P(\mathbf{W}_*)$ also satisfies Cond. (ii) of Def. 3 and by Cor. 2, it means that whenever there is a directed path from $T \in \mathbf{T}$ to $W_i \in V[\mathbf{W}_*]$ in $\mathcal{G}_{\overline{\mathbf{T} \setminus W}}$, $t$ is in the subscript of $W_i$, i.e. $t \in \mathbf{t}_i$. Applying the exclusion operator on $W_{i_{\mathbf{t}_i}}$ removes variables in $\mathbf{t}_i$ that does not have a directed edge to $W_i$ in $\mathcal{G}_{\overline{\mathbf{T}_i}}$. Thus, it does not affect those that satisfy the antecedent of Cond. (ii) of Def. 3. As a result, whenever, the antecedent of Cond. (ii) of Def. 3 holds, $t$ still belongs to the subscript of $W_i$. So Cond. (ii) of Def. 3 still holds.

Given that $P(\|\mathbf{W}_*\|)$ satisfies both conditions of Def. 3, it is in $\mathbf{P}^{\mathcal{L}_{2.25}}$. $\qquad\square$

**Lemma C.3.** *Given a causal diagram $\mathcal{G}$ over $\mathbf{V}$ and a set of counterfactual events $\mathbf{W}_*$, if $P(\mathbf{W}_*)$ is in $\mathbf{P}^{\mathcal{L}_{2.5}}$ of all SCMs compatible with $\mathcal{G}$, then $P(\|\mathbf{W}_*\|)$ is also in $\mathbf{P}^{\mathcal{L}_{2.5}}$ of all SCMs compatible with $\mathcal{G}$.*

*Proof.* The proof is very similar to Lemma C.2, with the key point being that the exclusion operator on $W_{i_{\mathbf{t}_i}}$ removes variables in $\mathbf{t}_i$ that does not have a directed edge to $W_i$ in $\mathcal{G}_{\overline{\mathbf{T}_i}}$. Thus, it does not affect those that satisfy the antecedent of Cond. (ii) of Def. 4. $\qquad\square$

**Lemma C.4.** *Given a causal diagram $\mathcal{G}$ over $\mathbf{V}$ and a set of counterfactual events $\mathbf{W}_* = \{W_{i_{[\mathbf{x}_i]}}\}$ with all subscripts taking consistent values from the same set $\mathbf{v} \in Val(\mathbf{V})$, if $\|W_{i_{[\mathbf{x}_i]}}\| = \|W_{i_{[\cup_i \mathbf{x}_i]}}\|$ for all $i$, then $P(\mathbf{W}_*)$ is in $\mathbf{P}^{\mathcal{L}_{2.25}}$ of all SCMs compatible with $\mathcal{G}$.*

*Proof.* The exclusion operator removes subscripts $x$ from $W_i$ if there is no directed path from $X$ to $W_i$ in $G_{\overline{\cup_i \mathbf{X}_i}}$. Thus, the subscripts that remain after exclusion capture precisely the cases in which the antecedent of Cond.(ii) in Definition 3 holds. If $\|W_{i_{[\mathbf{x}_i]}}\| = \|W_{i_{[\cup_i \mathbf{x}_i]}}\|$, the subscript in $\mathbf{x}_i$ accounts for all instances in $\cup_i * x_i$ that are restricted by Cond. (ii). Therefore, $P(\mathbf{W}_*)$ satisfies Def. 3 and belongs to $\mathbf{P}^{\mathcal{L}_{2.25}}$. $\square$

**Lemma C.5** (ctf-calculus — do-calculus reduction (Lemma 6 in [6]))**.** *ctf-calculus subsumes do-calculus.*

**Lemma C.6** (ctf-calculus 2.25 — do-calculus reduction)**.** *ctf-calculus restricted to $\mathbf{P}^{\mathcal{L}_{2.25}}$ subsumes do-calculus.*

*Proof.* This result follows from the proof of Lemma C.5 where all steps in the reduction only involves quantities within $\mathbf{P}^{\mathcal{L}_{2.25}}$. $\square$

Given a graphical model with bidirected edges, $\mathcal{G}$, the set $\mathbf{V}$ of observable variables represented as vertex can be partitioned into subsets called *c-components* [21] such that two variables belong to the same c-component if they are connected in $\mathcal{G}$ by a path made entirely of bidirected edges.

**Definition 29** (Ancestral components [6])**.** *Let $\mathbf{W}_*$ be a set of counterfactual variables, $\mathbf{X}_* \subseteq \mathbf{W}_*$, and $\mathcal{G}$ be a causal diagram. Then the ancestral components induced by $\mathbf{W}_*$, given $\mathbf{X}_*$, are sets $\mathbf{A}_{1*}, \mathbf{A}_{2*}, \ldots$ that form a partition over $An\mathbf{W}_*$, made of unions of ancestral sets $An[\mathcal{G}_{\underline{\mathbf{X}_*(W_{\mathbf{t}})}}]W_{\mathbf{t}}, W_{\mathbf{t}} \in \mathbf{W}_*$. Sets $An[\mathcal{G}_{\underline{\mathbf{X}_*(W_{1[\mathbf{t}_1]})}}]W_{1[\mathbf{t}_1]}$ and $An[\mathcal{G}_{\underline{\mathbf{X}_*(W_{2[\mathbf{t}_2]})}}]W_{2[\mathbf{t}_2]}$ are put together if they are not disjoint or there exists a bidirected arrow in $\mathcal{G}$ connecting variables in those sets.*

**Lemma C.7** (Ancestral Set Factorization (Lemma 3 in [6]))**.** *Let $\mathbf{W}_*$ be an ancestral set, that is, $An(\mathbf{W}_*) = \mathbf{W}_*$, and let $\mathbf{w}_*$ be a vector with a value for each variable in $\mathbf{W}_*$. Then,*

$$P(\mathbf{W}_* = \mathbf{w}_*) = P(\bigwedge_{W_{\mathbf{t}} \in \mathbf{W}_*} W_{\mathbf{pa}_w} = w) \tag{147}$$

*where each $w$ is taken from $\mathbf{w}_*$ and $\mathbf{pa}_w$ is determined for each $W_{\mathbf{t}} \in \mathbf{W}_*$ as follows:*

*(i) the values for variables in $\mathbf{Pa}_w \cap \mathbf{T}$ are the same as in $t$, and*

*(ii) the values for variables in $\mathbf{Pa}_w \backslash \mathbf{T}$ are taken from $\mathbf{w}_*$ corresponding to the parents of $W_{\mathbf{t}}$.*

**Lemma C.8** (C-component Factorization (Lemma 4 in [6]))**.** *Let $P(\mathbf{W}_* = \mathbf{w}_*)$ be a distribution such that each variable in $\mathbf{W}_*$ has the form $W_{\mathbf{pa}_w}$, let $W_1 < W_2 < \cdots$ be a topological order over the variables in $\mathcal{G}[\mathbf{V}(\mathbf{W}_*)]$, and let $\mathbf{C}_1, \ldots, \mathbf{C}_k$ be the c-components of the same graph. Define $\mathbf{C}_{j_*} = \{W_{\mathbf{pa}_w} \in \mathbf{W}_* \mid W \in \mathbf{C}_j\}$ and $\mathbf{c}_{j_*}$ as the values in $\mathbf{w}_*$ corresponding to $\mathbf{C}_{j_*}$, then $P(\mathbf{W}_* = \mathbf{w}_*)$ decomposes as*

$$P(\mathbf{W}_* = \mathbf{w}_*) = \prod_j P(\mathbf{C}_{j_*} = \mathbf{c}_{j_*}) \tag{148}$$

**Lemma C.9** (Ancestral Set in $\mathcal{L}_{2.25}/\mathcal{L}_{2.5}$)**.** *$P(\mathbf{W}_*)$ is in $\mathcal{L}_{2.25}/\mathcal{L}_{2.5}$ if and only if the distribution over its ancestral set $P(An(\mathbf{W}_*))$ is also in $\mathcal{L}_{2.25}/\mathcal{L}_{2.5}$.*

*Proof.* For $\mathcal{L}_{2.25}$, $CRS(An(\mathbf{W}_*)) = CRS(\mathbf{W}_*)$ by Def. 7; and for $\mathcal{L}_{2.5}$, $An(An(\mathbf{W}_*)) = An(\mathbf{W}_*)$ by Def. **??**. Thus, $\mathbf{W}_*$ satisfies Lemma 1 if and only if $An(\mathbf{W}_*)$ satisfies Lemma 1. $\square$

**Lemma C.10** (Ancestral Set Factor in $\mathcal{L}_{2.25}/\mathcal{L}_{2.5}$)**.** *Let $\mathbf{W}_*$ be an ancestral set, that is, $An(\mathbf{W}_*) = \mathbf{W}_*$, and let $\mathbf{w}_*$ be a vector with a value for each variable in $\mathbf{W}_*$. Then, $P(\mathbf{W}_*)$ is in $\mathcal{L}_{2.25}/\mathcal{L}_{2.5}$ only if its ancestral set factor $P(\bigwedge_{W_{\mathbf{t}} \in \mathbf{W}_*} W_{\mathbf{pa}_w} = w)$ is in $\mathcal{L}_{2.25}/\mathcal{L}_{2.5}$.*

*Proof.* If $P(\mathbf{W}_*)$ is in $\mathcal{L}_{2.25}/\mathcal{L}_{2.5}$, then there does not exist two variables $W_{\mathbf{t}}$ and $W_{\mathbf{s}}$ in $\mathbf{W}_*$ with inconsistent subscripts. Therefore, the ancestral set factorization will also have distinct $W$ for each $W_{\mathbf{pa}_w}$. It satisfies conditions in Def. 3/Def. 4 with consistent values from $\mathbf{w}_*$ for $\mathcal{L}_{2.25}$ and with $\mathbf{Pa}_w$ blocking all directed path from other variables to $W$. $\square$

**Algorithm 2** CTFIDU($\mathbf{Y}_*, \mathbf{y}_*, \mathbb{Z}, \mathcal{G}$)

---

**Input:** $\mathcal{G}$ causal diagram over variables $\mathbf{V}$; $\mathbf{Y}_*$ a set of counterfactual variables in $\mathbf{V}$; $\mathbf{y}_*$ a set of values for $\mathbf{Y}_*$; and available distribution specification $\mathbb{Z}$.

**Output:** $P(\mathbf{Y}_* = \mathbf{y}_*)$ in terms of available distributions or FAIL if not identifiable from $\langle \mathcal{G}, \mathbb{Z} \rangle$

1: let $\mathbf{Y}_* \leftarrow \|\mathbf{Y}_*\|$.
2: **if** there exists $Y_{\mathbf{x}} \in \mathbf{Y}_*$ with two or more different values in $\mathbf{y}_*(Y_{\mathbf{x}})$ or $Y_y \in \mathbf{Y}_*$ with $\mathbf{y}_*(Y_y) \neq y$ **then return** $0$.
3: **end if**
4: **if** there exists $Y_{\mathbf{x}} \in \mathbf{Y}_*$ with two consistent values in $\mathbf{y}_*(Y_{\mathbf{x}})$ or $Y_y \in \mathbf{Y}_*$ with $\mathbf{y}_*(Y_y) = y$ **then** remove repeated variables from $\mathbf{Y}_*$ and values $\mathbf{y}_*$.
5: **end if**
6: let $\mathbf{W}_* \leftarrow An(\mathbf{Y}_*)$, and let $\mathbf{C}_{1*}, \ldots, \mathbf{C}_{k*}$ be corresponding ctf-factors in $\mathcal{G}[\mathbf{V}(\mathbf{W}_*)]$.
7: **for each** $\mathbf{C}_i$ s.t. $(\mathbf{C}_{i*} = \mathbf{c}_{i*})$ is not inconsistent, $\mathbf{Z} \in \mathbb{Z}$ s.t. $\mathbf{C}_i \cap \mathbf{Z} = \emptyset$ **do**
8:     let $\mathbf{B}_i$ be the c-component of $\mathcal{G}_{\overline{\mathbf{Z}}}$ such that $\mathbf{C}_i \subseteq \mathbf{B}_i$, compute $P_{\mathbf{V} \setminus \mathbf{B}_i}(\mathbf{B}_i)$ from $P_{\mathbf{Z}}(\mathbf{V})$.
9:     **if** IDENTIFY($\mathbf{C}_i, \mathbf{B}_i, P_{\mathbf{V} \setminus \mathbf{B}_i}(\mathbf{B}_i), \mathcal{G}$) does not FAIL **then**
10:         let $P_{\mathbf{V} \setminus \mathbf{C}_i}(\mathbf{C}_i) \leftarrow$ IDENTIFY($\mathbf{C}_i, \mathbf{B}_i, P_{\mathbf{V} \setminus \mathbf{B}_i}(\mathbf{B}_i), \mathcal{G}$).
11:         let $P(\mathbf{C}_{i*} = \mathbf{c}_{i*}) \leftarrow P_{\mathbf{V} \setminus \mathbf{C}_i}(\mathbf{C}_i)$ evaluated with values $(\mathbf{c}_{i*} \cup \bigcup_{C_{\mathbf{t}} \in \mathbf{C}_{i*}} \mathbf{pa}_c)$.
12:         move to the next $\mathbf{C}_i$.
13:     **end if**
14: **end for**
15: **if** any $P(\mathbf{C}_{i*} = \mathbf{c}_{i*})$ is inconsistent or was not identified from $\mathbb{Z}$ **then return** FAIL.
16: **end if**
17: **return** $P(\mathbf{Y}_* = \mathbf{y}_*) \leftarrow \sum_{\mathbf{w}_* \setminus \mathbf{y}_*} \prod_i P(\mathbf{C}_{i*} = \mathbf{c}_{i*})$.

---

**Lemma C.11** (C-component Factor in $\mathcal{L}_{2.25}/\mathcal{L}_{2.5}$). *Let $P(\mathbf{W}_* = \mathbf{w}_*)$ be a distribution such that each variable in $\mathbf{W}_*$ has the form $W_{\mathbf{pa}_w}$, with its c-component factorization $P(\mathbf{W}_* = \mathbf{w}_*) = \prod_j P(\mathbf{C}_{j_*} = \mathbf{c}_{j_*})$. Then, $P(\mathbf{W}_*)$ is in $\mathcal{L}_{2.25}/\mathcal{L}_{2.5}$ only if its c-component factors $P(\mathbf{C}_{j_*} = \mathbf{c}_{j_*})$ are in $\mathcal{L}_{2.25}/\mathcal{L}_{2.5}$.*

*Proof.* If $P(\mathbf{W}_*)$ is in $\mathcal{L}_{2.25}/\mathcal{L}_{2.5}$, then it has distinct $W$ for each counterfactual in the set and satisfies Def. 3/Def. 4. This property is not affected by c-component factorization as it only partitions $\mathbf{W}_*$ into subsets connected by bidrected paths. As a result, each $P(\mathbf{C}_{j_*} = \mathbf{c}_{j_*})$ will also satisfy Def. 3/Def. 4. $\square$

**Lemma C.12** (Consistency (Lemma 1 in [6])). *Given SCM $\mathcal{M}$ and $X, Y \in \mathbf{V}$, $\mathbf{T}, \mathbf{R} \subseteq \mathbf{V}$, and let $x$ be a value in the domain of $X$. Then,*

$$P(Y_{\mathbf{T}_*}, X_{\mathbf{T}_*} = x) = P(Y_{\mathbf{T}_* x}, X_{\mathbf{T}_*} = x), \tag{149}$$

*where $\mathbf{T}_*$ represent any combination of counterfactuals based on $\mathbf{T}$.*

**Lemma C.13** (Exclusion operator (Lemma 2 in [6])). *Let $Y_{\mathbf{x}}$ be a counterfactual variable, $\mathcal{G}$ a causal diagram, and*

$$Y_{\mathbf{z}} \text{ such that } \mathbf{Z} = \mathbf{X} \cap An_{\mathcal{G}_{\overline{\mathbf{X}}}}(Y) \text{ and } \mathbf{z} = \mathbf{x} \cap \mathbf{Z}. \tag{150}$$

*Then, $Y_{\mathbf{z}} = Y_{\mathbf{x}}$ holds for any model compatible with $\mathcal{G}$. Moreover, this transformation is denoted as $\|(Y_{\mathbf{x}})\| := Y_{\mathbf{z}}$.*

**Lemma C.14** (Independence in $\mathcal{L}_{2.25}/\mathcal{L}_{2.5}$). *Given a CBN2.25/CBN2.5, Theorem 6 is sound when the AMWN is constructed over $\mathbf{W}_*$ where $P(\mathbf{W}_*)$ is in $\mathcal{L}_{2.25}/\mathcal{L}_{2.5}$.*

*Proof.* The soundness follows from soundness of Theorem 6, where the ancestral set factorization constructed over $\{\mathbf{X}_{\mathbf{t}}, \mathbf{Y}_{\mathbf{r}}, \mathbf{Z}\}$ in the proof is also in the corresponding layers $\mathcal{L}_{2.25}/\mathcal{L}_{2.5}$ by Lemma C.9 and Lemma C.10. $\square$

**Algorithm 3** CTFID($\mathbf{Y}_*, \mathbf{y}_*, \mathbf{X}_*, \mathbf{x}_*, \mathbb{Z}, \mathcal{G}$)

---

**Input:** $\mathcal{G}$ causal diagram over variables $\mathbf{V}$; $\mathbf{Y}_*, \mathbf{X}_*$ a set of counterfactual variables in $\mathbf{V}$; $\mathbf{y}_*, \mathbf{x}_*$ a set of values for $\mathbf{Y}_*$ and $\mathbf{X}_*$; and available distribution specification $\mathbb{Z}$.

**Output:** $P(\mathbf{Y}_* = \mathbf{y}_* \mid \mathbf{X}_* = \mathbf{x}_*)$ in terms of available distributions or FAIL if non-ID from $\langle \mathcal{G}, \mathbb{Z} \rangle$.

1: Let $\mathbf{A}_{1*}, \mathbf{A}_{2*}, \ldots$ be the ancestral components of $\mathbf{Y}_* \cup \mathbf{X}_*$ given $\mathbf{X}_*$.
2: Let $\mathbf{D}_*$ be the union of the ancestral components containing a variable in $\mathbf{Y}_*$ and $\mathbf{d}_*$ the corresponding set of values.
3: let $Q \leftarrow$ CTFIDU($\bigcup_{D_\mathbf{t} \in \mathbf{D}_*} \mathbf{D}_{\mathbf{pa}_d}, \mathbf{d}_*, \mathbb{Z}, \mathcal{G}$).
4: **return** $\sum_{\mathbf{d}_* \setminus (\mathbf{y}_* \cup \mathbf{x}_*)} Q / \sum_{\mathbf{d}_* \setminus \mathbf{x}_*} Q$.

---

## C.2 Proofs for Main Theorems

**Theorem 1** ($\mathcal{L}_{2.25}$-Connection — SCM-CBN2.25)**.** *The Causal diagram $\mathcal{G}$ induced by the SCM $\mathcal{M}$ following the constructive procedure in Def. 13 is a CBN2.25 for $\mathbf{P}^{\mathcal{L}_{2.25}}$, the collection of all $\mathcal{L}_{2.25}$ distributions induced by $\mathcal{M}$.*

*Proof.* Let $\mathcal{M}$ be an SCM, $\mathbf{P}^{\mathcal{L}_{2.25}}$ the $\mathcal{L}_{2.25}$ distributions it induces and $\mathcal{G}$ its causal diagram. We prove that $\langle \mathcal{G}, \mathbf{P}^{\mathcal{L}_{2.25}} \rangle$ is a CBN2.25, by showing that the 3 conditions defined in Def. **??** holds in $\mathbf{P}^{\mathcal{L}_{2.25}}$ according to $\mathcal{G}$.

(Independence Restrictions) Given a potential response of the form $W_{\mathbf{pa}_w}$, its value only depends on the exogenous variables $\mathbf{U}_w$ which appear as arguments in $f_W$. Let $\mathbf{W}_*$ be the set of counterfactuals of the form $W_{\mathbf{pa}_w}$ with $\mathbf{pa}_w$ taking consistent values from $\mathbf{v} \in Val(\mathbf{V})$, $P(\mathbf{W}_*)$ falls in $\mathcal{L}_{2.25}$ as it satisfy conditions of Def. 3. Let $\mathbf{C}_1, ..., \mathbf{C}_l$ be the c-components of $G[\mathbf{V}(\mathbf{W}_*)]$, and $\mathbf{C}_{1*}, ..., \mathbf{C}_{l*}$ the corresponding partition over $\mathbf{W}_*$. Then the set of exogenous variables $\mathbf{U}(\mathbf{W}_*)$ can be partitioned as $\mathbf{U}(\mathbf{C}_{1*}), ..., \mathbf{U}(\mathbf{C}_{l*})$ where $\mathbf{U}(\mathbf{C}_{i*})$ and $\mathbf{U}(\mathbf{C}_{j*})$ are disjoint for all $i, j = 1, ..., l, i \neq j$, due to the absence of bidirected paths between variables in $\mathbf{C}_i$ and and variables $\mathbf{C}_j$. Then by Def. 3,

(Exclusion restrictions) Given a potential response of the form $Y_{\mathbf{pa}_y, \mathbf{z}}$, its value only depends on the exogenous variables $\mathbf{U}_y$ which appear as arguments in $f_Y$ as $\mathbf{pa}_y$ are fixed. Thus, $Y_{\mathbf{pa}_y, \mathbf{z}}(\mathbf{u}) = Y_{\mathbf{pa}_y}(\mathbf{u})$. Then by Def. 3, for any counterfactual set $\mathbf{W}_*$ such that $P(Y_{\mathbf{pa}_y, \mathbf{z}} = y, \mathbf{W}_* = \mathbf{w}_*) \in \mathbf{P}^{\mathcal{L}_{2.25}}$,

$$P(Y_{\mathbf{pa}_y, \mathbf{z}} = y, \mathbf{W}_* = \mathbf{w}_*) = \sum_{\mathbf{u}} \mathbf{1}(Y_{\mathbf{pa}_y, \mathbf{z}}(\mathbf{u}) = y, \mathbf{W}_*(\mathbf{u}) = \mathbf{w}_*) P(\mathbf{u}) \tag{151}$$

$$= \sum_{\mathbf{u}} \mathbf{1}(Y_{\mathbf{pa}_y}(\mathbf{u}) = y, \mathbf{W}_*(\mathbf{u}) = \mathbf{w}_*) P(\mathbf{u}) \tag{152}$$

$$= P(Y_{\mathbf{pa}_y} = y, \mathbf{W}_* = \mathbf{w}_*) \tag{153}$$

which proves the exclusion restrictions are satisfied.

(Consistency restrictions) Given $\mathbf{u} \in Val(\mathbf{U})$ such that $Y_\mathbf{z}(\mathbf{u}) = y, \mathbf{X}_\mathbf{z}(\mathbf{u}) = \mathbf{x}, \mathbf{W}_*(\mathbf{u}) = \mathbf{w}_*$, for some $Y \in \mathbf{V}, \mathbf{X} \subseteq \mathbf{Pa}_y, \mathbf{Z} \subseteq \mathbf{V} \setminus (\mathbf{X} \cup \{Y\}), \mathbf{R} = \mathbf{Pa}_y \setminus (\mathbf{X} \cup \mathbf{Z})$, we have

$$Y_\mathbf{z}(\mathbf{u}) = f_Y(\mathbf{z} \cap \mathbf{pa}_y, \mathbf{X}_\mathbf{z}(\mathbf{u}), \mathbf{R}_\mathbf{z}(\mathbf{u}), \mathbf{u}(\mathbf{U}_y)) \tag{154}$$

$$= f_Y(\mathbf{z} \cap \mathbf{pa}_y, \mathbf{x}, \mathbf{R}_\mathbf{z}(\mathbf{u}), \mathbf{u}(\mathbf{U}_y)) \tag{155}$$

$$= Y_{\mathbf{zx}}(\mathbf{u}) \tag{156}$$

Then by Def. 3, for any counterfactual set $\mathbf{W}_*$ such that $P(Y_\mathbf{z} = y, \mathbf{X}_\mathbf{z} = \mathbf{x}, \mathbf{W}_* = \mathbf{w}_*) \in \mathbf{P}^{\mathcal{L}_{2.25}}$,

$$P(Y_\mathbf{z} = y, \mathbf{X}_\mathbf{z} = \mathbf{x}, \mathbf{W}_* = \mathbf{w}_*) = \sum_{\mathbf{u}} \mathbf{1}(Y_\mathbf{z}(\mathbf{u}) = y, \mathbf{X}_\mathbf{z}(\mathbf{u}) = \mathbf{x}, \mathbf{W}_*(\mathbf{u}) = \mathbf{w}_*) P(\mathbf{u}) \tag{157}$$

$$= \sum_{\mathbf{u}} \mathbf{1}(Y_{\mathbf{xz}}(\mathbf{u}) = y, \mathbf{X}_\mathbf{z}(\mathbf{u}) = \mathbf{x}, \mathbf{W}_*(\mathbf{u}) = \mathbf{w}_*) P(\mathbf{u}) \tag{158}$$

$$= P(Y_{\mathbf{xz}} = y, \mathbf{X}_\mathbf{z} = \mathbf{x}, \mathbf{W}_* = \mathbf{w}_*) \tag{159}$$

which proves the consistency restrictions are satisfied. $\qquad\square$

**Definition 30** (Counterfactual Reachability Set). *Given a graph $\mathcal{G}$ and a potential outcome $Y_{\mathbf{x}}$, the counterfactual reachability set of $Y_{\mathbf{x}}$, denoted $CRS(Y_{\mathbf{x}})$, consists of each $\|W_{\mathbf{x}}\|$ s.t. $W \in (An(Y) \cup \{De(V) : \forall V \in \mathbf{X}\})\backslash\mathbf{X}$ and $\|W_{\mathbf{x}\backslash w}\|$ s.t. $W \in (An(Y) \cup \{De(V) : \forall V \in \mathbf{X}\}) \cap \mathbf{X}$. For a set $\mathbf{W}_*$, $CRS(\mathbf{W}_*)$ is defined to be the union of the CRS of each potential outcome in the set, such that for any set of variables $\{W_{i_{[\mathbf{x}_i]}}\}_i \subseteq \mathbf{W}_*$ with their CRS set having counterfactual variables $\{R_{[\mathbf{x}_i]}\}_i$ over the same variable $R$, $\{R_{[\mathbf{x}_i]}\}_i$ is merged into one variable $\|R_{[\cup_i \mathbf{x}_i]}\|$ if $\|W_{i_{[\cup_i \mathbf{x}_i]}}\| = W_{i_{[\mathbf{x}_i]}}$ for all $i$.*

**Lemma 1.** *A distribution $Q = P(\mathbf{W}_*)$ is in the $\mathcal{L}_{2.25}/\mathcal{L}_{2.5}$ distributions induced by any SCM compatible with a given graph $\mathcal{G}$ if and only if the set $CRS(\mathbf{W}_*)$ satisfies (i) and (ii) / $An(\mathbf{W}_*)$ satisfies (i): (i) Does not contain any pair of potential outcomes $W_{\mathbf{s}}, W_{\mathbf{t}}$ of the same variable $W$ under different regimes where $\mathbf{s} \neq \mathbf{t}$; (ii) $\mathbf{W}_*$ does not contain any pair of potential outcomes $R_{\mathbf{s}}, W_{\mathbf{t}}$ with inconsistent subscripts where $\mathbf{s} \cap \mathbf{T} \neq \mathbf{t} \cap \mathbf{S}$.*

*Proof.* Consistent values across the variables are enforced by (ii). Each CRS set corresponding to a potential outcome $Y_*$ includes all variables that must remain consistent with $Y_*$ under the regime $*$. When taking the union of CRS sets over multiple potential outcomes, and if the union does not contain any pair of potential outcomes $W_{\mathbf{s}}, W_{\mathbf{t}}$ for the same variable $W$ under different regimes, then two cases arise:

(a) All CRS sets are disjoint with respect to the variables from which their potential outcomes are derived. This implies that the ancestral and descendant sets of these variables are also disjoint, so there is no directed path crossing the CRS sets in a way that would trigger the antecedent of Cond. (ii) in Definition 3.

(b) Any overlapping CRS sets must involve counterfactuals over the same variable, which are merged as $|W_{i_{[\cup_i * x_i]}}| = |W_{i_{[* x_i]}}|$ for all $i$. This condition implies that the variables underlying these merged CRS sets are consistent, by Lemma C.4.

Therefore, $P(\mathbf{W}_*)$ satisfies conditions in Def. 3 and belongs to $\mathbf{P}^{\mathcal{L}_{2.25}}$.

The graphical check for $\mathcal{L}_{2.5}$ is proved in Corollary 3.7 of [19]. $\qquad\square$

**Theorem 2** (Soundness and Completeness for CBN2.25/CBN2.5 Identifiability). *An $\mathcal{L}_{2.25}/\mathcal{L}_{2.5}$ quantity $Q$ is identifiable from a given set of observational and interventional distributions and a CBN2.25/CBN2.5 if and only if there exists a sequence of applications of the rules of ctf-calculus for CBN2.25/CBN2.5 and the probability axioms restrained within $\mathcal{L}_{2.25}/\mathcal{L}_{2.5}$ that reduces $Q$ into a function of the available distributions.*

*Proof.* The soundness of the calculus for $\mathcal{L}_{2.25}/\mathcal{L}_{2.5}$ follows from the soundness of the ctf-calculus rules. The soundness of the ctf-calculus rules in turn follows from Lemma C.12 for Rule 1, Lemma C.14 for Rule 2 and Lemma C.13 for Rule 3.

To prove that it is complete, we rely on the completeness of the CTFID algorithm reproduced as Algo. 3 and Algo. 2 [7]. Specifically, we show that if the query is in $\mathcal{L}_{2.25}/\mathcal{L}_{2.5}$, all steps of the CTFID algorithm can be justified by the rules of ctf-calculus for CBN2.25/CBN2.5 and the probability axioms restrained within $\mathcal{L}_{2.25}/\mathcal{L}_{2.5}$.

Line 1 and 2 of Algo. 3 are justfied by Lemma C.9 and Lemma C.10: if the input query $P(\mathbf{Y}_* = \mathbf{y}_* | \mathbf{X}_* = \mathbf{x}_*)$ is in $\mathcal{L}_{2.25}/\mathcal{L}_{2.5}$, then the ancestral set factorization $P(\bigcup_{D_t \in \mathbf{D}_*} D_{\mathbf{pa}_d} = d)$ over $\mathbf{D}_* = An(\mathbf{Y}_*, \mathbf{X}_*)$ and $\mathbf{d}_* \in Val(\mathbf{D}_*)$ consistent with $\mathbf{y}_*, \mathbf{x}_*$ is also in $\mathcal{L}_{2.25}/\mathcal{L}_{2.5}$. Thus the probability axioms underlying the marginalization step have all quantities within the corresponding layers.

Line 1 of Algo. 2 is justified by rule 3 of the ctf-calculus and Lemma C.2 and Lemma C.3 where both $\mathbf{D}_*$ and $\|\mathbf{D}_*\|$ are in the corresponding layers. Line 2 to 3 are justified by quantities in $\mathcal{L}_{2.25}/\mathcal{L}_{2.5}$ having consistent values. Line 4 to 5 follow from probability axiom to remove redundant variables. From line 6 to 14, the algorithm identifies the factors based on c-componentes using IDENTIFY [21] which soundness can be justified with do-calculus [9], which in turn is subsumed by ctf-calculus 2.25 by Lemma C.6. At line 17, the algorithm returns the result as a product that is justified by Lemma C.11.

Therefore, given a query in $\mathcal{L}_{2.25}/\mathcal{L}_{2.5}$, CTFID is both sound and complete to determine if it is identifiable from the available data without any intermediate step having quantities outside the layer. □

**Theorem 3** (PCH*). *Given an SCM $\mathcal{M}$ and its induced collections of observational ($\mathbf{P}^{\mathcal{L}_1}$), interventional($\mathbf{P}^{\mathcal{L}_2}$), $\mathcal{L}_{2.25}$ ($\mathbf{P}^{\mathcal{L}_{2.25}}$), $\mathcal{L}_{2.5}$ ($\mathbf{P}^{\mathcal{L}_{2.5}}$), and counterfactual ($\mathbf{P}^{\mathcal{L}_3}$) distributions: $\mathbf{P}^{\mathcal{L}_1} \subseteq \mathbf{P}^{\mathcal{L}_2} \subseteq \mathbf{P}^{\mathcal{L}_{2.25}} \subseteq \mathbf{P}^{\mathcal{L}_{2.5}} \subseteq \mathbf{P}^{\mathcal{L}_3}$.*

*Proof.* With PCH already established and proved for $\mathcal{L}_1, \mathcal{L}_2$ and $\mathcal{L}_3$ [3], we prove that (1) $\mathbf{P}^{\mathcal{L}_2} \subseteq \mathbf{P}^{\mathcal{L}_{2.25}}$, (2) $\mathbf{P}^{\mathcal{L}_{2.25}} \subseteq \mathbf{P}^{\mathcal{L}_{2.5}}$ and (3) $\mathbf{P}^{\mathcal{L}_{2.5}} \subseteq \mathbf{P}^{\mathcal{L}_3}$.

It is easy to show that $\mathbf{P}^{\mathcal{L}_2} \subseteq \mathbf{P}^{\mathcal{L}_{2.25}}$, because each distribution in $\mathbf{P}^{\mathcal{L}_2}$ can be derived from a marginalization of a distribution in $\mathbf{P}^{\mathcal{L}_{2.25}}$:

$$P(\mathbf{Y} = \mathbf{y} | do(\mathbf{X} = \mathbf{x})) = \sum_{X \in \mathbf{Y} \cap \mathbf{X}} P(\bigwedge_{V_i \in \mathbf{Y} \setminus \mathbf{X}} V_{i_{[\mathbf{x}]}} = v_i, \bigwedge_{V_i \in \mathbf{Y} \cap \mathbf{X}, v_i = V_i \cap \mathbf{x}} V_{i_{[\mathbf{x} \setminus v_i]}} = v_i) \quad (160)$$

where the subscripts for all variables take the whole set $\mathbf{x}$. Clearly, it is in $\mathbf{P}^{\mathcal{L}_{2.25}}$ as the consistent subscripts satisfy conditions of Def. 3.

It is also easy to see that $\mathbf{P}^{\mathcal{L}_{2.5}} \subseteq \mathbf{P}^{\mathcal{L}_3}$ because $\mathbf{P}^{\mathcal{L}_3}$ contains all possible joint distributions over all counterfactual variables, whereas $\mathbf{P}^{\mathcal{L}_{2.5}}$ imposes additional constraints over the joint of counterfactual variables.

To prove that $\mathbf{P}^{\mathcal{L}_{2.25}} \subseteq \mathbf{P}^{\mathcal{L}_{2.5}}$, we show that if a distribution satisfies Def. 3, it also satisfies Def. 4. First, note that the key difference between Def. 3 and Def. 4 lies in the two conditions. Thus, we only need to prove that a distribution of the form $P(\bigwedge_{V_i \in \mathbf{Y} \setminus \mathbf{X}} V_{i_{[\mathbf{x}]}} = v_i, \bigwedge_{V_i \in \mathbf{Y} \cap \mathbf{X}, v_i = V_i \cap \mathbf{x}} V_{i_{[\mathbf{x}_i \setminus v_i]}} = v_i)$ satisfying the two conditions in Def. 3 must also satisfy the two conditions in Def. 4.

For Cond. (i), both languages require the subscripts to cover the whole space of $\mathbf{X}$. However, Def. 3 is stronger by restricting the value assignments to the set $\mathbf{x}$, while Def. 4 allows $\mathbf{x}_i$ to take different values from $Val(\mathbf{X}_i)$. Thus, if Cond. (i) of Def. 3 holds, Cond. (i) of Def. 4 immediately holds.

For Cond. (ii) and by Cor. 2, the antecedent in Def. 4 checks if there is a directed path from $B \in \mathbf{X}$ to $V_i \in Ch(B)$ to $V_j$ in $G_{\overline{\mathbf{X}_j}}$. If such a path exists, we denote it by $p$. There are two possibilities: (a) $p$ is in $G_{\overline{\mathbf{X} \setminus V_j}}$; (b) $p$ is not in $G_{\overline{\mathbf{X} \setminus V_j}}$. For (a), Cond. (ii) of Def. 3 will enforce $b$ to appear in the subscript of both $V_i$ and $V_j$. For (b), it implies that there exists a variable $X \in \mathbf{X} \setminus \mathbf{X}_j$ that lies on $p$ between $V_i$ and $V_j$. We focus on the subpath $p'$ of $p$ directed from $X$ to $V_j$. If $X$ is in $An(V_j)$ in $G_{\overline{\mathbf{X}_j}}$, then $X$ must be in $\mathbf{X}_j$ by Cond. (ii) of Def. 3 which leads to a contradiction. If $X$ is not in $An(V_j)$ in $G_{\overline{\mathbf{X}_j}}$, then there exists another $X' \in \mathbf{X} \setminus \mathbf{X}_j$ that lies on $p'$ between $X$ and $V_j$. We can apply the same logic to shorten $p$ until there is no more variable in $\mathbf{X} \setminus \mathbf{X}_j$ that fulfills the same condition. When this terminal condition is hit, the final subpath enforces the variable in $\mathbf{X} \setminus \mathbf{X}_j$ on the path to be in the subscript of $V_j$. The same contradiction is achieved. As a result, there cannot be any variable $X \in \mathbf{X} \setminus \mathbf{X}_j$ that lies on $p$ between $V_i$ and $V_j$. Therefore, whenever the antecedent of Cond. (ii) of Def. 4 is triggered, Cond. (ii) of Def. 3 also holds to enforce consistent subscripts between $V_i$ and $V_j$.

This proves that all distributions in $\mathbf{P}^{\mathcal{L}_{2.25}}$ are also in $\mathbf{P}^{\mathcal{L}_{2.5}}$, or equivalently $\mathbf{P}^{\mathcal{L}_{2.25}} \subseteq \mathbf{P}^{\mathcal{L}_{2.5}}$.

□

**Theorem 4** (Hierarchy of Graphical Models, PCH*). *Given a causal diagram $\mathcal{G}$, the set of constraints it encodes when it is interpreted as a graphical model on layer $i$ is a subset of the constraints it encodes when it is interpreted as a graphical model on layer $j$, when $i \leq j$.*

*Proof.* The constraints encoded by a BN are included as Cond. (i) of the corresponding CBN, making the containment relationship is straightforward. The hierarchical relationship among the constraints encoded by CBN2.25, CBN2.5, and CTFBN is also straightforward, as they share the same structural form while progressively increasing the flexibility of distributions allowed at each level in the model hierarchy. The containment relationship between CBN and CBN2.25 follows from the fact that do-calculus is subsumed by the ctf-calculus 2.25 (Lemma C.6), and that the constraints defined in CBN imply all rules of do-calculus, while those in CBN2.25 imply all rules of ctf-calculus 2.25.

| Graphical Model | Meaning of Missing Directed Edge | Meaning of Missing Bidirected Edge |
|---|---|---|
| $\mathcal{L}_1$: BN | $P(v_i\|\mathbf{pa}_i,\mathbf{nd}_i) = P(v_i\|\mathbf{pa}_i)$ | |
| $\mathcal{L}_2$: CBN | $P(v_{i_{\mathbf{pa}_i,\mathbf{z}}}) = P(v_{i_{\mathbf{pa}_i}})$ | $P(v_i\|do(\mathbf{x}),\mathbf{pa}_i^c,do(\mathbf{pa}_i^u))$ 
 $= P(v_i\|do(\mathbf{x}),\mathbf{pa}_i^c,\mathbf{pa}_i^u)$ |
| $\mathcal{L}_{2.25}$: CBN2.25 | $P(v_{i_{\mathbf{pa}_i,\mathbf{z}}},\mathbf{w}_*) = P(v_{i_{\mathbf{pa}_i}},\mathbf{w}_*),$ 
 with $P(v_{i_{\mathbf{pa}_i,\mathbf{z}}},\mathbf{w}_*) \in \mathbf{P}^{\mathcal{L}_{2.25}}$ | $P(v_{i_{\mathbf{pa}_i}},v_{j_{\mathbf{pa}_j}}) = P(v_{i_{\mathbf{pa}_i}})P(v_{j_{\mathbf{pa}_j}}),$ 
 with $V_i \neq V_j$ 
 and $\mathbf{pa}_i$ and $\mathbf{pa}_j$ taking consistent values |
| $\mathcal{L}_{2.5}$: CBN2.5 | $P(v_{i_{\mathbf{pa}_i,\mathbf{z}}},\mathbf{w}_*) = P(v_{i_{\mathbf{pa}_i}},\mathbf{w}_*),$ 
 with $P(v_{i_{\mathbf{pa}_i,\mathbf{z}}},\mathbf{w}_*) \in \mathbf{P}^{\mathcal{L}_{2.5}}$ | $P(v_{i_{\mathbf{pa}_i}},v_{j_{\mathbf{pa}_j}}) = P(v_{i_{\mathbf{pa}_i}})P(v_{j_{\mathbf{pa}_j}}),$ 
 with $V_i \neq V_j$ |
| $\mathcal{L}_3$: CTFBN | $P(v_{i_{\mathbf{pa}_i,\mathbf{z}}},\mathbf{w}_*) = P(v_{i_{\mathbf{pa}_i}},\mathbf{w}_*),$ 
 for any $\mathbf{w}_*$ | $P(v_{i_{\mathbf{pa}_i}},v_{j_{\mathbf{pa}_j}}) = P(v_{i_{\mathbf{pa}_i}})P(v_{j_{\mathbf{pa}_j}})$ 
 with $\mathbf{pa}_i \neq \mathbf{pa}_j$ if $V_i = V_j$ |

Table 4: Summary of how missing edges are interpreted in graphical models at different layers

Since the constraints encoded by graphical models are encoded by the missing edges in $\mathcal{G}$, we can alternatively establish the hierarchy by comparing how different models interpret these missing edges, as summarized in Table 4. For missing directed edges, the constraint forms are consistent across layers, but higher layers allow increasing flexibility in the sets $\mathbf{w}_*$ that can be jointly conditioned on. Similarly, for missing bidirected edges, the independence constraints in CBN2.25s, CBN2.5s, and CTFBNs share a common structure, with each successive model relaxing the limitations on how these independencies are expressed:

- Independence constraints in CBN2.25s only apply to distributions over distinct variables that share consistent parent values.

- Independence constraints in CBN2.5s extend to distributions over distinct variables, allowing their parents' values to vary freely.

- Independence constraints in CTFBNs apply to distributions over any variables, including those of the form $P(W_{\mathbf{pa}_w}, W_{\mathbf{pa}'_w})$ as long as $\mathbf{pa}_w \neq \mathbf{pa}'_w$.

$\square$

**Theorem 5** ($\mathcal{L}_{2.5}$-Connection — CBN2.5 (Markovian and Semi-Markovian)). *The Causal diagram $\mathcal{G}$ induced by the SCM $\mathcal{M}$ following the constructive procedure in Def. 13 is a CBN2.5 for $\mathbf{P}^{\mathcal{L}_{2.5}}$, the collection of all $\mathcal{L}_{2.5}$ distributions induced by $\mathcal{M}$.*

*Proof.* The proof is similar to the proof for Theorem 1 with the independence restrictions expanded to allow inconsistent parent values, and the exclusion and consistency restrictions expanded to join more $\mathbf{W}_*$ such that the distributions are within $\mathcal{L}_{2.5}$ instead of $\mathcal{L}_{2.25}$. $\square$

## D  Frequently Asked Questions

Q1. Where is the causal diagram coming from? Is it reasonable to expect the data scientist to create one?

**Answer.** First, the assumption of the causal diagram is made out of necessity. The causal diagram is a well-known flexible data structure that is used throughout the literature to encode a qualitative description of the generating model, which is often much easier to obtain than the actual mechanisms of the underlying SCM [15, 20, 17]. The goal of this paper is not to decide which set of assumptions is the best but rather to provide tools to perform the inferences once the assumptions have already been made, as well as understanding the trade-off between assumptions and the guarantees provided by the method.

Second, the true underlying causal diagrams cannot be learned only from the observational distribution in general. More specifically, there almost surely exist situations that $\mathcal{M}_1$ and $\mathcal{M}_2$ induce the same observational distribution but are compatible with different causal diagrams (see [3, Sec. 1.3] for details). With higher layer distributions (such as distributions from $\mathcal{L}_2$), it is possible to recover a more informative equivalence class of diagrams that encode additional constraints present in the input layer [12, 11, 10, 13, 22].

Q2. What is a graphical model and how can it help us in causal inference?

**Answer.** A graphical model is a modeling tool that allows one to represent a compatibility relationship between a causal diagram $\mathcal{G}$ and a collection of distributions $\mathbf{P}$. Specifically, it encodes how the topological structure of the diagram can be interpreted to impose constraints on the associated distributions. For instance, when restricting attention to $\mathcal{L}_1$ distributions (i.e., purely observational), Bayesian Networks (BNs) are the most prominent graphical models to encode conditional independence constraints of the observational distribution [14]. As we climb up the PCH and include more distributions into the collection, more constraints start to emerge. To encode the richer set of causal constraints in $\mathcal{L}_2$ distributions (i.e., interventional), the Causal Bayesian Network (CBN) was introduced [3]. More recently, CTFBN is introduced to encode the compatibility relationship between the causal diagram and $\mathcal{L}_3$ distributions (i.e., counterfactual) [1]. The models defined in this work further refine the space of $\mathcal{L}_3$ distributions by restricting to constraints that are, at least in principle, empirically falsifiable. In a nutshell, a graphical model should not be viewed merely as a causal diagram, but rather as a formal specification of the compatibility relationship between a pair $\langle \mathcal{G}, \mathbf{P} \rangle$. An example of a CBN is illustrated in Fig. 15, where missing edges in the causal diagram represent invariance constraints in the distributions.

The causal diagram in the graphical model offers a compact representation for constraints in the associated distributions. These constraints are fundamental to causal inference, as they constitute one of the three core inputs to the causal inference engine (Fig. 1). As discussed ealier, the main task in causal inference is to determine whether a query from a higher layer of the PCH can be identified as a function of observed data from lower layers. For example, the task may be to identify a causal effect $P(y|do(x))$ when only the observational data $P(\mathbf{v})$ is available. According to the Causal Hierarchy Theorem (CHT), these layers are strictly distinct, and it is impossible to ascend to a higher layer without additional assumptions about that layer [3, Thm. 1]. The constraints encoded by graphical models serve precisely this role – they encode the assumptions about higher layers that enable us to bridge the gap and make such inferences possible. Given the CBN in Fig. 15, the invariance constraint $P(Y|do(X)) = P(Y|X)$ allows us to identify the $\mathcal{L}_2$ query $P(y|do(x))$ as $P(y|x)$, which only involves observational distributions. Question 9 below will provide further details on the inferential process by explaining how the local constraints defined in a graphical model can be composed to derive additional constraints implied by the model.

Q3. Why do we need to introduce new layers to the PCH, besides the existing ones?

**Answer.** The original three layers of the PCH, capturing observational, interventional, and counterfactual distributions, provide a natural partition among distinct capabilities in causal reasoning. Layers 1 and 2 correspond to well-understood physical procedures: random sampling for observational distributions and random experimentation for interventional distributions. In contrast, Layer 3 consists of purely counterfactual quantities, that are traditionally considered detached from empirical data collection in principle. In addition, while Layers 1 and 2 are well-structured and homogeneous (each quantity within a layer having a similar interpretation), Layer 3 is more heterogeneous and contains quantities that represent different aspects of the underlying data-generating process.

More recently, Bareinboim, Forney and Pearl introduced a new experimental procedure, counterfactual randomization, that allowed one to sample directly from an $\mathcal{L}_3$ distribution [4]. This work was further extended in [19]. The introduction of counterfactual randomization reveals a finer structure within Layer 3, distinguishing between counterfactual distributions that are empirically accessible and those that are not. This fine-graining of Layer 3 is illustrated in Fig. 16. Notably, these new families of distributions have attractive properties, including well-defined symbolic languages as well as a closed set of inferential rules, as shown in this work. This new view opened up a natural way of partitioning $\mathcal{L}_3$. In this work, we studied the interplay between graphical models that inherent these features of the PCH and have the property of empirical falsifiability.

To answer the question, the new layers introduced in the refined PCH may not be necessary for all researchers. The original PCH already represents a major milestone in formalizing the logic of causal inference. Still, for some researchers, the refinement and further partitioning of Layer 3 can offer valuable insights. In particular, it allows for a more precise understanding of the trade-off between empirical falsifiability and the inferential power of

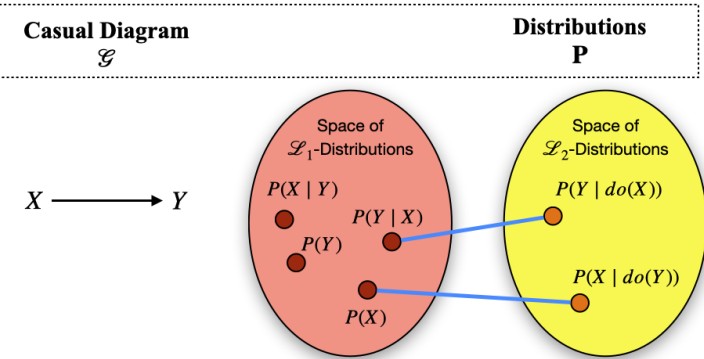

**Graphical Model - CBN**

Figure 15: A CBN is a pair $\langle \mathcal{G}, \mathbf{P} \rangle$. Blue lines represent invariant constraints in $\mathbf{P}$, which are represented by features from $\mathcal{G}$: missing directed edge from $Y$ to $X$ corresponds to the invariance constraint $P(X|do(Y)) = P(X)$ and missing bidirected edge between $X$ and $Y$ corresponds to the invariance constraint $P(Y|do(X)) = P(Y|X)$.

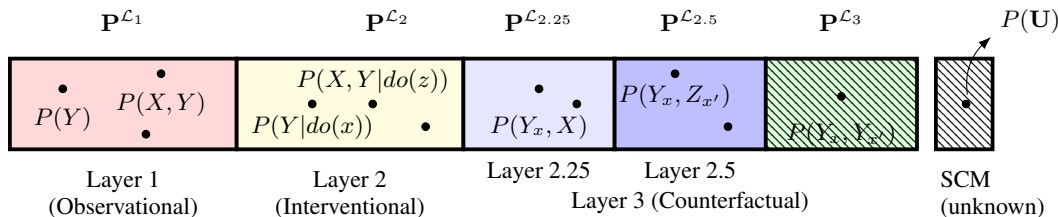

Figure 16: Pearl Causal Hierarchy (PCH*) induced by an unknown SCM $\mathcal{M}$. Layers 1 and 2 are realizable, and Layer 3 is partially realizable. The realizable portion of Layer 3 are further refined into two new layers: 2.25 and 2.5.

graphical models, and provides a tighter feedback loop between theoretical assumptions and experimental capabilities.

Q4. What is the difference between layers 2.25 and 2.5?

**Answer.** The main difference between $\mathcal{L}_{2.25}$ and $\mathcal{L}_{2.5}$ lies in the type of counterfactual randomization allowed. For $\mathcal{L}_{2.25}$, a counterfactual randomization applied to a variable $X$ assigns the same value $x$ across all its children and descendants. As a result, distributions in this layer cannot contain pairs of potential outcomes $W_{\mathbf{s}}, R_{\mathbf{t}}$ with conflicting subscripts where $x \in \mathbf{s}, x' \in \mathbf{t}$ and $x \neq x'$. In contrast, the counterfactual randomization action on a variable $X$ in $\mathcal{L}_{2.5}$ is more flexible and allows each outgoing edge from $X$ to take a different value. This flexibility leads to the possibility of some distributions in the layer to include potential outcomes with different subscripts. This difference is graphically illustrated in Fig. 3. However, all descendants of each child of $X$ must still share the same value of $x$, unless all directed paths from $X$ to the descendant are blocked by other intervened variables. This restriction stems from the rules of counterfactual randomization, which prohibit an intervention to bypass a child and directly affect a descendant's perception of $X$. In summary, the constraint on consistent subscript begins at the intervened variable $X$ in $\mathcal{L}_{2.25}$, but shifts to the children of $X$ in $\mathcal{L}_{2.5}$. These differences are reflected in the relaxed conditions that define the symbolic language of $\mathcal{L}_{2.5}$, relative to those of $\mathcal{L}_{2.25}$.

Q5. Are all distributions within Layers 2.5 realizable?

**Answer.** Theoretically, all distributions in $\mathcal{L}_{2.5}$ are realizable if every action in the maximal feasible action set is permitted. That is, *in principle*, an agent could draw samples from any distribution in this layer through experimental procedures. However, whether a distribution is realizable *in practice* depends on the physical constraints of the system. If certain actions – such as counterfactual randomization on specific variables – are not feasible, then some distributions in $\mathcal{L}_{2.5}$ will not be realizable in real-world settings [19].

| Query Layer | Graphical Model | Sufficient | Necessary |
|---|---|---|---|
| $\mathcal{L}_1$ | BN | ✓ | ✓ |
| $\mathcal{L}_1$ | CBN | ✓ | x |
| $\mathcal{L}_2$ | BN | x | ✓ |
| $\mathcal{L}_2$ | CBN | ✓ | ✓ |
| $\mathcal{L}_2$ | CBN2.25 | ✓ | x |
| $\mathcal{L}_{2.25}$ | CBN | x | ✓ |
| $\mathcal{L}_{2.25}$ | CBN2.25 | ✓ | ✓ |
| $\mathcal{L}_{2.25}$ | CBN2.5 | ✓ | x |
| $\mathcal{L}_{2.5}$ | CBN2.25 | x | ✓ |
| $\mathcal{L}_{2.5}$ | CBN2.5 | ✓ | ✓ |
| $\mathcal{L}_{2.5}$ | CTFBN | ✓ | x |
| $\mathcal{L}_3$ | CBN2.5 | x | ✓ |
| $\mathcal{L}_3$ | CTFBN | ✓ | ✓ |

Table 5: Examples of Matching between Graphical Models and Queries. Rows highlighted in green represent a match between the model and the query such that the assumptions in the model are both sufficient and necessary for making inference about the query.

The same principle applies to other layers of the PCH. For example, all distributions in $\mathcal{L}_2$ are realizable *in principle*, assuming the agent can freely intervene on all variables. However, practical constraints – such as cost, ethics, or technological barriers – may render some interventions infeasible, thereby restricting the subset of $\mathcal{L}_2$ distributions that can be realized.

Given a causal diagram and a specification of the allowed actions, one can determine whether a given set of distributions is realizable [19]. Viewed this way, the full collection of distributions in $\mathcal{L}_{2.5}$ can be interpreted as the theoretical boundary of what is empirically accessible through physical experimentation.

Q6. How does the hierarchical structure defined over graphical models provide useful information on the models?

**Answer.** The hierarchical structure over graphical models offers a clear picture of the differences in the strength of assumptions encoded by each model. In causal inference specifically, the strength of the assumptions determines what queries the model may in principle support – specifically, whether the causal inference engine can proceed and provide useful insights about the query. For instance, an $\mathcal{L}_2$ query $P(y|do(x))$ cannot be answered by a BN, which only encods $\mathcal{L}_1$ constraints that does not have the power to bridge the gap between the two layers. This limitation is formally captured by the Causal Hierarchy Theorem (CHT), which states that to answer questions at one layer, one needs assumptions at the same layer or even higher. This understanding allows practitioners to select models from the hierarchy with sufficient inferential power for the query at hand.

On the other hand, the hierarchy also provides guidance in the opposite direction – helping to identify when a model might be stronger than necessary. For instance, while any model at or above a CBN in the hierarchy can answer an $\mathcal{L}_2$ query $P(y|do(x))$, using a model that makes counterfactual assumptions (e.g., a CBN2.5) would be unnecessarily strong and harder to falsify. Therefore, knowing the hierarchy of graphical models also allows practitioners to avoid choosing models that make extra assumptions not required in the target inferential task.

Putting these observations together, Table 5 summarizes when a model is sufficient and/or necessary for queries from each layer of the PCH. In short, the hierarchy serves as a practical guide for selecting models that are both sufficient and necessary – maximizing inferential power while minimizing unfalsifiable assumptions.

Q7. What is the difference between the hierarchical structure of languages and graphical models?

**Answer.** The hierarchical structure of the languages (i.e., the PCH) defines how different families of distributions are related – specifically, each layer's distributions form a subset of those in the layer above. In parallel, the hierarchy of graphical models reflects how constraints on these distributions are encoded through the topological properties of the

causal diagram. Each graphical model at layer $i$ encodes constraints over the corresponding family of distributions in layer $i$ of the PCH. Therefore, the hierarchy of the languages directly informs the hierarchy of graphical models.

However, since a graphical model is defined as a compatibility relationship between a pair $\langle \mathcal{G}, \mathbf{P} \rangle$, the expressiveness of the topological features in $\mathcal{G}$ also plays a critical role. As we move up the hierarchy, the causal diagrams must support richer or more expressive interpretations of missing edges to capture the increasingly complex constraints required by higher-layer distributions. Both hierarchies are illustrated in Fig. 6, where square boxes depict the hierarchy over distributions, and round boxes represent the hierarchy over the constraints encoded by graphical models.

Q8. Why should a data scientist care about the trade-off between expressive power and empirical falsifiability of the graphical models?

**Answer.** In any modeling task, it is generally desirable to construct a model that accurately reflects the underlying generative process while also supporting future inferential tasks. Achieving stronger inferential power often requires incorporating stronger assumptions into the model. However, these assumptions can make the model more prone to errors that does not match with reality. Empirical falsifiability acts as a form of regularization, enabling the data scientist to identify, falsify and possibly correct wrong assumptions using empirical evidence. As a result, the model can yield more reliable and trustworthy causal conclusions. The importance of falsifiability echoes Karl Popper's philosophy, which argues that scientific theories must be testable and refutable – setting science apart from pseudoscience [18]. Thus, understanding where each graphical model falls on the spectrum of expressive power versus empirical falsifiability is essential for practitioners who align with Popper's principle.

Q9. What are the differences between local constraints and global constraints?

**Answer.** As discussed earlier when we introduce the inferential machinery for CBN2.25/CBN2.5, local constraints refer to those that are defined over distributions involving a variable and its parents, and they are the constraints that are explicitly stated in the definitions of graphical models. For example, the local constraints in a BN are the conditional independencies of the form $P(v_i|\mathbf{pa}_i, \mathbf{nd}_i) = P(v_i|\mathbf{pa}_i)$, where $\mathbf{pa}_i$ denotes the parents and $\mathbf{nd}_i$ the non-descendants of $V_i$. Given a BN over the chain diagram $X \rightarrow Z \rightarrow W \rightarrow Y$, the local constraints include $P(w|z, x) = P(w|z)$ and $P(y|w, z, x) = P(y|w)$.

Global constraints, on the other hand, involve arbitrary subsets of variables, possibly far apart in the causal diagram. These constraints are not explicitly listed in the model's definition but can be derived by composing local constraints. For example, given the same BN over the chain above, a global constraint is $P(y|z, x) = P(y|z)$, where the direct parent of $Y$, namely $W$, is no longer explicitly conditioned on.

This distinction highlights the role of local constraints as a basis for implying the full set of global constraints that a graphical model implies, as illustrated in Fig. 10. This relationship is mirrored in the connection between a graphical model and its associated inferential calculus: the calculus rules form the closure of all global constraints that logically follow from the local ones encoded in the model.

The process by which local constraints can be composed to yield global constraints was illustrated in Example 3. We revisit this idea with a new example in Fig. 10. Consider a CBN over the chain diagram $X \rightarrow Z \rightarrow Y$. The local constraints specified in the definition of the CBN are depicted as connecting lines between nodes within the small yellow circle. These local constraints can imply additional constraints not explicitly listed in the definition. One such global constraint is $P(y|do(x)) = P(y|x)$, represented by the red connection line in the figure. This global constraint can be derived by composing – or "gluing" – a sequence

of local invariance constraints, shown as blue connection lines.

$$P(y|do(x)) = \sum_z P(y|do(x),z)P(z|do(x)) \qquad \text{(Probability Axiom)} \qquad (161)$$

$$= \sum_z P(y|do(xz))P(z|do(x)) \qquad \text{(Cond. (iii) of Def. 19)} \qquad (162)$$

$$= \sum_z P(y|do(z))P(z|do(x)) \qquad \text{(Cond. (ii) of Def. 19)} \qquad (163)$$

$$= \sum_z P(y|z)P(z|x) \qquad \text{(Cond. (iii) of Def. 19)} \qquad (164)$$

$$= \sum_z P(y|xz)P(z|x) \qquad \text{(Cond. (i) of Def. 19)} \qquad (165)$$

$$= P(y|x) \qquad \text{(Probability Axiom)} \qquad (166)$$

In summary, although not all constraints are explicitly included in the local basis of a graphical model definition, many are implied through its structure. Since the 1980s, this ability to encode a parsimonious, polynomial-sized set of local constraints that implicitly represent an exponential number of global constraints has been an attractive feature contributing to the popularity and usefulness of graphical models in inferential tasks.

Q10. What is the connection between realizability and empirical falsifiability?

**Answer.** Realizability is a property of distributions, indicating that an agent can draw samples from them through physical experimentation. For example, if an agent can intervene on a variable $X$ and fix it to a value $x$, it gains access to the interventional distribution $P(\mathbf{v} \mid do(x))$ in layer $\mathcal{L}_2$.

In the context of graphical models, empirical falsifiability is property of constraints over these distributions. To empirically falsify a constraint, the agent must have the experimental capabilities to draw samples from all distributions involved in the constraint. In other words, the constraint's falsifiability requires the realizability of the associated distributions. For instance, testing the constraint $P(y \mid do(x,z)) = P(y \mid do(x))$ requires the ability to sample from both $P(y \mid do(x,z))$ and $P(y \mid do(x))$. Whether this is feasible depends on the experimental capabilities and limitations of the system in question.

Q11. What is the difference between an SCM and Layer 3 distributions or Layer 3 graphical models?

**Answer.** An SCM is a more granular level model with details about the exogenous variables $\mathbf{U}$, which induces the full set of distributions over the endogenous variables $\mathbf{V}$ in the PCH, as illustrated in Fig. 4 and Fig. 8. Specifically, given a distribution over the exogenous variables $P(\mathbf{u})$ and the structural equations that determine each endogenous variable as a function of its parents (both endogenous and exogenous), we can compute all distributions over the endogenous variables following the formula in the PCH definition (Def. 11). In contrast, the PCH abstracts away from the exogenous variables, treating them as hidden background factors unobserved by the agent. As a result, Layer 3 distributions and its corresponding graphical models are defined solely in terms of the endogenous variables.

Given an SCM, it is also possible to evaluate individual level effects when the exogenous state of a specific unit $\mathbf{u} \in Val(\mathbf{U})$ is known, by solving the set of mechanisms following the topological order of evaluation. Layer 3 distributions and graphical models, on the other hand, offer population-level descriptions of causal relationships, without access to individual-level information.

In a nutshell, an SCM provides full access to Layer 3 distributions and graphical models, as it encodes the necessary generative mechanisms. However, the reverse does not hold: Layer 3 distributions or graphical models do not determine a unique SCM, since an SCM requires additional, often unobservable, information about the exogenous variables and structural mechanisms.

Q12. Given that the constructive procedure for the causal diagram is the same, why do we need, or even have, different layers of graphical models?

**Answer.** Even though the same causal diagram $\mathcal{G}$ is shared across many different models, the compatibility relationships it represents differ depending on the model. As discussed

earlier, a graphical model is a pair $\langle \mathcal{G}, \mathbf{P} \rangle$, where graphical feature in $\mathcal{G}$ are interpreted to represent constraints in $\mathbf{P}$. As $\mathbf{P}$ expands to include distributions from higher layers of the PCH, the set of constraints that the graph must represent also becomes richer. As a result, each missing edge is required to encode stronger and more expressive constraints over a broader class of distributions. This is illustrated in Example 6 and Table 4.

