# OpenReview forum: "A Hierarchy of Graphical Models for Counterfactual Inferences"
_NeurIPS.cc/2025/Conference — NeurIPS 2025 poster_

### Official Review · Reviewer_pvuV · 2025-06-25

**Clarity:** 2
**Significance:** 3
**Originality:** 3
**Rating:** 4
**Confidence:** 4

**Summary:**

PCH consists of three layers, observational, interventional, and counterfactual. When the underlying SCM is fully known, then we can compute any query to the model. However, in reality, the model might not be fully known. At the interventional level, it is possible, at least in principle, to obtain full knowledge by doing experiments. This it not true for the counterfactual level. in [19], the concept of counterfactual realizability was introduced, which formalizes the concept of being able to sample from a counterfactual distribution by doing experiments. Not all counterfactual distributions are realizable. In the present paper, the authors formalize two subclasses of the counterfactual layer of PCH by restricting the kind of counterfactual worlds that can be present. These subclasses are realizable in the sense of [19].

The main results are the following:
1) The proof that two corresponding classes of causal Bayesian networks capture the distributions defined by the new layers (Thm 1).
2) The adaption of the counterfactual calculus to the subclasses and the proof that a quantity is identifiable if it can be expressed in terms of the given distribution using the rules (Thm 2).
3) The proof that the distributions defined in terms of SCMs of the different layers form a hierarchy (Thm 3) as well as the constraints encoded in the corresponding graphical models (Thm 4).

**Questions:**

No particular question, but I am thankful for any comments regarding the mentioned weaknesses.

**Ethical Concerns:**

["NO or VERY MINOR ethics concerns only"]

**Final Justification:**

I have raised my score by 1 taking the discussions with the authors into account.

The paper introduces two new layers in PCH between interventional and counterfactual reasoning. Per se, this is a valuable contribution. However, the paper still has a few shortcomings, but I think the new conceptual contributions outweigh these shortcomings.

My main points of critic are:
- shortcomings in the presentation, which will be hopefully improved
- some of the proofs are of incremental nature (which to some extend cannot be helped)
- relevance of the new introduced layer (which time has to tell)

**Limitations:**

yes

**Quality:**

2

**Strengths And Weaknesses:**

Strengths:
- The paper makes interesting new theoretical contributions by defining sublayers of the counterfactual layer of PCH which might be easier to falsify by experiments. These layers might be easier to understand, since the way counterfactual worlds can be used is restricted.

Weaknesses:
- On the technical side, I think the contributions are rather incremental.
  - The proof of Theorem 1 is just a check of the properties, which is similar to proofs for the standard layers. Which is to some extend to be expected, as the construction procedures for the CBNs are the same.
  - In the proof of Theorem 2, soundness is obvious and completeness follows by the same argument as for layer 3, even using the same algorithm.
  - For the proof of Theorem 3, one only has to check that layer 2.25 is subsumed by layer 2.5, which is quite obvious since Definition 2 generalizes Definition 1, as is the fact that it transfers to the corresponding CBN.
- One of the selling points of the paper is that the new layers allow empirical falsifiability. The example in the paper are of mathematical/explanatory nature, which is fine. However, the examples one finds in other papers (like changing the color of cars in images) are a little contrived.

Comments:

- I think the naming of the new layers is not good. The numbers are arbitrary. Why not 2.33 and 2.66? Furthermore, if someone next introduces a layer between interventional and counterfactual that is not comparable to your new layers, you created a mess, since real numbers have a total order. That being said, already numbering the layers of PCH might have been a mistake, since there is no natural numerical parameter.
- The paper is not easy to read (which to some part cannot be avoided, since counterfactual reasoning is complicated). Section 2 is pretty dense, while I find section 3 rather lengthy. While Appendix A contains lots of information, I find it to lengthy to be a good reference manual but contains to few explanations to be a good introduction. I do not think that the [..]-notation is explained anywhere, but I might have overlooked it.

Minor comments:

- line 110: V -> U
- line 113: Val -> \mathit{Val} (or whatever you prefer)
- line 118: \in -> \subseteq
- line 158: maybe use x and x'. The indices do not correspond to the indices used in Definition 1
- page 5, footnote: An "Appendix" is missing (most likely a c in \cref)
- line 222: Definition 11 only appears in the appendix
- Eq (35): fi -> f_i
- Definition 12: The second sentence is not a proper mathematical definition
- It might be helpfull in Def. 18 to discuss the relation to Def. 13
- Definition 24 is also not a proper mathematical definition. What is a randomization device is this context?
- line 693: a blank is missing
- Lemma F.1 seems to be folklore to me.
- line 1018 2.25 -> 2.5
- line 1247: looks like some latex issue

---

> ### Author Rebuttal · Authors · 2025-07-31
>
> We thank the reviewer for the time and effort in reviewing our paper. While we appreciate the thoughtful feedback, we believe a few misunderstandings may have contributed to the evaluation being a bit harsh. We respectfully ask the reviewer to reconsider our paper based on the clarifications below.
>
> >*Weakness 1*: On the technical side, I think the contributions are rather incremental.
> The proof of Theorem 1 is just a check of the properties, which is similar to proofs for the standard layers. Which is to some extent to be expected, as the construction procedures for the CBNs are the same.
> In the proof of Theorem 2, soundness is obvious and completeness follows by the same argument as for layer 3, even using the same algorithm.
> For the proof of Theorem 3, one only has to check that layer 2.25 is subsumed by layer 2.5, which is quite obvious since Definition 2 generalizes Definition 1, as is the fact that it transfers to the corresponding CBN.
>
> Thank you for your careful reading of the proofs and detailed comments. While we agree the proof techniques share structural similarities with CTFBNs, we respectfully disagree that the contributions are merely incremental. The resemblance arises naturally because layers 2.25 and 2.5 also concern counterfactuals. Yet, the novelty of the work does not lie solely in the complexity of the proofs, but more in the implications of highlighting the heterogeneity of the counterfactual space, a space that has remained largely underexplored at least for the past three decades, since Pearl’s groundbreaking 1995 Biometrika paper that started the field in the context of AI.
>
> As discussed in the introduction, the formal definitions of the intermediate two layers provide a foundation for “a finer-grained understanding of the region between $\mathcal{L}_2$ and $\mathcal{L}_3$ of the PCH”, and “a precise formalization of the empirical heterogeneity of distributions between” these two layers. The corresponding graphical models inherit the empirical properties in their assumptions, which equip data scientists with a systematic way to evaluate and justify assumptions based on empirical evidence, expert knowledge, or domain-specific reasoning.
>
> To illustrate the practical value of these layers, we provide examples throughout the main text. In particular, we illustrate in Examples 5 (ETT) and 7 (NDE) that important causal quantities across disciplines lie within the two new layers and can be identified by the corresponding graphical models. These examples show that our framework is not only theoretically motivated but also practically meaningful in real-world applications. A significant amount of discussion in the literature follows precisely from the lack of these clearly delineated spaces that we circumscribe in our work.
>
> We believe that recognizing the two intermediate layers refines our understanding of the causal hierarchy and opens up new directions for working with counterfactual assumptions in a more granular and transparent way. We will clarify this perspective more explicitly in the revised version and appreciate your feedback for prompting it.
>
> >*Weakness 2*: One of the selling points of the paper is that the new layers allow empirical falsifiability. The examples in the paper are of mathematical/explanatory nature, which is fine. However, the examples one finds in other papers (like changing the color of cars in images) are a little contrived.
>
> Thanks for the positive recognition of the examples in our paper.
>
> Regarding the “changing the color of cars in images” example, we believe you are referring to Example 1 in [Raghavan & Bareinboim, 2025], since it is not in our paper. Based on our understanding, that example is designed for illustrative purposes, showing how counterfactual randomization can be implemented in real-world contexts via a counterfactual mediator.
>
> We agree that having natural and intuitive examples is important for conveying the practical value of the new layers. The empirical accessibility of Layers 2.25 and 2.5 can also be illustrated through many widely recognized real-world studies. One example is the field experiment on labor market discrimination by [Bertrand & Mullainathan,2004]. In this study, the authors experimentally manipulated perception of race by changing the names of fictitious job applicants to study racial discrimination in hiring. With access to both the original names and the hiring outcomes based on altered names, the resulting data corresponds to a counterfactual distribution in Layer 2.25. As a result, assumptions made over this distribution are also empirically testable. Another example showing empirical access to NDE is the gender bias study in language models by [Vig et al., 2020].
>
> Examples like these highlight how counterfactual randomization is feasible in real-world studies and support the practical relevance of the two new layers. They also demonstrate that the empirical falsifiability of counterfactual assumptions, like those in CBN2.25/2.5, is not only a theoretical construct but a property that can be exploited in practice.
>
> We recognize that examples in our paper like Example 5(ETT) and Example 7(NDE) are primarily explanatory, without details on specific application domains. However, the causal quantities in these examples are common across scientific fields, and our results directly apply whenever they are of interest. We will add discussion in the revised version to clarify the connection between our theoretical results and real-world applications, and citations to empirical studies where such quantities are evaluated. We also plan to explore broader empirical evaluations in future work. Thanks for highlighting this important point.
>
> >*Comment 1*: I think the naming of the new layers is not good. The numbers are arbitrary. Why not 2.33 and 2.66? Furthermore, if someone next introduces a layer between interventional and counterfactual that is not comparable to your new layers, you created a mess, since real numbers have a total order. That being said, already numbering the layers of PCH might have been a mistake, since there is no natural numerical parameter.
>
> Right, the numbers may be a bit arbitrary, there is indeed no natural numerical parameter. The current naming arose organically: we introduced Layer 2.5 first as a natural midpoint between Layers 2 and 3, and then named Layer 2.25 to reflect its position between Layers 2 and 2.5. While the absolute values are not meaningful, the relative ordering reflects the intended hierarchy among the layers. That said, we are happy to consider alternatives and suggestions from the reviewers, maybe more interpretable names for the two new layers (perhaps following a similar approach as the other three layers of the PCH: observational, interventional, and counterfactual.)
>
> >*Comment 2*: The paper is not easy to read (which to some part cannot be avoided, since counterfactual reasoning is complicated). Section 2 is pretty dense, while I find section 3 rather lengthy. While Appendix A contains lots of information, I find it too lengthy to be a good reference manual but contains too few explanations to be a good introduction. I do not think that the [..]-notation is explained anywhere, but I might have overlooked it.
>
> We acknowledge that some sections are a bit dense, largely due to the need to compress content within the page limit while preserving the technical contributions. Section 2 is a bit dense as it introduces core definitions and theorems (otherwise, based on our experience, the paper would be deemed as not self-contained). Section 3 was given slightly more space to highlight the significance of the two new layers in their role of refining the hierarchy of causal models. With the additional page allowance for the revised version, in case the paper is accepted, we plan to rebalance the sections and incorporate comments from reviewers to improve clarity and flow.
>
> For Appendix A, we agree that it currently sits between a reference and an introduction. It is primarily meant to serve as a concise reference for readers who want a quick overview of the SCM framework and the existing graphical models within it. For a more in-depth understanding, we encourage readers to consult the original sources cited in the appendix (e.g., [Bareinboim et al., 2022]). That said, we recognize that it would benefit from improved structure and narrative flow, and we will revise it accordingly to make it more reader-friendly.
>
> We also welcome any suggestions to improve the paper’s structure and flow that the reviewer may see fit.
>
> As for the [..]-notation, we apologize for the potential oversight. The square brackets indicate the variables being intervened upon to generate the potential outcome variables. For example, in $V\_{i\_{[\mathbf{x}\_i]}}$, $i$ is the index for the variable $V_i$, while $\mathbf{x}\_i$ inside the square bracket denotes the interventional value set. This was introduced to distinguish intervention targets from subscripts used for indexing variables. We will add a clarifying note in the revised version to ensure the notation is clear.
>
> > *Minor Comments*:
> line 110: V -> U
> (...)
>
> Many thanks, we will correct these typos in the revised version.
>
> *References*:
> - [Raghavan & Bareinboim, 2025] Arvind Raghavan and Elias Bareinboim. Counterfactual Realizability. ICLR, 2025.
> - [Bertrand & Mullainathan, 2004] Marianne Bertrand and Sendhil Mullainathan. Are Emily and Greg more employable than Lakisha and Jamal? A field experiment on labor market discrimination. AER, 2004.
> - [Vig et al., 2020] Jesse Vig, Sebastian Gehrmann, Yonatan Belinkov, Sharon Qian, Daniel Nevo, Yaron Singer, Stuart Shieber. Investigating Gender Bias in Language Models Using Causal Mediation Analysis. NeurIPS, 2020.
> - [Bareinboim et al., 2022] Elias Bareinboim, Juan D Correa, Duligur Ibeling, and Thomas Icard. On Pearl’s Hierarchy and the Foundations of Causal Inference. ACM, Special Turing Series, 2022.

---

> > ### Comment · Reviewer_pvuV · 2025-08-03
> >
> > Thank you very much for your detailed response, it will take your arguments into account for my final evaluation. I appreciate the new concepts of your paper and think that more real-world examples will make them much clearer. Furthermore, I still suggest to give the new layers more interpretable names similar to observational, interventional, and counterfactual.

---

> > > ### Author Response · Authors · 2025-08-05
> > >
> > > We appreciate your follow-up and your willingness to consider our responses in your evaluation. We will incorporate this discussion and your suggestions into the next revision of the paper. Since we believe we have addressed all the concerns raised, we would be happy to clarify any remaining issues if anything is still unclear. Thank you again for your feedback.

---

> > > > ### Comment · Reviewer_pvuV · 2025-08-05
> > > >
> > > > Yes, thank you. You addressed all my concerns.

---

### Official Review · Reviewer_mqbs · 2025-07-02

**Clarity:** 3
**Significance:** 4
**Originality:** 4
**Rating:** 5
**Confidence:** 2

**Summary:**

The paper introduces two new classes of causal graphical models, CBN2.25 and CBN2.5, which encode constraints for distributions made experimentally accessible via counterfactual randomization. These models bridge the gap between interventional (L2) and counterfactual (L3) layers in Pearl's Causal Hierarchy (PCH). The authors demonstrate a novel hierarchy of graphical models where higher-level models offer greater expressive power at the cost of reduced empirical falsifiability. They provide theoretical guarantees for their calculus, proving soundness and completeness for inference within these new frameworks.

**Questions:**

1. How do the computational costs of ctf-calculus in CBN2.25/CBN2.5 compare to Pearl’s do-calculus or ctf-calculus in CTFBN? Could the additional constraints impact scalability?
2. What concrete applications have been tested with these models, and how do their results compare to prior methods like CTFBN? For instance, in estimating ETT (Example 5), are there cases where CBN2.25/CBN2.5 fail due to insufficient assumptions?
3. The authors discuss falsifiability but do not address potential societal risks (like biases in counterfactual reasoning). Are there scenarios where CBN2.25/CBN2.5 could amplify ethical concerns compared to prior models?

**Ethical Concerns:**

["NO or VERY MINOR ethics concerns only"]

**Final Justification:**

See the response to the authors.

**Limitations:**

The authors adequately address limitations by explicitly analyzing trade-offs between model expressiveness and falsifiability (Section 3).

**Quality:**

4

**Strengths And Weaknesses:**

## Quality
### Strength
1. The paper is technically robust. Definitions of distributions (Def 1/2), graphical constraints (Def 3/4), and theorems linking models to SCMs (Thm 1) are precise. Lemma 1 (CRS criterion) provides a critical tool for verifying distribution membership in PCH layers, ensuring theoretical validity.
2. The authors prove that ctf-calculus is sound and complete for CBN2.25/CBN2.5 (Thm 2), which is foundational for their utility in causal inference. This strengthens the practical relevance of the proposed machinery.
3. The refinement of L3 into experimentally accessible sublayers (L2.25/L2.5) and formalization of a hierarchy connecting BN, CBN, and CTFBN is conceptually original. Thm 4 rigorously establishes how constraints propagate across layers.
### Weaknesses:
1. The paper does not provide empirical validation or experiments to demonstrate the practical utility of these models. While theoretical completeness is ensured, real-world performance remains unexplored.

---
## Clarity
### Strength
1. The paper follows a logical flow: problem motivation -> formalisms -> hierarchical analysis. Figures 3–6 and tables like Table 1 aid conceptual understanding.
2. The paper systematically maps new models into PCH (Thm 3) and establishes their hierarchical relationships via constraint nesting (Thm 4). This enhances conceptual clarity for model selection in causal inference tasks.
### Weaknesses:
1. Dense mathematical notation in Def 1/2 and equations for distributions. For example, the distinction between P^L_{2.25} and P^L_{2.5} could benefit from clearer intuitive explanations alongside formal definitions.

---
## Significance
### Strength
1. The work advances the understanding of counterfactual inference by bridging theoretical limitations (non-experimental L3) with empirical feasibility via counterfactual randomization. This is critical for AI systems requiring robust causality foundations, particularly in fields like healthcare or policy analysis.
2. The hierarchy framework provides a principled tool for model selection: researchers can choose models that balance expressive power and falsifiability per task.
### Weaknesses:
1. The paper focuses almost exclusively on theoretical contributions without illustrating their application to real-world problems. For instance, it lacks examples of how these models improve over existing methods, like CTFBN, in scenarios like estimating ETT or NDE.

---
## Originality
### Strength
1. The formalization of L2.25/L2.5 as refinements of PCH’s L3 is novel, especially the introduction of AMWN for inferential machinery and the CRS criterion (Lemma 1).
2. The paper clearly distinguishes its work from prior models and situates them within an expanded PCH framework (Thm 3/4). This contextualizes the contribution without unnecessary redundancy.

---

> ### Author Rebuttal · Authors · 2025-07-29
>
> We appreciate your positive assessment and valuable feedback. Please find our detailed responses to your questions and comments on weaknesses below.
>
>  >*Quality*: The paper does not provide empirical validation or experiments to demonstrate the practical utility of these models. While theoretical completeness is ensured, real-world performance remains unexplored.
>
> We acknowledge that the primary focus of this paper is on establishing the proper foundations for a hierarchy of graphical models in causal reasoning. We believe the foundations are important so that the applications can flourish. Additionally, no paper can do everything (i.e., solve all problems), and we hope the reviewer can appreciate this point. Although the paper is theoretically profound and very mathematical, we still attempted to include as many illustrative examples as possible to highlight the basic concepts and how the various moving parts connect, as well as to help the community start visualizing connections with real-world applications. For instance, Examples 5 and 7 demonstrate how the effect of treatment on the treated (ETT) and the natural direct effect (NDE) can be identified using the constraints encoded in CBN2.25 and CBN2.5, respectively. We agree that a broader empirical evaluation is a crucial future step, and we look forward to continuing the empirical work.
>
> >*Clarity*: Dense mathematical notation in Def 1/2 and equations for distributions. For example, the distinction between P^L_{2.25} and P^L_{2.5} could benefit from clearer intuitive explanations alongside formal definitions.
>
> We acknowledge that the notation in Definitions 1 and 2 is somewhat dense, and we aimed to provide as much intuitive explanation as possible to support the reader's understanding. For example, immediately following Definition 2, we highlight that “the key difference between $\mathcal{L}\_{2.25}$ and $\mathcal{L}\_{2.5}$ lies in how the distributions are indexed: $\mathcal{L}\_{2.25}$ are indexed by specific interventional value sets $\mathbf{x} \in Val(\mathbf{X})$, while $\mathcal{L}\_{2.5}$ are indexed by interventional variable sets $\mathbf{X}$. The more refined index for $\mathcal{L}\_{2.25}$ creates more restrictions on the expressiveness for its distributions, which is also reflected in the differences between conditions of the two definitions.” This distinction is further illustrated in Example 1, where distributions from the two layers are compared based on a specific SCM.
>
> However, due to space constraints, additional explanations and examples were included in Appendix B.2. In particular, we believe that Figure 10 offers an intuitive visual comparison of Layers 2, 2.25, and 2.5. Given the additional one page allowed and if the reviewers deem it suitable, we would be happy to relocate this figure to the main text in the revised version to enhance clarity.
>
> >*Significance*: The paper focuses almost exclusively on theoretical contributions without illustrating their application to real-world problems. For instance, it lacks examples of how these models improve over existing methods, like CTFBN, in scenarios like estimating ETT or NDE.
>
> As explained earlier, we agree that the paper is primarily theoretically positioned. That said, we did include concrete examples, Example 5 (ETT) and Example 7 (NDE), to illustrate that the constraints encoded in CBN2.25 and CBN2.5 are sufficient to identify ETT and NDE, respectively. This demonstrates that some of the constraints encoded by CTFBN are, in fact, not necessary for identifying queries that lie in $\mathcal{L}\_{2.25}$ and $\mathcal{L}\_{2.5}$.
>
> The same idea is also illustrated in Table 1 (reproduced below; see also Table 4, for an even more detailed view), where we show that to identify an $\mathcal{L}\_{2.5}$ query, constraints encoded in CBN are not sufficient, while those encoded in CTFBN are sufficient but not necessary. In contrast, CBN2.5 provides precisely the right set of constraints, which are both necessary and sufficient. These examples clarify how CBN2.25 and CBN2.5 refine the space of graphical models by precisely aligning the level of constraints with the layer of the query.
>
> | Q Layer | GM           | Suff.                        |Nec.      |
> |----------|---------------|----------|--------------|
> | $\mathcal{L}\_{2.5}$      | CBN | x   | $\checkmark$ |
> | $\mathcal{L}\_{2.5}$    | CBN2.5 | $\checkmark$   | $\checkmark$ |
> | $\mathcal{L}\_{2.5}$   | CTFBN | $\checkmark$  |  x  |
>
> >*Question 1*: How do the computational costs of ctf-calculus in CBN2.25/CBN2.5 compare to Pearl’s do-calculus or ctf-calculus in CTFBN? Could the additional constraints impact scalability?
>
> Based on the number of encoded constraints in the models, the computational cost of ctf-calculus in CBN2.25 and CBN2.5 is expected to lie between that of do-calculus in CBN and ctf-calculus in CTFBN. However, as with do-calculus, exhaustively searching for all possible derivations based on the rules is known to be computationally hard in general.
>
> For both CBN and CTFBN, prior work has developed algorithmic identification approaches based on tree structures that significantly improve scalability and efficiency by solving the problem in a more systematic way (for details, please see Section 4.4 and 5.4 in [Bareinboim, 2025]. Although such methods are not yet fully explored for CBN2.25 and CBN2.5, we believe it may be possible to apply similar approaches to these two models as well. This is a natural extension of the foundational work presented in this paper, and we plan to explore these computational aspects further in future work.
>
> >*Question 2*: What concrete applications have been tested with these models, and how do their results compare to prior methods like CTFBN? For instance, in estimating ETT (Example 5), are there cases where CBN2.25/CBN2.5 fail due to insufficient assumptions?
>
> As this paper is focused on establishing the theoretical foundations of the proposed hierarchy of graphical models, we have not yet conducted systematic empirical evaluations based on real-world datasets. The examples included are primarily designed to provide a clear picture of the relative positions of the models in the hierarchy, so they do offer information about comparisons between the two models and the existing models, CBN and CTFBN.
>
> On the more substantive aspect regarding Example 5, we confirm that there are cases where the ETT is not identifiable. For instance, when there is an unobserved confounder between $X$ and $Y$, the counterfactual independence constraint $Y_x\perp\\!\\!\\!\perp X$ no longer holds. As a result, $P(Y_x,X)$ will not be identifiable from the observational distribution $P(\mathbf{V})$. However, the key point is that ETT lies in $\mathcal{L}\_{2.25}$, and its identifiability can be determined by a CBN2.25. In other words, the constraints encoded in CBN2.25 are sufficient for deciding if ETT is identifiable or not, and there is no need to assume additional constraints from higher layers, $\mathcal{L}\_{2.5}$ and $\mathcal{L}\_{3}$, as encoded in CBN2.5 and CTFBN.
>
> >*Question 3*: The authors discuss falsifiability but do not address potential societal risks (like biases in counterfactual reasoning). Are there scenarios where CBN2.25/CBN2.5 could amplify ethical concerns compared to prior models?
>
> Thank you for raising this important issue. We agree that the ethical implications of causal modelling, particularly those involving counterfactuals, deserve careful evaluation. Specifically, in situations where the assumptions are not well-justified, there could be potential risks of encoding or amplifying biases.
>
> That said, we believe that the newly proposed framework can actually aid in mitigating such risks by making assumptions more transparent and explicit. The models we introduce offer a systematic approach to classifying assumptions based on their empirical falsifiability. This transparency allows practitioners to more rigorously assess which assumptions are warranted in a given application, whether via data, expert knowledge, or domain-specific validation.
>
> Still, we recognize that ethical and societal implications must be addressed more fully in follow-up work, especially when applying these models in sensitive domains, such as,  healthcare and criminal justice. We will include a brief discussion of these concerns in the revised version to highlight the potential risks and the responsibility that comes with using these models involving counterfactual assumptions. Thank you again for sharing this thought.
>
> *References*:
> - [Bareinboim, 2025] Elias Bareinboim. Causal Artificial Intelligence: A Roadmap for Building Causally Intelligent Systems, 2025.

---

> > ### Comment · Reviewer_mqbs · 2025-08-05
> >
> > Thank you for your comprehensive rebuttal. I acknowledge that the work is more theoretical in nature than application-oriented, and I appreciate that you have thoroughly addressed all of my questions and concerns. Your responses have been satisfactory, and I will maintain my positive evaluation score.

---

### Official Review · Reviewer_u4aR · 2025-07-02

**Clarity:** 2
**Significance:** 4
**Originality:** 4
**Rating:** 5
**Confidence:** 4

**Summary:**

This is a theoretical paper on the lack of falsifiability in the third, counterfactual layer of Pearl's hierarchy. Two new classes of models to express counterfactual quantities that are experimentally accessible are presented, along with a sound and complete calculus (mostly based on Bareinboim's CTF calculus).

**Questions:**

As mentioned, the supplementary material contains a long (~40 pages) version of the paper, which is very clear and well-organised. The same cannot be said for the 10-page version, which is obviously more dense, but also less precise in its organisation of the contents. A natural question is whether the authors believe they could make the conference version of their work entirely self-consistent and clearer than what appears to be now.

Example 7 is excellent to advocate the importance of the "2.5" layer. Is it possible to do something similar for "2.25"?

Does the calculus derived here make the standard CTF calculus useless?

**Ethical Concerns:**

["NO or VERY MINOR ethics concerns only"]

**Final Justification:**

After reading the other reviews and the (satisfactory) rebuttals I confirmed my positive opinion about the paper.

**Limitations:**

Yes.

**Paper Formatting Concerns:**

-

**Quality:**

4

**Strengths And Weaknesses:**

[+] This paper might be very influential, as it extends the classical Pearlian hierarchy using the key concept of experimental falsifiability
[+] The counterfactuals considered by this paper include some relevant queries for applications (ex. NDE, which is important for fairness analysis)
[-] The paper is a shrunk version of a much longer paper (included in the supp-mat). The shrinking process was a bit hurried, which made the clarity of the short version of the paper somewhat poor.

---

> ### Author Rebuttal · Authors · 2025-07-29
>
> Thank you for taking the time to review our work and for your positive assessment. We respond to your questions below.
>
> >*Weakness*: [-] The paper is a shrunk version of a much longer paper (included in the supp-mat). The shrinking process was a bit hurried, which made the clarity of the short version of the paper somewhat poor.
>
> >*Question 1*: As mentioned, the supplementary material contains a long (~40 pages) version of the paper, which is very clear and well-organised. The same cannot be said for the 10-page version, which is obviously more dense, but also less precise in its organisation of the contents. A natural question is whether the authors believe they could make the conference version of their work entirely self-consistent and clearer than what appears to be now.
>
> Indeed, the compression step made the paper a bit dense, even though our goal was to preserve the core technical contributions and the more interesting results in the main body. To answer your question, yes, with the additional page allowed in the camera-ready version, we will revise the current version by incorporating comments from reviewers to improve consistency, structure, and clarity, while maintaining the technical depth of the work.
> We also welcome specific suggestions regarding content reordering or additional examples that could improve clarity and flow.
>
> >*Question 2*: Example 7 is excellent to advocate the importance of the "2.5" layer. Is it possible to do something similar for "2.25"?
>
> Yes, Example 5 highlights the importance of Layer 2.25. It shows that the effect of treatment on the treated (ETT), $P(Y_x,X)$, a quantity of significant practical interest across the sciences, belongs to $\mathcal{L}\_{2.25}$. Moreover, it is identifiable from the observational distribution $P(\mathbf{V})$ given the causal assumptions encoded in the CBN2.25 associated with the causal diagram in Figure 4(a).
>
> >*Question 3*: Does the calculus derived here make the standard CTF calculus useless?
>
> No, the calculus for CBN2.25 and CBN2.5 does not make the calculus for CTFBN useless. As discussed in section 2.3 and FAQ Question 9, the constraints listed in the definitions of CBN2.25 and CBN2.5 are local, namely, they involve counterfactual variables with their parents in the subscripts. The corresponding calculus can be viewed as an algorithm to derive all global constraints that are implied by the local constraints listed in the definitions. The same idea applies to CTFBN and its calculus as well.
>
> Since CTFBN encodes a larger set of local constraints than CBN2.25 and CBN2.5, its calculus also allows for the derivation of more global constraints that are implied by these additional assumptions. For example, consider the simple causal chain $X\rightarrow Z \rightarrow Y$. The distribution $P(Y\_x, Y\_{x’},X)$ lies beyond layer 2.5 because it joins two potential outcomes over $Y$ with different subscripts. As a result, CBN2.25 and CBN2.5 and their corresponding calculus cannot make any statements about this distribution. In contrast, by leveraging the richer local assumptions encoded in CTFBN, the ctf-calculus can derive that $P(Y\_x, Y\_{x’},X)=P(Y\_x, Y\_{x’})P(X)$.
>
> In summary, ctf-calculus remains a valuable tool for deriving global constraints implied by stronger assumptions that lie beyond Layer 2.5.

---

> > ### Comment · Reviewer_u4aR · 2025-08-06
> >
> > Thanks for your rebuttal. All my doubts have been clearly addressed and I don't need further interactions. I keep my very positive opinion and evaluation about this work.

---

### Official Review · Reviewer_XJnG · 2025-07-13

**Clarity:** 3
**Significance:** 4
**Originality:** 4
**Rating:** 5
**Confidence:** 3

**Summary:**

The paper proposes two novel types of causal queries, labeled $L_{2.25}$ and $L_{2.5}$, that lie between the interventional ($L_2$) and counterfactual ($L_{3}$) layers. Both can be realized using the method of counterfactual randomization. To represent the distributions associated with these intermediate layers, the paper proposes two graphical models based on CBNs with additional constraints. An inference procedure based on the ctf-calculus [Correa & Bareinboim] is developed to identify these queries, and is proven to be both sound and complete. Finally, the paper offers a detailed discussion on the expressivity and falsifiability of different layers within the causal hierarchy.

**Questions:**

1. Definition 1: is the choice of $\mathbf{x}_i$ unique? It'll be great to clarify it here.
2. If I understood correctly, the definitions imply that queries in $L_{2.25}$ disallow conflicting values of treatments and outcomes, and queries in $L_{2.5}$ disallow conflicting values in the outcomes?
3. I wasn't able to find the reference for counterfactual BN (CTFBN)? Is CTFBN the same as AMWN?

**Ethical Concerns:**

["NO or VERY MINOR ethics concerns only"]

**Final Justification:**

The paper is technically rigorous and contains important results.

**Limitations:**

yes

**Quality:**

4

**Strengths And Weaknesses:**

Strengths:
1. The paper is well-structured and clearly written overall.
2. The two new layers in the causal hierarchy are quite novel and theoretically meaningful. Additionally, the paper includes a comprehensive appendix with detailed background and proofs.
3. I found the examples very effective in aiding readers' understanding of these abstract ideas. Furthermore, the discussion in Section 3 (on expressivity and falsifiability) offers valuable insights from both theoretical and practical standpoints.

Weaknesses:
1. I believe more real-world examples for Layers 2.25 and 2.5 would help demonstrate their importance. I understand their theoretical significance, but how are they practically meaningful (comparable to L2 and L3)? I know additional details are given in the appendix, but it’s still worth emphasizing the importance (ubiquity) of counterfactual randomization in the main paper.
2. Theorem 1: Def 10 in the supplementary material does not describe a constructive process. Did you mean Definition 11 here?
3. Lemma 1: "set CRS(W*) satisfies (i) and (ii) / An(W*) satisfies (I)" — I'm confused about this part. What does "(ii) / An(W*)" mean here?

Minor typos:

5. footnote 2: "B" -> "Appendix B"
6. line 200: "constrains" -> "constraints"

---

> ### Author Rebuttal · Authors · 2025-07-29
>
> We appreciate your time and positive assessment of our work. We provide answers to your questions below.
>
> >*Weakness 1*: I believe more real-world examples for Layers 2.25 and 2.5 would help demonstrate their importance. I understand their theoretical significance, but how are they practically meaningful (comparable to L2 and L3)? I know additional details are given in the appendix, but it’s still worth emphasizing the importance (ubiquity) of counterfactual randomization in the main paper.
>
> Yes, we agree on the importance of emphasizing the significance of the counterfactual randomization action and the practical implications of the two layers of distributions that are derived from this action. In both the introduction and Section 3, we highlighted that the counterfactual randomization action “characterizes the set of  $\mathcal{L}_3$ distributions that are realizable in principle”. To connect with real-world applications, we included Example 5 (ETT) and Example 7 (NDE), which show that several quantities of significant practical interest across the sciences lie within the two new layers. Due to space constraints, more explanations and examples are currently placed in Appendix A and B. That said, we will consider moving more of the intuitive explanations and visual aids (e.g., Figure 10) from the appendix into the main text given the additional one page allowed. We also welcome any suggestions on specific examples or explanations that can make the importance of the layers clearer.
>
> >*Weakness 2*: Theorem 1: Def 10 in the supplementary material does not describe a constructive process. Did you mean Definition 11 here?
>
> Yes, it is Definition 11. We apologize for the confusion since the numbering is not fully consistent. In the version submitted with the supplementary material, the numbers are consistent across the main body and appendix.
>
> >*Weakness 3*: Lemma 1: "set CRS(W*) satisfies (i) and (ii) / An(W*) satisfies (I)" — I'm confused about this part. What does "(ii) / An(W*)" mean here?
>
> In this lemma, we use the slash symbol “/” to distinguish conditions that apply to the two different layers. Specifically, conditions to the left of “/” apply to $\mathcal{L}\_{2.25}$, and those to the right apply to $\mathcal{L}\_{2.5}$. In other words, a distribution belongs to $\mathcal{L}\_{2.25}$ if “$CRS(\mathbf{W}_\*)$ satisfies (i) and (ii)”, whereas it belongs to L2.5 if “$An(\mathbf{W}\_*)$ satisfies (i)”. We did a bit of gymnastics to make things fit in the allotted space, but will clarify this notation in the revised version to make it more immediately interpretable. Thank you for the observation.
>
> >*Question 1*: Definition 1: is the choice of x_i unique? It'll be great to clarify it here.
> We clarify that the choice of $\mathbf{x}_i$ is not unique.
>
> No, the choice of $\mathbf{x}_i$ is not unique. There can be multiple valid assignments as long as they satisfy the two conditions in Definition 1, i.e.,
> "(i) $\mathbf{x}_i\subseteq \mathbf{x}$ and $\cup_i \mathbf{x}_i =\mathbf{x}$; and
> (ii) For any $v_i\in \mathbf{x}$, for all $V_j\in\mathbf{Y}$, if $V_i\in An(V_j)$ in $\mathcal{M}\_{\mathbf{x}\backslash V_j}$, then $v_i\in \mathbf{x}_j$."
> This is illustrated in Example 17 in Appendix B.2. For example, a causal chain $Z\rightarrow X\rightarrow Y$, the $\mathcal{L}\_{2.25}$ distributions indexed by the interventional value set $y$ include $P (Z_y, X, Y ), P (Z, X_y , Y )$, and $P (Z_y , X_y , Y )$. Although the subscripts on $Z$ and $X$ differ across these distributions, they all satisfy the required conditions.
>
> >*Question 2*: If I understood correctly, the definitions imply that queries in L2.25 disallow conflicting values of treatments and outcomes, and queries in L2.5 disallow conflicting values in the outcomes?
>
> Yes, your intuition on ‘disallowing conflicting values of treatment’ is correct. As elaborated in Section 2.1, the distributions in $\mathcal{L}\_{2.25}$ come from fixing “a single value of $X$ across all its children”, while the distributions in $\mathcal{L}\_{2.5}$ allow “each children of $X$ to take a potentially different value”.
>
> Therefore, it is correct that queries in $\mathcal{L}\_{2.25}$ disallow both conflicting values of the treatment variable itself and conflicting subscripts across the outcomes. This is the reason that distributions in $\mathcal{L}\_{2.25}$ are indexed by a fixed “interventional value set” to ensure consistent values over the treatments and subscripts.
>
> In contrast, queries in $\mathcal{L}\_{2.5}$ may involve different value assignments for the same treatment variable across different children, which in turn may induce different subscripts on downstream outcomes. As such, $\mathcal{L}\_{2.5}$ distributions are indexed by a more relaxed “interventional value set”, allowing variation in the treatment assignments, as long as they satisfy the two conditions in Definition 2.
>
> This distinction is illustrated in Example 1 and further visualized in Figure 10 of Appendix A.4:
> Figure 10(b) corresponds to $\mathcal{L}\_{2.25}$, where all outgoing edges of the treatment variable $X$ receive the same value $x$, yielding the distribution $P(X, Y_x, Z_x)$.
> Figure 10(c) corresponds to $\mathcal{L}\_{2.5}$, where different values $x$ and $x'$ are assigned along different outgoing edges, yielding the distribution $P(X, Y_x, Z\_{x'})$.
>
> Finally, if by “conflicting values in the outcomes” you are referring to the same outcome variable appearing more than once with different subscripts (e.g., $P(Y_x, Y\_{x'})$), then yes, you are correct that neither layer allows for it. Following Definitions 1 and 2, each variable $V_i$ appears at most once in the symbolic representation of a distribution in these two layers.
>
> >*Question 3*: I wasn't able to find the reference for counterfactual BN (CTFBN)? Is CTFBN the same as AMWN?
>
> Thank you, we updated the reference to the correct one. The definition and examples of CTFBN can also be found in Appendix A.4 of the submitted paper. A CTFBN is not the same as AMWN, but rather it is a graphical model, similar in nature to CBN, CBN2.25, and CBN2.5. Specifically, a CTFBN encodes a collection of local constraints over distributions in $\mathcal{L}_3$ of the PCH. In contrast, AMWN is an algorithm that takes a CTFBN as input and checks whether a global counterfactual independence is implied by the local constraints specified in the CTFBN definition. One analogy to this is the relationship between the causal diagram and Pearl’s celebrated twin network, where the former encodes the local constraints, and the latter is a data structure that allows for the evaluation of global constraints implied by the local ones.
>
> >*Minor typos*: footnote 2: "B" -> "Appendix B"; line 200: "constrains" -> "constraints"
>
> Thanks, we will correct these typos in the revised version of the paper.

---

> > ### Comment · Reviewer_XJnG · 2025-08-05
> >
> > Thanks for the clarifications. My concerns have been addressed. And right, I meant the same Y with different subscripts. Regarding the reference for ctfbn, is "[6] Juan D Correa and Elias Bareinboim. Counterfactuals — A Graphical Perspective. 2025" published or a working paper?

---

> > > ### Author Response · Authors · 2025-08-05
> > >
> > > Thank you again for your thoughtful engagement and helpful follow-up. You are right, the reference for CTFBN is to a working paper we had early access to. We regret the oversight and had mistakenly assumed it was already public. The relevant definitions and theorems are also present in Chapter 13 (Sec. 13.2.2) of the forthcoming book (publicly available):
> > >
> > > [1] Elias Bareinboim. Causal Artificial Intelligence: A Roadmap for Building Causally Intelligent Systems, 2025.
> > >
> > > We will update the citation in the revision. We appreciate your close attention to this, as well as your constructive engagement throughout the review process.

---

> > > > ### Comment · Reviewer_XJnG · 2025-08-05
> > > >
> > > > Got it. Thanks for the reference.

---

### Decision · Program_Chairs · 2025-09-17

**Decision:**

Accept (poster)

**Comment:**

(a) Scientific claims and findings
The paper addresses falsifiability in Pearl’s counterfactual layer by introducing two novel classes of models for experimentally testable counterfactuals and a sound and complete calculus based on Bareinboim’s CTF framework.

(b) Strengths
Novel theoretical contribution; addresses a fundamental question in causal inference; introduces new model classes and a rigorous calculus; reviewers viewed the work as original and timely.

(c) Weaknesses
Clarity and quality of writing are limited, making the paper hard to follow. The exposition would benefit from clearer explanations, illustrative examples, and stronger contextualization. The work is purely theoretical with no empirical evaluation, and the examples provided are mathematical rather than illustrative. Reviewer pvuV noted that while this is acceptable, the lack of more concrete or less contrived examples (as sometimes seen in related literature) limits accessibility.

(d) Reasons for decision
I recommend acceptance. The contribution is theoretically significant and the reviewers were positive. However, the paper does not merit spotlight/oral due to issues of clarity, absence of evaluation, and lack of accessible examples.

(e) Rebuttal and discussion
During the rebuttal stage, reviewers raised concerns about the clarity of writing, the heavy reliance on a long appendix, and the lack of illustrative examples or empirical/theoretical evaluation. Reviewer pvuV additionally questioned the incremental nature of some of the theorems. The authors responded to these points, clarifying the role of theorems and committing to improving exposition in the main text. While the rebuttal did not introduce new results, the discussion addressed reviewers’ doubts, and by the end all reviewers were satisfied and maintained their positive recommendations.